

# Pre-launch calibration results of the TROPOMI payload on-board the Sentinel 5 Precursor satellite

Quintus Kleipool[1], Antje Ludewig[1], Ljubiša Babić[1,4], Rolf Bartstra[1,3], Remco Braak[1,a],
Werner Dierssen[1], Pieter-Jan Dewitte[1,3], Pepijn Kenter[1,3], Robin Landzaat[1,2], Jonatan Leloux[1,2],
Erwin Loots[1], Peter Meijering[1,2], Emiel van der Plas[1], Nico Rozemeijer[1,2], Dinand Schepers[1,3],
Daniel Schiavini[1,2], Joost Smeets[1,3], Giuseppe Vacanti[1,4], Frank Vonk[1,2], and Pepijn Veefkind[1]

[1]KNMI, Royal Netherlands Meteorological Institute, De Bilt, The Netherlands
[2]TriOpSys B.V., Utrecht, The Netherlands
[3]S&T Science and Technology B.V., Delft, The Netherlands
[4]Cosine B.V., Leiden, The Netherlands
[a]Deceased on fifth of February 2014.

*Correspondence to:* kleipool@knmi.nl

**Abstract.** The Sentinel 5 precursor satellite was successfully launched on 13th October 2017, carrying the Tropospheric Monitoring Instrument TROPOMI as its single payload. TROPOMI is the next generation atmospheric sounding instrument, continuing the successes of GOME, SCIAMACHY, OMI and OMPs, with higher spatial resolution, improved sensitivity and extended wavelength range. The instrument contains four spectrometers, divided over two modules sharing a common tele-

scope, measuring the ultraviolet, visible, near-infrared and shortwave infrared reflectance of the Earth. The imaging system enables daily global coverage using a push-broom configuration, with a spatial resolution as low as 7 x 3.5 km2 in nadir from a Sun-synchronous orbit at 824 km and an equator crossing time of 13:30 local solar time.

     This article reports the pre-launch calibration status of the TROPOMI payload as derived from the on-ground calibration effort. Stringent requirements are imposed on the quality of on-ground calibration in order to match the high sensitivity of the

instrument. In case that the systematic errors that originate from the calibration exceed the random errors in the observations, the scientific products may be compromised. A new methodology has been employed during the analysis of the obtained calibration measurements to ensure the consistency and validity of the calibration. This was achieved by using the production grade Level 0 to 1b data processor in a closed-loop validation setup. Using this approach the consistency between the calibration and the L1b product could be established, as well as confidence in the obtained calibration result.

This article introduces this novel calibration approach, and describes all relevant calibrated instrument properties as they were derived before launch of the mission. For most of the relevant properties compliance with the requirements could be established, including the knowledge of the instrument spectral and spatial response functions, and the absolute radiometric calibration. Partial compliance was established for the straylight correction; especially the out-of-spectral-band correction for the NIR channel needs further validation. Incompliance was reported for the relative radiometric calibration of the Sun port

diffusers. These latter two subjects will be addressed during the in-flight commissioning phase in the first 6 months following launch.





# 1 Introduction

The Sentinel-5 Precursor (S5P) mission represents the first in a series of atmospheric observing systems within Copernicus (Ingmann et al. (2012)). Copernicus is the European programme for the establishment of a European capacity for Earth Observation and is a joint initiative of the European Community and the European Space Agency ESA.

The S5P mission is a single-payload satellite in a low Earth orbit that provides daily global information on concentrations of trace gases and aerosols important for air quality, climate forcing, and the ozone layer. The payload of the mission is the TROPOspheric Monitoring Instrument TROPOMI, which is jointly developed by the The Netherlands and ESA, and consists of a spectrometer with spectral bands in the ultraviolet, the visible, the near-infrared and the shortwave infrared (Veefkind et al. (2012)). The selected wavelength range for TROPOMI allows observation of key atmospheric constituents, including

ozone ($O_3$), nitrogen dioxide ($NO_2$), carbon monoxide (CO), sulphur dioxide ($SO_2$), methane ($CH_4$), formaldehyde ($CH_2O$), aerosols and clouds.

    With its launch on October $13^{th}$ 2017 the S5P mission will avoid large gaps in the availability of global atmospheric products between its predecessors SCIAMACHY (Bovensmann et al. (1999)), GOME-2 (Munro et al. (2016)) and OMI (Levelt et al. (2006)) and the future missions Sentinel-4 and Sentinel-5, scheduled for launch in 2020 and 2021, respectively.

There is a synergy between TROPOMI and the U.S. NPP (Suomi National Polar-orbiting Partnership) satellite. Identified synergies include the use of the VIIRS (Visible / Infrared Imager Radiometer Suite) for high spatially resolved cloud information and OMPS (Ozone Monitoring and Profiling Suite) for high vertically resolved stratospheric ozone profiles. It is planned to fly the S5P mission within approximately 5 minutes of NPP, thus building upon the successes of the "A-Train" constellation of Earth observation satellites.

The Sentinel 5 Precursor satellite has been injected into a near-polar, near sun-synchronous orbit by a ROCKOT launcher. The initial 6 months of in-orbit operation will cover spacecraft, TROPOMI and ground segment level commissioning activities (Phase E1). The E1 Phase will be followed by a 6.5 years exploitation E2 Phase during which the spacecraft will be operated according to a stable, fully repetitive scenario, with systematic processing and archiving of data products within the S5P Payload Data Ground Segment (PDGS). The mission products will be disseminated to both operational users (e.g. Copernicus

Services, national NWP centres, value adding industry) and the scientific user community.

# 2 Instrument description

## 2.1 Instrument overview

The TROPOMI instrument is a space-borne nadir-viewing push-broom imaging spectrometer with four separate spectrometers covering wavelength bands between the ultraviolet and the shortwave infrared, which was designed with OMI heritage. Using

passive remote sensing techniques the instrument measures at the top of the atmosphere the solar radiation reflected by and radiated from the Earth. It operates in a push-broom configuration with a wide swath of $108°$. Combined with a polar circular orbit of $824\,\mathrm{km}$ altitude, the wide swath allows TROPOMI to achieve complete daily global surface coverage. Light from the



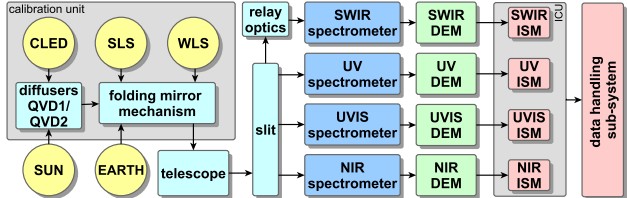

**Figure 1.** Schematic overview of the TROPOMI instrument. Shown are the different light paths from Earth, Sun and the internal sources CLED, WLS and SLS to the TROPOMI spectrometers. The detector read-out is performed by DEMs, the DEM control and data acquisition by the ISMs.

**Table 1.** Main spectral characteristics of the four TROPOMI spectrometers and the definition of the TROPOMI spectral bands with identifiers 1–8. The listed values are based on on-ground calibration measurements and valid at the detector centre. The performance range is the range over which the requirements are validated, the full range in general is larger.

| Spectrometer | UV | | UVIS | | NIR | | SWIR | |
|---|---|---|---|---|---|---|---|---|
| Band ID | 1 | 2 | 3 | 4 | 5 | 6 | 7 | 8 |
| Performance range [nm] | 270–320 | | 320–490 | | 710–775 | | 2305–2385 | |
| Spectral range [nm] | 267–300 | 300–332 | 305–400 | 400–499 | 661–725 | 725–786 | 2300–2343 | 2343–2389 |
| Spectral resolution [nm] | 0.45–0.5 | | 0.45–0.65 | | 0.45–0.35 | | 0.227 | 0.225 |
| Spectral dispersion [nm per pixel] | 0.065 | | 0.195 | | 0.125 | | 0.094 | |

entire swath is recorded simultaneously and dispersed onto 2-dimensional imaging detectors: the position along the swath is projected onto one direction of the detectors, and the spectral information for each position is projected onto the other direction.

TROPOMI utilizes a single telescope to image the target area onto a rectangular slit that acts as the entrance slit of the spectrometer system. There are four different spectrometers, each with its own optics and detector: mediumwave ultravi-
5 olet (UV), longwave ultraviolet combined with visible (UVIS), near-infrared (NIR), and shortwave infrared (SWIR). The spectrometers for UV, UVIS and NIR are jointly referred to as UVN. The detectors for the UVN spectrometers are charge coupled devices (CCDs). The SWIR part of the instrument is much colder than the UVN part and uses a complementary metal–oxide–semiconductor (CMOS) detector. An optical relay separates the SWIR part thermally from the rest of the instrument. The interface is the pupil stop for the SWIR spectrometer.

TROPOMI has calibration light sources on board: a spectral line source (SLS) consisting of five laser diodes in the SWIR range, a white light source (WLS), a common light emitting diode (CLED) and detector LEDs (DLEDs). The detector LEDs are situated close to each of the detectors, while the other sources are located in a calibration unit. A schematic overview of TROPOMI is shown in Fig. 1.

Via different settings of the calibration unit, the telescope receives light from different sources: the Earth, the Sun and the
15 on-board calibration light sources. Additionally, the telescope can be closed off.



**Table 2.** Main optical design parameters of each TROPOMI spectrometer.

| Spectrometer | UV | UVIS | NIR | SWIR |
|---|---|---|---|---|
| Telescope entrance area [mm$^2$] | 25 | 25 | 25 | 25 |
| Spatial focal length [mm] | 34 | 34 | 34 | 29 |
| Spectral focal length [mm] | 68 | 68 | 68 | 29 |
| Spatial slit size [mm] | 64 | 64 | 64 | 25.5 |
| Spectral slit size [mm] | 0.56 | 0.28 | 0.28 | 0.308 |
| Spatial f-number | $F$/9 | $F$/9 | $F$/9 | $F$/1.33 |
| Spectral f-number | $F$/10 | $F$/10 | $F$/10 | $F$/1.33 |
| Spectral IFOV [°] | 0.48 | 0.24 | 0.24 | 0.25 |
| Spatial sampling distance [°] | 0.125 | 0.125 | 0.125 | 0.42 |
| Spatial sampling angle along-track [°] | 0.50[a] | 0.50 | 0.50 | 0.16 |
| Spatial sampling angle across-track [°] | 0.50 | 0.50 | 0.50 | 0.059 |

[a] For band 1 the spatial sampling angle is 1.5° and 2.0° in along- and across-track direction respectively.

Each of the detectors is divided in two halves, which yields a total of eight spectral bands. Table 1 summarizes the main characteristics of each of the TROPOMI optical spectrometers and the definition of the spectral bands.

## 2.2 Telescope

The telescope is a two-mirror reflective telecentric telescope that follows an $f$–$\theta$ law, i.e., it has the property of creating a flat
image field at the plane of interest, with a focused beam that is always perpendicular to that plane. It has a large field-of-view (108°) in the across-track (swath, across-flight, or spatial) direction, and a small field-of-view in the along-track (along-flight or spectral) direction. The two telescope mirrors are referred to as primary and secondary mirror, in the order in which light from the Earth passes through the telescope. A strip on the Earth's surface is imaged by the primary mirror. The intermediate image, located close to the primary mirror, is re-imaged by the secondary mirror on the entrance slit of the spectrometer system. At
the same time, the entrance pupil is imaged to infinity. Both mirrors are concave and aspherical. In the optical path between the two mirrors, coincident with the intermediate pupil, there is a polarization scrambler preceded by a rectangular aperture. It is this aperture that determines the telescope's throughput. The polarization scrambler comprises of two wedge pairs. The edges are optimized in thickness to cancel out the birefringence of each wedge pair along the optical axis.

The width of the entrance slit defines the field-of-view in the spectral (along-track) direction. The light for the UVIS and
NIR spectrometers passes through the slit, while light destined for the UV and the SWIR spectrometers is reflected from the side of the slit. Both the SWIR and the UV spectrometers have their own slit. A consequence of this arrangement is that the UV and the SWIR are not co-registered with the UVIS and the NIR. The light detected in the UV and SWIR originates from another position than the light detected in the UVIS and NIR. The difference in flight time between the two positions is about





seconds, which corresponds to two read-outs in the baseline configuration. Some of the key geometrical parameters of the instrument are given per spectrometer in Table 2.

## 2.3 UVN module

The three UVN spectrometers are conceptually almost the same: each images a slit on a detector, dispersing the light by means

of a grating. The UVIS and NIR spectrometers share the same 280 µm wide slit. Light for the UV and SWIR spectrometers first reflects off the UVIS-NIR slit and then, after passing through a dichroic that directs the shortwave infrared component of the light towards the SWIR relay optics, is imaged on a second slit on a conjugate plane to the focal plane of the telescope. This slit is 696 µm wide on its conjugate plane, which corresponds to a virtual slit with a width of 560 µm. Using cylindrical optics, the slit is imaged only in the spectral dimension. This removes the strict requirement on the sharpness of the slit edges

in the spatial dimension.

At the end of the spectrometer, the light falls onto a charge coupled device (CCD). One direction of the CCD corresponds to the spatial (across-track) dimension, the other direction corresponds to the spectral (along-track or flight) dimension. The CCD pixel size is 26 µm × 26 µm, and the total number of pixels in the imaging area is 1024 × 1024. The image of the slit in the across-track direction is about 862 pixels wide, the remaining pixels being used for calibration and monitoring purposes.

The dimensions of the slit image on the detectors are summarized in Table 3. During data acquisition, pixels can be binned in the spatial direction to decrease noise at the cost of resolution.

### 2.3.1 UVN spectrometers

The beam from the telescope is imaged on the slit telecentrically, as mentioned above. Light for the UV spectrometer actually reflects off the slit, therefore this first slit is not yet the limiting slit for the spectrometer. The reflected image is re-imaged

by a collimator and a folding mirror to the actual entrance slit (UV slit) of the spectrometer. A dichroic separates the SWIR band from the UV band. Folding mirrors, one of them out-of-plane, guide the light onto the grating. The diffracted light passes through the imaging optics. The imaging optics consist of three lenses, all three de-centred and tilted with respect to the optical axis in order to get a good co-registration performance and to remove unwanted specular reflections ("ghosts") from the system. The last lens and the CCD plane are tilted, in order to correct for axial colour aberrations. To reduce the amount of spectral

straylight that could reach the detector, a spatially varying coating is used on the last surface before the detector. This surface is the flat side of the last lens. At each location on the lens the coating transmits light of the expected wavelength, and it reflects light whose wavelength is 15 nm larger.

For the UVIS spectrometer, the light is collimated after passing through a dichroic. Via four folding mirrors the light is guided to the grating. Just as in the UV spectrometer, there is an out-of-plane folding mirror. The grating disperses the light,

and the subsequent imaging system images the slit on the detector. The imager consists of five lenses, the last one has a flat side facing the detector.

For the NIR spectrometer, the slit image is guided via a dichroic and three folding mirrors and a collimating lens, onto the grating. The dispersed light is imaged on the detector by three de-centred lenses.





**Table 3.** Dimensions of the slit as imaged on the detectors. The dimensions (spectral × spatial) include the spectral smile.

| Spectrometer | UV | UVIS | NIR | SWIR |
|---|---|---|---|---|
| Performance range [pixels] | $778 \times 862$ | $882 \times 862$ | $551 \times 862$ | $960 \times 215$ |
| Full range [pixels] | $994 \times 857$ | $994 \times 858$ | $994 \times 856$ | $960 \times 215$ |

Each UVN spectrometer contains LEDs to illuminate the detectors directly. These detector LEDs (DLEDs) are used for calibration and performance monitoring purposes. The DLEDs emit green light with a wavelength of 570 nm. The response of the individual UVN detectors to the DLED is not identical, as the optical coatings differ per spectrometer.

### 2.4  SWIR relay optics

Relay optics are necessary to be able to thermally separate the SWIR spectrometer from the UVN spectrometers. The SWIR spectrometer is cooled to around 205 K and its detector to 140 K, while the common telescope, calibration unit and UVN optics are much warmer (290 K). The light from the telescope is imaged onto a pupil at the entrance of the SWIR spectrometer. The reflective slit, the collimator lens, and the front surface of the dichroic are shared with the UV spectrometer. The dichroic is made of silicon, providing a good filter function for a wide range of wavelengths. The pupil stop of $9 \times 18 \, \mathrm{mm}^2$ is at the end of

the relay optics. This stop deliberately vignettes the beam, as this is the stop for the SWIR spectrometer. There is some amount of chromatic aberration, leading to a small magnification error of 0.008 %.

### 2.5  SWIR module

Functionally, the SWIR spectrometer is similar to the UVN spectrometers. The optical system is divided into six sections: SWIR entrance pupil, telescope, slit prism, collimator, grating, and imager. The imager interfaces with the focal plane array

within the detector module.

The SWIR entrance pupil is formed by the UVN telescope and SWIR relay optics, and it forms the optical interface between the UVN and SWIR subsystems. The SWIR telescope receives its input from the main UVN optics and focuses it on a slit prism. The slit delimits the spatial extent of the image, defining the along-track field-of-view, as well as the spectral resolution (in combination with the collimator and the grating). The collimator then transfers the image of the slit to a grating that causes

spectral separation. A final imager forms an image of the scene on the SWIR detector. The DLED for SWIR is located close to the detector and emits light at around 2200 nm.

### 2.6  Calibration unit

The calibration unit contains two rotation mechanisms: one diffuser carousel and one folding mirror. The diffuser carousel includes two quasi volume diffusers (QVD) and has six defined positions: one position for each of the two Sun diffusers

(QVD1 and QVD2), one position for the white light source (WLS), one position for the spectral line source (SLS, laser diodes)



and two positions for the common LED (CLED). When the diffuser carousel is in the WLS position, the calibration port towards the Sun is closed, so that when the WLS is switched off, a dark measurement can be performed. One of the two Sun diffusers (QVD2) is used to monitor optical degradation and is employed only occasionally for Sun measurements during operation.

The second rotation mechanism deploys a folding mirror in front of the telescope. This folding mirror directs the light from
the Sun or the calibration sources towards the primary mirror of the telescope. The folding mirror is curved such that the large field-of-view of the telescope can be completely illuminated. When the folding mirror is deployed, the nadir view is blocked. This means that this mechanism can also function as an instrument shutter.

When the telescope looks directly to Earth, the diffuser carousel is in the closed position to block off the Sun port. For a Sun calibration, the diffuser carousel is rotated such that it opens the Sun port and directs the light towards the deployed folding
mirror. The folding mirror itself closes the nadir port. In the monitoring diffuser mode (QVD2), the carousel is rotated by 180° with respect to the other diffuser mode. For the measurements with Sun and CLED the light passes through the diffusers, for WLS and SLS the light is reflected off the side of QVD1 and QVD2 respectively. For calibration with the WLS the diffuser carousel is placed in the same position as when closed. The difference is that the folding mirror is deployed so that the telescope receives light from the calibration source. A curved mirror on the back of the Sun diffuser reflects light from the WLS towards
the folding mirror. The monitoring diffuser has a flat diffusing surface to reflect the light from the SLS towards the telescope. The optical path of the CLED can be chosen to be either through the Sun or the monitoring diffuser. The CLED light passes through the same optical elements as the Sun, but then in opposite direction. In CLED mode, the light of the Sun is closed off.

## 2.7 UVN Detectors and electronics

The three UVN spectrometers are equipped with separate but identical detector modules (DEMs) of the same type and config-
uration. Each UVN-DEM is equipped with a CCD sensor and front-end electronics, providing the raw digitized measurement data to the instrument control unit. The SWIR module has a CMOS detector and a different DEM than the one used in the UVN module, and a description can be found in Hoogeveen et al. (2013).

### 2.7.1 DEM construction

The main structure of each of the UVN-DEMs is the detector-side housing which is an aluminium frame. It is the interface
to the optical bench on the outside of the UVN-DEM. The support for the focal plane assembly, including the detector circuit and the thermal bus unit (TBU) side housing, are mounted onto the detector housing. The walls near the CCD are designed with an increased thickness in order to maximize radiation shielding. The focal plane assembly consists of the detector circuit including detector, the heat sink, the detector mount and the thermal strap. Its functions are to provide mechanical, thermal, and electrical connections between the detector and the respective interfaces. The TBU-side housing acts as the closing lid of
the UVN-DEM and is also the mechanical interface for the main circuit card and for the thermal strap, and acts as the thermal interface to the warm TBU stage.

One of the main difficulties of the UVN-DEM thermal design is that the CCD has to remain at a much colder temperature than the electronics boards, which are in the same housing. This creates temperature gradients and the main task of the thermal



design is to control these gradients so that both CCD and electronics can be operated in their own temperature range, which is 208 K for the three UVN CCDs, and 294 K for the three UVN DEMs. Critical to the scientific performance is the insulation of the detector from the electronics because their operating temperatures differ so much. This is achieved by encapsulating the CCD in a box shaped multi-layer insulation. As the CCD has to be operated at cold temperatures, it is thermally linked to a

cold finger interface. Additionally, the detector is mounted on a hexapod whose legs provide the required thermal insulation from the rest of the housing.

### 2.7.2 CCD layout

The UVN CCDs are constructed with two sections, namely an image section in which the light flux is accumulated during exposure, and a storage section in which the signal is stored for read-out, while the next image is accumulated in the image

section. The storage section is shielded from incident light by a mask, but otherwise identical to the image section. The detectors are frame transfer type CCDs, which means that the pixels of the image and storage sections and the read-out register are connected into vertical shift chains, i.e., the charges of the pixel cells can be shifted down line by line. A frame transfer moves the contents of the image section into the storage section. The transfer is an operation in which 1026 line shifts are performed in quick succession shifting in zero-charge lines from the top and shifting the charge of the bottom lines into the

read-out register. At the end of the frame transfer, the entire image section, as well as the top-most line of the storage section, are emptied of charge and the old image section contents are moved to the 1024 bottom-most lines of the storage section. After the frame transfer, a new exposure starts in the image section, while the finished exposure is read out from the storage section. The frame transfer operation is handled internally by the UVN-DEM, upon reception of a command to initiate a frame transfer. A two-phase clocking scheme is used to achieve a frame transfer time of $\leq 800\,\mu s$.

Once the frame is transferred to the storage section, the read-out starts. Only the charge in the storage section is shifted down one line at a time, leaving the contents of the image section unaffected. The operation shifts a zero-charge line into the top-most line of the storage section and the charge of the bottom-most line is shifted into the read-out register. The charge shifted into the read-out register is added to the charge already there. The process of shifting multiple lines into the read-out register before it is read out (and thereby emptied) is called binning or row binning.

The advantages of row binning are that it decreases the data rate and increases the signal-to-noise ratio. The disadvantage of binning is that resolution (information) is lost, as it is an irreversible process. The row binning factor is not necessarily the same across the CCD, but can be varied through instrument configuration. The binning can only be performed in the row direction, corresponding to the spatial dimension. This allows, for example, to influence the ground pixel size across-track or to bin all the rows from shielded rows of the image section into a single row. Spectral binning in the column direction is not possible.

Once the charge is shifted into the read-out register, it can be read out by shifting the charge pixel by pixel (column by column) onto the output node(s) of the detector. The TROPOMI UVN CCD use two output nodes, one at each side of the register, effectively dividing each detector into two bands. The read-out register has four pixels on each side in addition to the 1024 pixels (columns) of the image and storage sections of the detector. These pixels, referred to as pre-scan and post-scan





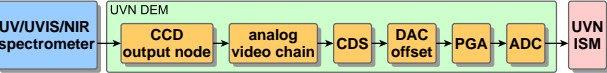

**Figure 2.** Schematic of the read-out chain of the UVN detectors. In the output node charge is converted to a voltage, the analog video chain amplifies the signal with a fixed gain, the signal is detected in the correlated double sampling unit (CDS), an analog offset can be added (DAC) and the signal can be amplified by the programmable gain amplifier (PGA) before its conversion to digital units by analog-to-digital converter (ADC).

pixels can be read out and used for calibration or monitoring purposes. In fact, it is possible to even read past the post-scan pixels; the resulting pixels are referred to as overscan pixels and can be used for calibration or monitoring purposes as well.

### 2.7.3   CCD video chain

Figure 2 shows the different elements of the read-out chain for the UVN detectors. The output amplifier, which is connected to
the output node, provides two gain settings: these are programmable according to the expected signal strength.

The main characteristics of the electronic circuitry associated with the sensor are summarized in Table 6. The individual pixels' full-well capacity amounts to about $0.8 \times 10^6$ electrons ($e^-$). After the output node, the signal of each of the detector bands is passed through a video amplifier with a fixed gain and then processed by a National Semiconductor LM9864QML integrated circuit. This circuit contains a correlated double sampling unit (CDS), a unit where a digital programmable analog
offset can be added (DAC), an 8-bit programmable gain amplifier (PGA) and a 14-bit analog-to-digital converter (ADC). The CDS can amplify the signal by a factor 2. Apart from the CDS gain, all values can be set per detector band.

### 2.8   Instrument control unit

The instrument control unit (ICU) is a unit consisting of eight boards with software running on a LEON processor. The main interfaces to the satellite platform are Mil-Bus for command and control and, SpaceWire for science data output.
The ICU hardware includes the following parts: the processing function for the SpaceWire bus; on-board time, synchronization, and clock functionality; image processing and packetization, implemented in one field-programmable gate array (FPGA) per DEM; an ICU interface to the four DEMs; thermal control hardware drivers; and a power supply providing ICU internal power and secondary power to the DEMs. The ICU hardware is fully redundant except for the DEM interfaces and image processing, which are implemented as one non-redundant block per detector. The ICU is powered using two redundant power
lines.

The ICU software performs the following main tasks: command and control, including command reception, scheduling and execution; parameter management; thermal control; internal and external data acquisition and generation of house-keeping and engineering data packets; science data management; and on-board time synchronization control. The ICU is equipped with four identical, instrument specific modules (ISM), each controlling one DEM. For the three UVN-DEMs, the ICU controls
the details of each frame transfer, line transfer, and line read-out. The UVN detectors are synchronized to minimize EMC



effects, such that no frame transfer can occur while a measurement is read from the register. In the case of the SWIR-DEM, the ICU only provides a signal to start the image acquisition, the details being left to the SWIR-DEM itself. For the UVN CCD detectors, each detector is split into two halves as dictated by the detector design. This corresponds with the two bands per CCD detector, for consistency, the SWIR CMOS detector/spectral range is divided into two halves/bands as well.

The received pixel data arrives row by row, i.e. all pixels of the first row are received first, followed by all pixels of the next row, and so on. Depending on the binning scheme used (applicable to UVN only), a read-out of an entire exposure might consist of any number of rows from 0 to 1024 for each detector half. Image processing in the ICU is performed independently for the halves of a detector.

Up to 800 pixels (columns) can be read out from each UVN detector half. The SWIR detector has 500 columns per band,
and 256 rows. One additional virtual row can be read-out for monitoring purposes. The co-addition factor can be set for all bands to be between 0 and 512, and it can be programmed separately per band. It is possible to co-add up to 256 consecutive images.

Information concerning the individual signals of a pixel that contribute (i.e. add up to) to a co-addition is lost, with one exception. One configurable detector column per band is also stored separately for every exposure/co-addition of an image.
The data for these "small-pixel columns" is included in the science data and provide information on a higher spatial resolution than the data for other columns.

## 3   Calibration approach

### 3.1   Purpose of calibration

For the retrieval of absolute densities of atmospheric constituents, the TROPOMI instrument needs to be calibrated prior to
launch with respect to known radiometric sources. During the operational phase relative calibrations using the Sun as a source are performed. Calibration measurements are used to derive Calibration Key Data (CKD) that is required for the L01b processor to compute absolute geolocated radiance values from the raw instrument L0 data (KNMI (2017)). From the calibration of OMI (Dobber et al. (2006)), the precursor of TROPOMI, several lessons were learned with respect how to validate the results obtained. The most important improvement with respect to OMI is the use of the production grade L01b processor during the
calibration campaign, and in the analysis of all measurement data.

### 3.2   Use of the L01b processor

During the on-ground calibration of the TROPOMI instrument the production-grade L01b processor is directly employed to determine CKD from the calibration measurement data to ensure that a consistent set of CKD is derived. Furthermore the processor is used to both verify and validate the CKD by reprocessing the measurement data, analysing the remaining
uncorrected effects and determining the CKD independently from different data sets. The use of the L01b processor ensures that the same algorithms are used in the correct order for all data and no additional sources for errors and inconsistencies are





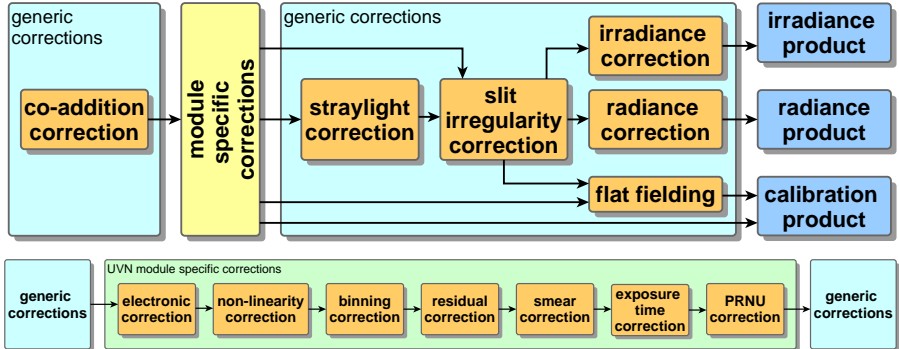

**Figure 3.** Top panel: High level overview of the different processing steps in the L01b processor. For the output products "radiance", "irradiance" and "calibration" different algorithms are applied. The generic corrections are applied to all detectors. Bottom panel: For the UVN specific part, the algorithms correct for the different detector and read-out features. Note that not all correction steps are shown.

introduced. The L01b processor also computes the error and the noise associated with the processed signal, which has the advantage that an error margin is provided for all CKD. Figure 3 gives a high level overview of the different processing steps in the L01b processor for the instrument and UVN specific corrections.

The advantage of using the production grade processor over a separate prototype processor is that the former is intended to
be used for systematic data processing in an production environment (the PDGS), and therefore is capable of digesting large volumes of data. In addition a production grade processor must meet all sorts of requirements on reliability and accuracy, which have to be validated before the software is accepted. A disadvantage is that such a processor is more difficult to operate and configure than a prototype processor, which calls for additional software to harness it, for TROPOMI this was implemented as the Calibration Framework.

The measurements and requirements for the used sources for the on-ground calibration of TROPOMI were devised by the science team under the lead of the Royal Netherlands Meteorological Institute (KNMI) and assisted by the Space Research Organization the Netherlands (SRON) for the SWIR part. The implementation and execution of the calibration measurements was performed by the industry team under the lead of Airbus Defence & Space The Netherlands (ADSNL).

### 3.3   Calibration ground support equipment

A high level overview of the different sub-systems which are part of the calibration setup are shown in Figure 4. From the TROPOMI central checkout system (CCS) all systems can be controlled manually or via scripts. All operations of the instrument, optical-, electrical- and mechanical ground support equipment (OGSE, EGSE, MGSE) were performed by ADSNL and sub-contractors. The CCS was manned 24/7, allowing for a 24/7 measurement campaign. The CCS also shows the raw data near-real time and allows for functional checks of the equipment. The instruments is controlled via the electrical ground
support equipment (EGSE).





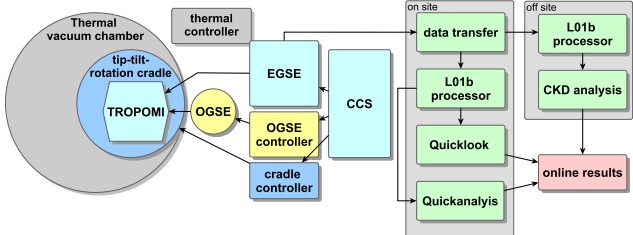

**Figure 4.** Calibration facilities. The instrument TROPOMI is placed on a tip-tilt-rotation stage ("cradle") inside a thermal vacuum chamber. The optical ground support equipment (OGSE), the cradle and the electric ground support equipment (EGSE) are all commanded from the the central checkout system (CCS). The measurement data and ancillary data from cradle and OGSE are transferred to an on-site server. Locally quicklook and quickanalysis is performed, the derivation of the CKD is performed off-site. Preliminary results and the quicklook data are made available online for all parties concerned.

The measurement data and ancillary data from cradle and OGSE are transferred to an on-site server (PRISM). The L01b processor is running locally on the measurement data and dedicated software combines the ancillary data from the setup with the measurement data. In parallel, the L0 data is transferred to off-site premises at the KNMI. Locally, quicklook and quickanalysis is performed, the derivation of the CKD is performed off-site. Preliminary results and the quicklook data are

made available online for all parties concerned. Changes to setups and individual measurement sessions are recorded in a an online-logbook.

From the EGSE the data is directly transferred to and processed on-site by the L01b processor by KNMI. Together with dedicated quicklook and analysis tools this provides the opportunity to do on-site near real time analysis on of the quality of the measurement. Also the L01b processor is tested in an operational scenario. During the calibration campaign there is not

enough time to do the whole CKD derivation to assess whether the measurement is successful. Instead, pre-processing with the L01b processor is used by the on-site science team to assess the quality of the measurement itself in terms of e.g. signal-to-noise and illumination homogeneity to be able to derive the CKD with sufficient accuracy. The capability to perform this quality assessment in parallel with the commissioning of a new measurement setup by the industry team was a key success driver for completing the calibration campaign within the given timeframe. The calibration key data itself was derived on

separate processing infrastructure by separate off-site science teams at KNMI (for UVN and instrument key data) and SRON (for SWIR specific key data).

## 3.4  Calibration facility

The on-ground calibration for TROPOMI was performed in a $6.5\,\text{m}$ diameter thermal vacuum tank located in a class 10000 (ISO7) cleanroom at the Centre Spatial de Liège (CSL) in Belgium. The campaign started end of December 2014 and lasted

until begin of May 2015. A thermal shroud cooled by liquid nitrogen ensured that the instrument was kept at operational temperatures during the entire calibration campaign. The instrument was mounted on a tip-tilt stage stacked on top of a rotation stage. This "cradle" allowed to calibrate the full instrument swath ($108°$) under operational conditions. The thermal vacuum



chamber has a single 30 cm diameter window. By moving the instrument with the "cradle" in front of the window, all swath angles can be illuminated one by one. Prior to the calibration campaign, pre- and post-environmental test were performed to assess the instrument's stability with respect to vibrations and thermal cycling. The on-ground calibration was split into two parts: the Earth port configuration and of the Sun port configuration. During a vacuum break between the two parts, the

instrument was repositioned on the cradle to be able to perform the Sun port calibration. For the re-positioning the area around the thermal vacuum chamber was kept at cleanroom class 100 (ISO5).

## 4  Detector calibration

In this section we will discuss the calibration of the CCD detectors of the UVN module, and the pixel response non-uniformity (PRNU) calibration of all detectors. All other detector calibration results of the CMOS SWIR detector are reported in Hoogeveen

et al. (2013).

Apart from the actual illuminating signal, the signal recorded by the CCD consists of a number of additional components. These components, dark current, electronic offset, residual and smear are all additive and need to be corrected for in the L01b processor. The calibration data for these components was derived using mainly internal instrument sources namely, dark and DLED. Only for the PRNU an external source was used.

### 4.1  Detector dark current calibration

Dark current is defined as the dark current flux times the elapsed time during which signal is acquired. The dark current flux itself can vary from pixel to pixel, and also depends on detector temperature. The temperature is constant over the entire detector, and therefore over all detector pixels. The detector temperature is actively thermally controlled. According to theory, each additional degree Kelvin causes an increase of the dark current of less than 3 %. Since the expected temperature fluctuations

are far smaller, the temperature dependence is ignored in the L01b correction.

The dark current is modelled as the sum of both the image section dark current and storage section dark current. The elapsed time associated with the image dark current is the net exposure time. During the transfer of the frame in the CCD image section to the CCD storage section, additional dark current is formed similar to smear. The storage dark current built up during the storage time, is a function of the row number.

Unbinned background measurements are used in this analysis, for at least two different exposure times. These are preprocessed up to and including binning correction. The image section dark current per pixel is derived as the slope of the graph of pixel signal against exposure time. The intercept of this graph contains a row dependency because of the residence time of the charge in the storage section. Therefore, the (smear-corrected) intercepts of one detector column form a linear function of the residence time; the slope of this function is the storage section dark current.

Specific dark current measurements were obtained with three different exposure times, the longest being more than 500 s, and are only taken at nominal temperature. The signals are amplified with a factor of approximately 15 using the CCD and PGA electronic gains.





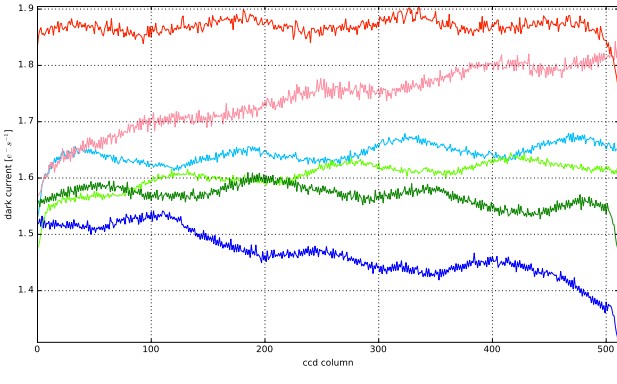

**Figure 5.** Image dark current flux of all UVN bands averaged over the row dimension. Clearly, some low-frequency patterns are visible, while the flux seems to drop at the most extreme columns.

Per pixel, a standard linear regression least-squares fit is made, the image area dark current flux is now the linear coefficient. Figure 5 shows the row averaged dark current flux in the image section for all UVN bands. The average dark current is lower than $2\,\mathrm{e^- s^{-1}}$, and thus very low. The average per band is shown in Table 6.

### 4.2 Detector smear

The purpose of the smear correction algorithm is to negate the effect of frame transfer smear, which is the added contribution of the signal due to the finite transfer time. The smear is always positive and the correction consists of a subtraction. With detailed knowledge of the row transfer process it is possible to compute the exact amount of smear generated. The detector smear algorithm is therefore a analytical correction in the L01b data processing chain that requires no calibration key data. For more detail the reader is referred to L01b data processor ATBDKNMI (2017).

### 4.3 Electronic conversion

The L01b processing chain contains numerous multiplicative factors that, taking into account the appropriate offset corrections, scale the measured signal from digital numbers (DN) to radiance or irradiance. These factors are, DN-to-voltage conversion, electronic gain, PRNU, slit irregularity, radiance and irradiance responsivity.

   Here, we concentrate on the electronic gain that converts a signal in volts back to the detected charge in a pixel. In the

forward direction, there are, among others, the CCD output node, the CDS amplifiers and the PGA amplifier. The former has two settings, both convert the signal from electrons to volts.

   A reference, or neutral gain setting is defined using the low CCD gain. The factor between a particular (PGA, CDS, CCD) setting and the neutral gain setting is called the relative gain. The absolute charge-to-voltage factor is a design value, directly derived from the industry specifications of approximately 0.80 μV/e⁻ for the low CCD gain. This value does not have to be

directly calibrated, as any deviation from it will automatically be absorbed by consecutive multiplicative calibrations.





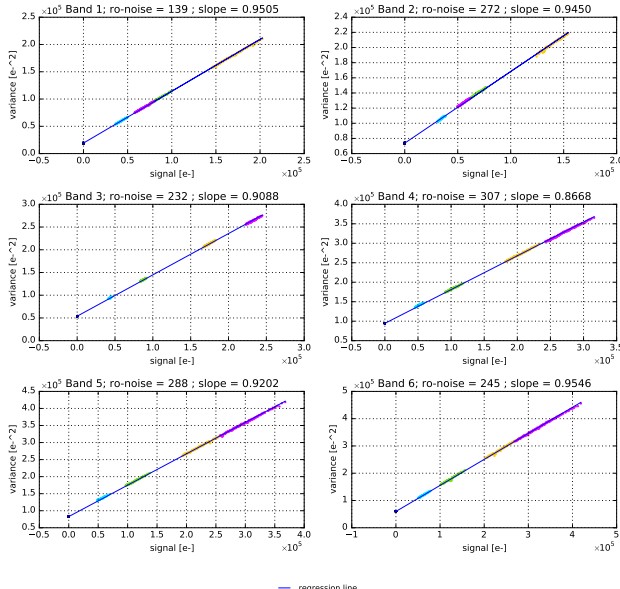

**Figure 6.** Signal versus variance for all UVN bands: Signals and corresponding temporal variances, sorted and block-wise averaged for several dark and illuminated instrument settings. The shot noise law is clearly visible: all points are close to the overall regression line. Only slight residuals as function of the signal magnitudes seem to remain.

The charge-to-voltage factor defined per band, leads to alignment problems between any two detector halves. Thus, only the relation between the two conversion factors has been calibrated, but not their absolute value. The observed shot-noise is a means to indirectly calibrate this absolute gain.

Since the noise of each signal consists of read-out noise and shot noise, and shot noise is a function of signal magnitude, it is possible to draw a graph of signal versus total noise (as variance). This graph is a straight line, and the intercept is the read-out noise. We use repeated DLED measurements with different exposure times and high repetitions to confirm the shot noise law. In Fig. 6 the result is shown, clearly indicating that the shot noise law applies. The slope is not exactly one, so we have to conclude that the design value for the voltage-to-charge conversion is not entirely correct. This is hardly surprising, since the factor is difficult to establish directly.

## 4.4 Detector non-linearity

The charge-to-voltage conversion at each CCD output is not entirely linear. Therefore, in the TROPOMI L01b processor, the voltage-to-charge conversion algorithm is followed by a non-linearity correction algorithm. The non-linearity is defined as the difference between the expected linear behaviour and the actually measured read-out register charges.

The non-linearity can be a non-trivial function of the measured register charge and is therefore provided to the L01b processor in the form of a number of Chebyshev expansion coefficients. This parametrized non-linearity function depends on the detector





band and the CCD gain setting (high or low). A Chebyshev approximation is valid only on a particular signal interval. For values below that interval, which may occur due to noise, the correction is clipped; for values above that interval the correction is carried out regardless (essentially extrapolating), but the data is flagged as saturated. The Chebyshev polynomial function is evaluated for the measured pixel signal. The outcome (i.e. the non-linearity) is subtracted from the signal.

Available are $n$ DLED-measurements with the same settings but varying exposure times. To distinguish between the register and the pixel non-linearity binned measurements are used to determine the former. The binning factor is five for the register non-linearity measurements. Per pixel, the measured signal $S_i, \quad i = 1 \cdots n$ values lie on a curve. The *expected* linear behaviour is that the signals lie on a straight line between the origin and point $(t_{\max}, L_{\max})$ where $L_{\max}$ is predefined and has the same value for all pixels in a band. $L_{\max}$ is reached at different exposure times, depending on the response of each pixel. As a rule

of thumb, $L_{\max}$ is chosen such that $S_{(n/2)} \approx L_{\max}$ for a pixel with average response. The difference between the constructed straight line and the curve of $n$ measured points $(t_i, D_i)$ is converted to a non-linearity as function of the measured charge $(S_i, D_i)$. The combined set of the non-linearities of all pixels can be modelled by a low-order polynomial curve. The 1-$\sigma$ width of the cloud of points (of which the non-linearity curve is the least-squares approach) forms the greater part of the CKD error.

There is one degree of freedom in establishing the non-linearity: at a charge of $L_{\max}$, the CCD responds with a certain

deviation $d$ (not necessarily zero) of the linear behaviour. Together with the constraint that the non-linearity for a zero signal is zero (otherwise an additional offset would exist), there are thus two defined points of a straight line: the "linearity".

Now, for each pixel we can draw a graph $S_{\mathrm{meas}}(t)$ of signal versus exposure time from the $n$ available measurements $(t_{\exp,i}, S_i[\mathbf{x}_{\mathrm{img}}], \quad i = 1 \cdots n$. If the response of the pixel is moderate (i.e. the pixel is not dead or shielded), the time $t_{\max}$ corresponding to the signal value $L_{\max}$ can be defined. The line through the origin and the point $[t_{\max}, L_{\max} - d]$ is the

*expected* linear signal $S_{\mathrm{lin}}(t)$. The difference between the measured curve and the straight line of the expected signal is the non-linearity. For this pixel, there are then $n$ points $[S_i[\mathbf{x}_{\mathrm{img}}], S_i[\mathbf{x}_{\mathrm{img}}] - S_{\mathrm{lin}}(t_{\exp,i})]$. The set of all non-linearity pairs of all pixels, depicted as a cloud in Fig. 7, can be approximated by a low-order polynomial curve. The curve is constrained to intersect the origin and the point $[L_{\max}, d]$. These two parameters are chosen per band. For the bands 4–6, $L_{\max}$ is one million; for bands 1–3, the pivoting points are $320 \, \mathrm{ke}^-$, $500 \, \mathrm{ke}^-$ and $820 \, \mathrm{ke}^-$, respectively. The non-linearity remains within $2\,\%$ and is known

better than $0.6\,\%$ after several validation tests.

### 4.5  Pixel full well capacity

Besides the register non-linearity, which was investigated by using measurements of an average binning factor 5, the pixel non-linearity was assessed. Although not a CKD, it is important to know the maximum amount of charge that an individual pixel can hold (pixel full well). This is measured with unbinned measurements. However, any non-linearity at pixel level cannot be

distinguished from the register non-linearity since the charge inevitable passes through this register. Therefore the assumption is that the pixel signal behaves perfectly linear up to some fraction of its pixel full well; unbinned measurements should not surpass this level.





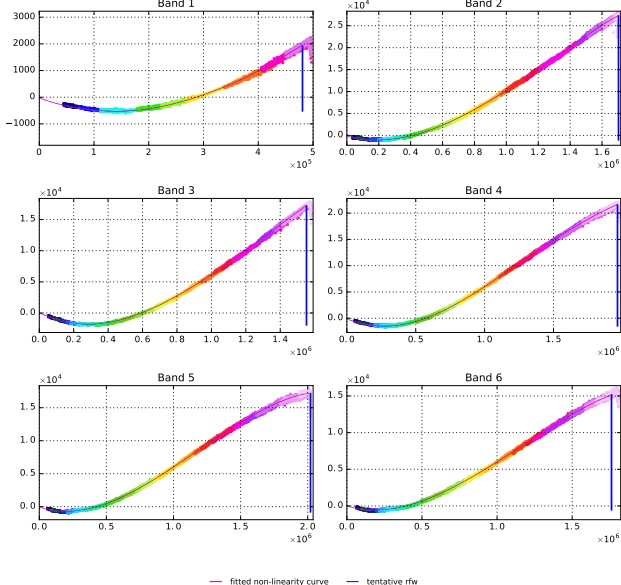

**Figure 7.** The obtained non-linearity curve, for all UVN bands (purple line) together with the scatter plot of the individual non-linearity tuples (coloured dots). The horizontal axis represents the measured charge (in electrons), the vertical axis the obtained non-linearity (measured charge minus expected charge). Each colour indicates a sub-cloud from the pixels of an individual measurement with certain exposure time. The blue vertical line indicates the limit set by the user of the valid region for the CKD.

The pixel full well is visible as an immediate flattening of the graph of pixel charge versus exposure time: pixel saturation. However, even detector pixels that receive a lower signal flux can be affected by saturation because the superfluous electrons spread over the detector and cause so-called blooming.

The analysis yields for each pixel that shows direct saturation a minimum saturation level $S_{m+1}$. The result per band is shown in Table 4, where over the set of all relevant pixels the minimum and some percentile values are shown. The values in this table can be used to set a threshold value for pixel saturation flagging.

The definition of pixel full well is ambiguous: not the maximum signal value a pixel can contain is important, but the maximum value a signal can attain while still being linear enough. "Enough" means here that the non-linearity correction is able to transfer the signal to be truly linear. Of course, a pixel can still bloom while its signal is in the linear range. Blooming is per definition not caused by the pixel signal itself. If neighbours are not flagged for pixel saturation, then we may assume that the pixel signal itself is "original": its signal value has not been increased by superfluous electrons from saturated neighbours. Conversely, unflagged pixels cannot be used safely unless it is clear that adjacent pixels are not flagged for saturation.





**Table 4.** The UVN saturation thresholds, measured in electrons, as obtained from unbinned measurements. Shown here are the minimum and three percentiles of the set of pixels that become directly saturated. Signal values above these thresholds should be flagged as "saturated" by the L01b processor, signal values below these thresholds can safely regarded as behaving linearly, unless saturation occurs in neighbouring pixels.

|  | Band 1 | Band 2 | Band 3 | Band 4 | Band 5 | Band 6 |
|---|---|---|---|---|---|---|
| **Minimum saturation [ke$^-$]** | 631 | 758 | 807 | 792 | 817 | 845 |
| **0.1 percentile [ke$^-$]** | 638 | 758 | 815 | 799 | 821 | 849 |
| **1 percentile [ke$^-$]** | 650 | 758 | 819 | 801 | 825 | 852 |
| **Median saturation [ke$^-$]** | 716 | 758 | 836 | 818 | 841 | 864 |

## 4.6 Detector pixel quality calibration

The L01b algorithm labels individual defective image pixels by raising a flag in the corresponding L01b product map if the quality factor of a pixel is below a certain threshold. This quality factor is calculated by the analysis algorithm and distinguishes between bad and dead pixels. The L01b processor only labels pixels as bad meaning either bad or dead.

The detector pixel quality flagging (DPQF) key data is a per pixel map of floating point values (quality factors) between 0.0 and 1.0, which correspond to the lowest and highest quality. The floating point threshold used in the L01b processor is currently set to 0.8. This means that the L01b processor will flag every pixel as bad when its quality factor value in the CKD is lower then 0.8. No key data can be calculated for the read-out register (ROR) pixels. These pixels are assigned a quality factor of 1.0. A UVN pixel is flagged as bad (dead) if it has either less than 90 % (10 %) quantum efficiency (QE) compared to its

neighbours, or shows darkness noise that is 10 (50) times higher than the CCD average, or creates dark current which is 10 (50) times higher than the CCD average. Pixels which display no large positive correlation between exposure time and signal response are declared dead.

To determine the dark current and darkness noise for the UVN detectors, unbinned dark measurements for three different exposure times (51 s, 255 s and 510 s) are used. To determine the quantum efficiency and response to light flux, unbinned

detector images illuminated with a flat light source (DLED) at nominal operational temperature and three different exposure times (0.36 s, 0.54 s and 1.0 s) are used.

The dark current check did not label any dead pixels, but it marked a number of bad pixels. Besides the original specifications, pixels are also marked as bad if the dark current turns out to be small or negative. Currently this low threshold is set to $-1 \, \mathrm{DNs}^{-1}$.

The second check is based on the darkness noise. The temporal standard deviation per pixel of the un-averaged measurements is used as a proxy for darkness noise.

The response to light flux is checked by calculating the correlation between pixel signal and exposure time, for all pixels. Since the correlation is higher than 0.9 for all pixels, no pixels are flagged with respect to this criterion.



**Table 5.** Number of bad and dead pixels per band, due to increased dark current (DC), increased darkness noise (not included in final map), decreased quantum efficiency (QE) or missing response.

| | DC | | Noise | | QE | | Response | Final map | |
|---|---|---|---|---|---|---|---|---|---|
| | **bad** | **dead** | **bad** | **dead** | **bad** | **dead** | **dead** | **bad** | **dead** |
| **Band 1** | 0 | 0 | 170 | 4 | 5 | 0 | 0 | 5 | 0 |
| **Band 2** | 0 | 0 | 52 | 0 | 3 | 0 | 0 | 3 | 0 |
| **Band 3** | 0 | 0 | 87 | 4 | 23 | 0 | 0 | 23 | 0 |
| **Band 4** | 1 | 0 | 58 | 0 | 5 | 0 | 0 | 6 | 0 |
| **Band 5** | 6 | 0 | 54 | 0 | 25 | 0 | 0 | 31 | 0 |
| **Band 6** | 6 | 0 | 123 | 1 | 24 | 0 | 0 | 30 | 0 |

Signal strength serves as a proxy for quantum efficiency. For each pixel (longest exposure time) the median of neighbouring pixels is computed. Currently, this is a square of 21 by 21 pixels. If the pixel signal strength is less than 90 % or 10 % of the computed median, the pixel is considered bad or dead, respectively. This resulted in no dead and some bad pixels based on the quantum efficiency.

Finally, the marked pixels based on dark and illuminated measurements are merged together to form the final DPQF map. Table 5 shows the number of bad and dead pixels that were marked by the OCAL algorithm. It is clear that most pixels have been flagged based on a deviating darkness noise. It was validated that all pixels, which were flagged based on darkness noise, were false positives, by using a prototype sigma clipping implementation. For now, the darkness noise criterion is switched off, so the total column and final map do not contain these flagged pixels. The number of flagged pixels is less than 0.04 % with

the current thresholds and with the darkness noise criterion.

### 4.7 Detector pixel response non-uniformity

The total radiometric response of the instrument consists of several components including both electronic and optical effects. Sometimes it is not possible to attribute a component solely to one of the two. This is especially true for the pixel response non-uniformity (PRNU) and relative radiometric response (RELRAD). The RELRAD can only be calibrated using an external

source, the PRNU must therefore also be obtained from the same measurement.

The PRNU relates a pixels individual response to that of its direct neighbours, whereas the RELRAD relates the radiometric response for every pixel to the pixel that is illuminated by light of the same wavelength on the detector row that corresponds to illumination from a source on the instrument's optical axis .

This section only reports on the results from the PRNU calibration. The concept of the calibration method and the details

behind the separation into PRNU and RELRAD are discussed in the section on relative radiance response 6.4.

The pixel response non-uniformity in the detector is corrected in the L01b data processor by multiplying by a pixel dependent map of values. Each detector has its own specific pattern, stemming from the way the detector was produced (coating,





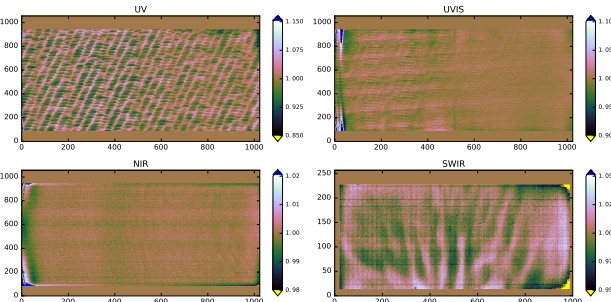

**Figure 8.** The PRNU for each detector as a function of the detector pixels. The horizontal axis is along the detector columns (spectral direction), the vertical axis along the detector rows (spatial direction). Measurements with the integrating sphere were used to derive the PRNU.

smoothing etc) as well as the specific response of the detector surface per wavelength. The PRNU is defined as a correction of a small deviation from the mean, that can vary from pixel to pixel. Therefore, the average value of this correction should be close to unity, and display a variation of the order of a few percent with a potentially very high pixel grid frequency.

The calibration measurement is performed using a 12" integrating sphere made from spectralon. It is equipped with quartz
tungsten halogen (QTH) light bulbs. The field of the stimulus is smaller than the azimuth range of the Earth port, and therefore all azimuth angles are measured separately by rotating the instrument with the cradle. These measurements are stitched together to yield a fully illuminated detector image. This image is separated into a high frequency PRNU component and a low frequency RELRAD component using a 2-dimensional Chebyshev fit.

Figure 8 shows how the pattern depends on the wavelength, and becomes smoother for larger wavelength going from UV to
UVIS and NIR respectively. The SWIR detector has a very different hardware architecture. This leads amongst other effects to an articulated even–odd effect in the rows of the SWIR channel. The pixel response non-uniformity is known better than $0.6\,\%$ after several validation tests.

## 5   Electronic calibration

Electronic calibration covers the offset and gain introduced in the the read-out chain of the front-end electronics. Here we treat
the UVN electronic calibration, the SWIR calibration is described in Hoogeveen et al. (2013). Table 6 gives a summary of the UVN calibrated electronic and detector properties which are part of the CKD in the L01b data processor.

### 5.1   Electronic offset

The definition of the electronic offset is the measured output signal when the input signal is zero and in absence of dark current. Each TROPOMI CCD detector has two distinct electronic detection chains, one for each band. Each chain itself has
two separate "lanes" that alternatively sample the sequence of pixels as they are read-out, causing an artificial odd-even effect





**Table 6.** The main characteristics of the UVN detectors and front end electronics. Unless stated otherwise, the values are given for offset = 0 (DAC), even parity, PGA = 1 (code 97), CCD gain low, CDS gain low. The dark current is based on the maximum column average value. The register full well value is based on the maximum value where the non-linearity correction is valid.

| | UV | | UVIS | | NIR | |
| Property | Band 1 | Band 2 | Band 3 | Band 4 | Band 5 | Band 6 |
|---|---|---|---|---|---|---|
| Digital number to voltage factor [mV DN$^{-1}$] | 0.122[a] | | 0.122[a] | | 0.122[a] | |
| Offset [mV] | 47.86[b] | 17.75[b] | 13.42[b] | 8.96[b] | 10.51[b] | 7.97[b] |
| CCD low gain | 1 | 1 | 1 | 1 | 1 | 1 |
| CCD high/low gain ratio | 1.79 | 1.78 | 1.82 | 1.83 | 1.81 | 1.82 |
| CDS low gain | 1 | 1 | 1 | 1 | 1 | 1 |
| CDS high/low gain ratio | 1.98 | 1.98 | 1.98 | 1.97 | 1.98 | 1.98 |
| AVC gain | 5[a] | 1.5[a] | 1.6[a] | 1.25[a] | 1.25[a] | 1.4[a] |
| Voltage to charge factor [Me$^{-}$V$^{-1}$] | 1.266[a] | 1.282[a] | 1.266[a] | 1.235[a] | 1.266[a] | 1.266[a] |
| Read-noise low CCD gain [e$^{-}$] | 123 | 187 | 193 | 245 | 197 | 155 |
| Read-noise high CCD gain [e$^{-}$] | 82 | 108 | 104 | 128 | 116 | 95 |
| RFW [Me$^{-}$] | 0.48 | 1.70 | 1.55 | 1.96 | 2.02 | 1.76 |
| PFW [ke$^{-}$] | 631 | 758 | 807 | 792 | 817 | 845 |
| Dark current [e$^{-}$s$^{-1}$] | ≤1.7 | ≤1.6 | ≤1.7 | ≤1.6 | ≤1.9 | ≤1.9 |

[a] Design value

[b] Shown is the average. The correction is per pixel.

in the offset. The measured offset does however depend on the instrument settings for gain and offset, albeit not in an exactly linear way. The offset is thus defined as a single quantity per band and per parity (even or odd pixel). If a single pixel shows a deviation in offset from this general offset, this is called the offset residual. The residual is an image per band that differs per instrument configuration (ICID).

5     There are eight ways to determine a value for the electronic offset, to treat them all is out of scope for this paper, but the important conclusion is that these methods do not yield the same answer within their precision. During the on-ground calibration it was observed that the offsets were not static as expected, but show drifts in time over several minutes. In addition, the offset shows patterns during a frame read-out which causes spatial variations referred to as pixel offset non-uniformity (PONU).

10     To reduce the impact on the L1b product, the electronic offset correction is implemented in the L01b data processor as a dynamic correction, the calibration key data is retrieved per measurement frame. The preferred order is to first apply a dynamic correction using overscan rows. The CCD gain should be set to the same value as the science data area. Secondly a dynamic correction using the read-out register (ROR) can be used. The CCD gain of the ROR should be set to the same value as the





science data area. The least preferred option is to use the static offset based on dark images. The temporal drift in offset will introduce an additive error component.

Spatial offset residuals in a frame (PONU) are not modelled in the offset correction, but are corrected for by the L01b residual correction algorithm. This correction step corrects many unwanted features like spatial spikes and asynchronous read-

out patterns. The offset residual of a bright image is acquired from its dual dark image (which itself has also been dynamically corrected for offset drift). The residuals themselves show some temporal drifts that cannot be corrected for; but that error is much smaller than the imposed error of not performing the residual correction.

## 5.2   Electronic amplification

The front-end electronics of the UVN detector modules consists of six separate electronic chains, one for each band. All pixels

in a band are read out sequentially through one of those chains. In each of these electronic chains several component gains must be calibrated. These individual gains must be multiplied to yield to the so-called system gain. This system gain (and offset) is thus equal for all pixels in a band by design, and therefore no pixel-dependent behaviour is expected. Some of these component gains can be actively controlled by changing the instrument settings. These gain settings are chosen to optimize signal-to-noise ratios while avoiding saturation. The CKD to be determined here is the *actual* system gain as function of individual settings

for gains. As shown in Fig. 2, the components between the detector and the ADC that yield the electronic gain are: the charge-coupled device (CCD) output node, with 2 possible settings (switchable per image row); the analog video chain amplifier (AVC) with a fixed gain; the correlated double sampler (CDS), with 2 possible settings; and the programmable gain amplifier (PGA), with 256 possible settings.

The goal of the calibration is to determine gain *ratios*, i.e. ratios of the gain resulting from a certain gain setting and the gain

resulting from the so-called reference setting. The total instrument gain itself or absolute radiometric calibration is addressed in a separate process which relates a known incoming photon flux to the instrument response at these reference gain setting.

The gain ratios must thus be determined for all potential combinations of the possible gain settings: $2 \times 2 \times 256$. In addition, the gain ratio can differ between odd and even columns. This is due to the fact that the internal design of the CDS has two separate electronic chains to increase the sampling speed. These two lanes are not entirely identical, leading to a slightly

different offset and gain between them. Because these lanes are used alternately for each consecutive sample, a sequential readout of the detector will result in a artificial column-wise odd–even effect. The instrument can be configured such that it is enforced which of the two lanes is used for the first column, this is called parity. In addition, the engineering data contains detailed information regarding which lane was used for each pixel. Nonetheless, the gain ratio calibration must also address this feature which effectively adds a fourth dimension to the calibration key data. The reference setting is defined as PGA gain

setting 97, low CDS gain, low CCD gain, and even parity.

The 4-dimensional calibration key data is derived from unbinned DLED measurements at various exposure times, and multiple combinations of prevalent gain settings. First, dark measurements are subtracted from the corresponding bright measurements. All measurements are divided into unique groups according to the instrument gain settings. Within each group, the measurements have the same settings except for the exposure time. For each gain setting group, for each pixel, the flux (being




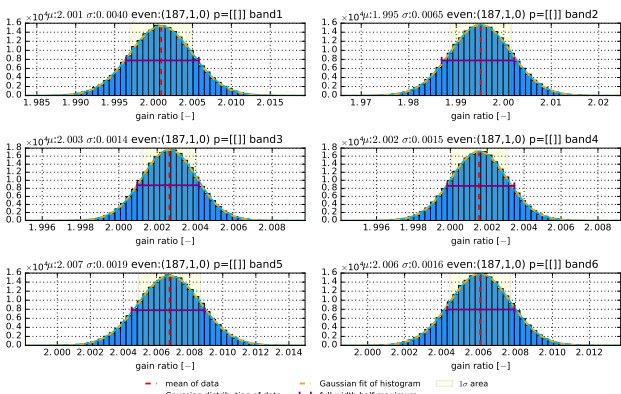

**Figure 9.** The histogram of the obtained pixel gain ratios for pixels with even parity for all UVN bands.

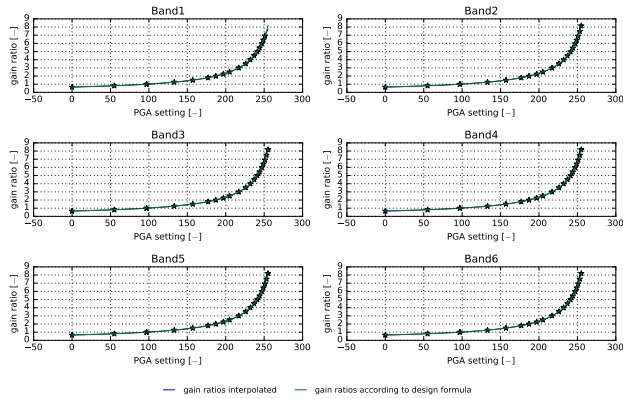

**Figure 10.** The obtained UVN gain ratio as a function of the PGA setting for even pixels and low CCD and CDS gain. Shown are the measured settings as stars, the interpolated gain ratio (blue curve) and the design formula by the supplier (green curve).

the slope of the graph of signal magnitude against exposure time) is computed using a linear fit. The flux is then divided by the flux at the reference gain setting. Finally, the gain ratio is defined as the average of the individual pixel gain ratios. In Fig. 9 the histograms of the derived gain ratio per detector pixel in each band are given. In this example the gain setting is chosen such that it would result in a gain ratio of about 2. Although there is a considerable spread in individual gain ratios, the mean of

5    the histogram is rather accurate because of the large number of pixels involved. In Fig. 10 the obtained gain ratios are shown for all calibrated PGA settings indicated with dots. The interpolated curve for all possible 256 PGA settings are also given, as well as the gain ratio according to the design formula from the supplier. Validation shows that the determined gain ratios are reasonably stable over shorter time periods and depend on the analysis method. Long term drifts exist that are relatively small for PGA and CDS gain settings (0.1 %) but high for the high CCD gain setting (1 %). The choice of measurements to

10   compute the fluxes also matters. In general, a large range of exposure times is better. Variations of up to 0.5 % exist. Likewise,



the binning factor matters: differences of 0.4 % are seen. The differences when varying binning factors and exposure times relate to signal magnitudes; they determine the residual patterns. These spatial patterns are non-deterministic and therefore no local corrections ("pixel gain ratios") are possible.

We have to conclude that the gain ratio, as a mean of pixel gain ratios, is not a stable entity; instead, the gain residuals seem to define the mean. We cannot maintain the original definition of the error as the standard deviation of the mean: the gain ratio distribution is not Gaussian, not uncorrelated, and the residuals are persistent. Therefore we take the pragmatic measure of determining the error as the standard deviation of the distribution itself.

# 6   Radiometric calibration

## 6.1   Slit irregularity

Each TROPOMI spectrometer images its slit on its detector, dispersing the spectral direction with a grating. For UVIS and NIR only the slit in the telescope is of consequence, while UV and SWIR each have an additional slit in their optical path.

The slit width determines – together with the grating and detector properties – the spectral resolution. The slit also determines the field of view and limits the optical throughput.

The slit length dimension maps the across-track spatial information onto the detector's row dimension. The dimensions of
the main slit are given in Table 2. It is desirable to have the slit width equal for all across-track positions to ensure an even optical throughput and also an even spectral sampling for all across-track positions. This means the slit must be manufactured such that the width is constant along the length of the slit. Deviations from a perfect slit are referred to as the slit irregularity. This slit irregularity becomes part of the calibration of the instrument. The throughput effect is covered by the radiometric calibration, whereas the spectral resolution effect is covered by the ISRF calibration.

Based on mechanical design considerations the slit irregularity of the TROPOMI instrument is expected to be very small: performance measurements of the slit block bread-board indicated the slit irregularity of the common slit to be smaller than 0.1 %. Nonetheless the slit irregularity could in principle be calibrated using dedicated measurements. To ensure consistency with other radiometric calibrations the approach is to simultaneously derive the relative radiometric response, PRNU and slit irregularity from a single measurement.

The algorithmic concept is such that an arbitrary cross-talk exists between these three CKDs. As a result the derived slit irregularity differs between various measurements with different illumination sources. In addition, the magnitude of the derived correction is much larger than what may be expected from the mechanical design of the hardware. In Fig. 11 the derived slit irregularity is shown obtained from internal WLS measurements, and the three external sources integrating sphere, external white light source (EWLS) and QTH2. The integrating sphere is the same as described in Section 4.7. The external white light
source (EWLS), in this case the Sun simulator, is a collimated and homogenized 1000 W Xe lamp illuminating the instrument via a calibrated diffuser. The same diffuser was used for the absolute radiance calibration (see Sec. 6.5). The source QTH2 is a uncalibrated 1000 W QTH FEL lamps with a colour temperature of about 3200 K. It was used together with a calibrated




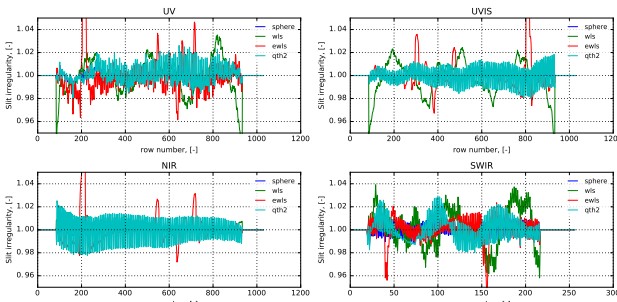

**Figure 11.** For each detector the slit irregularity as a function of the row number is shown. The fractional correction amounts to roughly 3 % peak-to-peak, which is larger than expected from mechanical design. Large differences between measurements with different stimuli (sphere, WLS, EWLS and QTH2) can also be seen. The output of the EWLS source (red) was not stable over the measurement, which results in the features in the row direction.

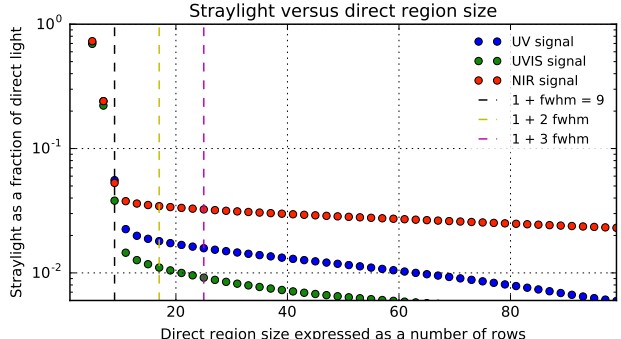

**Figure 12.** Shown is the integrated straylight fraction received per UVN detector as a function of the direct region size. The illumination is via the Earth port . The NIR detector (red dots) displays the most straylight followed by the UV (blue dots) and the UVIS (green dots). The vertical dashed lines indicate the direct illumination region sizes of 1 detector row (centre of mass of the column average of the direct signal) + 1× (black dashes), 2× (yellow dashes), and 3× (red dashes) the direct signal FWHM.

diffuser. All results differ significantly, and the values obtained for the slit irregularity are larger than is expected from the mechanical construction.

This lack of accuracy leads to the conclusion that the slit irregularity of TROPOMI can not be derived confidently. The chosen approach is to define the slit irregularity to be non-existent by setting the corresponding CKD to unity. The calibration algorithm is tuned to attribute the low-frequency patterns to the RELRAD, and *all* remaining high-frequency components to the PRNU. Following this method a consistent radiometric calibration is ensured.



**Table 7.** Integrated straylight for the UVN detectors calculated from the measurements with the EWLS via the Earth port. Shown is data for three different sizes of the direct region (DR). Straylight is expressed as a percentage of the direct light.

| DR = | UV | UVIS | NIR |
|---|---|---|---|
| **(1 + FWHM)** | 5.6 | 3.8 | 5.3 |
| **(1 + 2 × FWHM)** | 1.8 | 1.1 | 3.4 |
| **( 1 + 3 × FWHM)** | 1.6 | 0.9 | 3.2 |

## 6.2  In-band straylight calibration

Straylight is any light that falls on a detector pixel which by optical design is not intended to detect that light. As opposed to intended or direct light, straylight is also referred to as unintended or indirect light. Scattering from surface roughness, unwanted specular reflections (ghosts), and unwanted diffraction effects (e.g. diffraction at obstacles, Rowland ghosts, and unwanted grating orders) cause the light to follow different paths from those intended by optical design. The redirected light that reaches the detector, i.e. straylight, gives rise to spurious signals at the detector. Both to characterize and to correct for TROPOMI straylight, a set of straylight tests were designed. The straylight from outside the field-of-view of a spectrometer (out-of-field straylight) is determined with a external white light source (EWLS). In this case a 300 W Xe source with a field of $1.05° × 2.03°$ (across × along flight) is used. The light is homogenized by a small integrating sphere. This source is also used to construct straylight validation scenes and to determine the spatial part of the in-field straylight. The same xenon source, but then in combination with optical bandpass filters with a FWHM of about 10 nm is used to determine the far-field straylight within the spectral range of each UVN spectrometer. For the out-of-spectral-range straylight, long- and shortpass filters were used. The large field of view of the white light and bandpass measurements is complemented by measurements for the near-field with a pulsed laser. The laser light is homogenized by a small integrating sphere and has a field of $0.0131° × 2.061°$ (across × along flight). The output of the laser can be tuned in the range 210–2600 nm with stepsizes between 0.05 nm for the short wavelengths and 1 nm for the long wavelengths.

The in-field, in-spectral-range straylight in the SWIR detector was characterized by SRON, and is reported in Tol et al. (2017).

The UVN in-field, in-spectral-range straylight is corrected for in the L01b data processor using a convolution algorithm which first calculates a straylight signal for a given input signal, and then subtracts the calculated straylight signal from the input signal. The main ingredient of the L01b straylight algorithm is the convolution between an input signal and a straylight convolution kernel to obtain a straylight signal. In this sense, this algorithm is similar to convolution algorithms used in common commercial image processing to sharpen or to smooth images. There, images are smoothed or sharpened depending on which convolution kernel is used. In the straylight correction, a special straylight convolution kernel, called a straylight response function (SLRF), is used to calculate the straylight signal for a given input signal. The SLRF describes the relative straylight response of the system and is derived from on-ground calibration measurements.





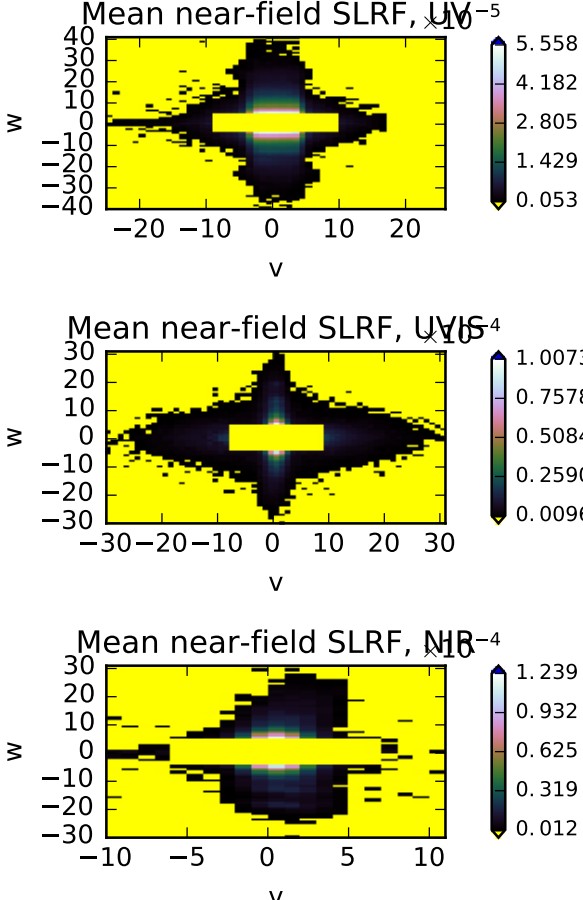

**Figure 13.** Mean near-field straylight response function, shown for the UV (top), UVIS (middle), and NIR (bottom) spectrometer. Each response function is displayed as 2-dimensional pseudo-colour plot, as a function of indices $w$ (vertical axis) and $v$ (horizontal axis). The unit for a straylight response function is 1. The rectangle in the centre is the excluded direct region.

To make a distinction between straylight near the directly illuminated region (direct region, DR), and straylight far away from the direct region, we call the former near-field straylight, and the latter far-field straylight. It is clear that the straylight calculated as a fraction of the direct signal depends on the choice of the size of the direct region. The direct region should of course be larger than the FWHM of the direct signal. Otherwise, the reported straylight level would be exaggerated. Figure 12 shows how the calculated relative straylight depends on the choice of the direct region size, which is expressed as a number of detector rows. Three vertical lines in Fig. 12 denote direct region sizes of 1 detector row (centre of mass of the column average of the direct signal) + $1\times$, $2\times$, and $3\times$ the direct signal FWHM. Table 7 summarizes straylight values calculated for the three direct region sizes.





**Table 8.** Average straylight before and after the straylight correction together with the corresponding error estimates, as well as the average straylight correction factor, as obtained from the EWLS measurement data via the Earth port. The error estimates are valid only if the systematic errors are negligible with respect to the random noise. Results are shown for the UVN detectors.

|  | Straylight before [%] | Error before [%] | Straylight after [%] | Error after [%] | Correction factor |
|---|---|---|---|---|---|
| **UV** | 2.052 | 0.004 | 0.811 | 0.004 | 2.5 |
| **UVIS** | 1.23 | 0.003 | 0.527 | 0.003 | 2.3 |
| **NIR** | 4.041 | 0.004 | 3.314 | 0.004 | 1.2 |

A mean near-field straylight response function was derived for each spectrometer of the UVN module. The obtained mean near-field straylight response functions are shown in Fig. 13. The indices $w$ (vertical axis) and $v$ (horizontal axis) indicate the distance in pixels from the direct signal.

The rectangle in the centre is the excluded direct region. The algorithm used direct region sizes of $9 \times 17$, $9 \times 17$, and $9 \times 13$
(spatial $\times$ spectral) detector pixels, for the UV, UVIS, and NIR spectrometer, respectively. The direct region sizes were chosen in such a manner to ensure that the straylight response does not include what is included in the instrument response. The direct region size in the spatial direction of nine detector pixels was used to achieve consistency with the PRF analysis, which uses seven detector pixels in the spatial direction. Similarly, the direct region sizes in the spectral direction were chosen to achieve consistency with the non-zero ISRF spectral ranges of [-0.5, 0.5], [-1.2, 1.2], and [-0.6, 06] nm, for the UV, UVIS, and NIR
spectrometer, respectively. After the near-field straylight response functions had been calculated, the far-field components were added to obtain the mean extended straylight response functions.

For the case of the EWLS measurements via the Earth port as a function of azimuth angle, the convolution algorithm reduces straylight outside the direct region on average by a factor of 2.5 (from 2.052 to 0.811 % of the direct signal), 2.3 (from 1.230 to 0.527 % of the direct signal), and 1.2 (from 4.041 to 3.314 % of the direct signal) for the UV, UVIS, and NIR spectrometer,
respectively. Worse straylight performance in the case of the NIR spectrometer is due to a very strong out-of-spectral-range straylight contribution caused by light at wavelengths longer than 795 nm, which accounts for about 75 % of the observed straylight. This contribution is expected to be smaller in-flight than it is in the on-ground calibration measurements. Unlike the Sun, the xenon lamp used as the source in the EWLS measurements, has strong emission lines in the wavelength range between 800 and 1000 nm, which are the reason for the observed dominant out-of-spectral-range straylight contribution.
To validate and verify the straylight performance at Level 1b, a special hole-in-the-cloud scene is defined. It describes a worst-case radiance observation in which the full swath of the instrument is illuminated by pure white clouds, apart from a small gap around nadir. For on-ground validation this scene is constructed from EWLS measurements and the L0 and L1b requirements are expressed in terms of the signals from the EWLS. The spectrum of the source differs from the in-flight radiance, so the scene represents a EWLS hole-in-the-cloud. The convolution algorithm reduces straylight in the middle of
the hole of the EWLS hole-in-the-cloud scene as given in Fig. 14. The correction factor exceeds 10 for some wavelengths in the case of the UV spectrometer, and it reaches a value of 5 and 1.4 for the UVIS and NIR spectrometer, respectively.




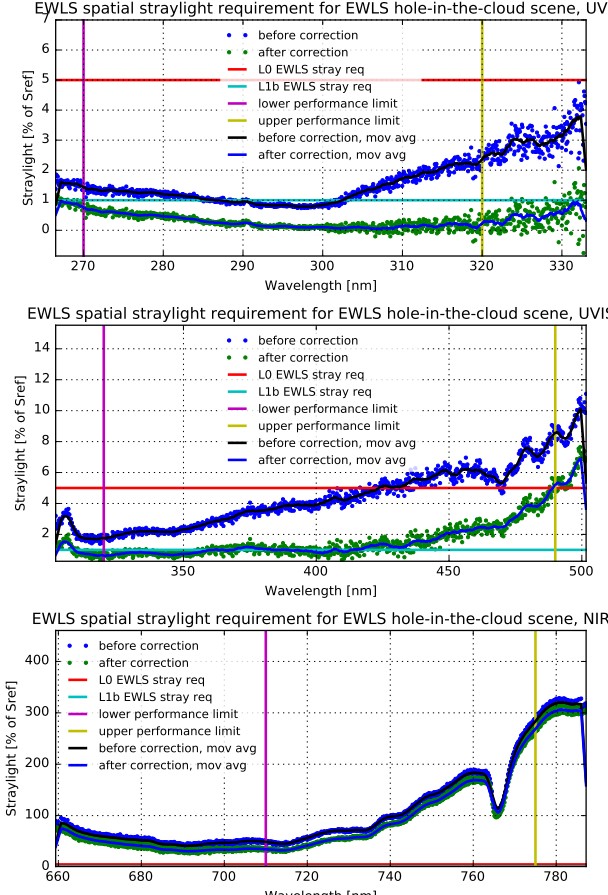

**Figure 14.** Straylight in the middle of the hole of the EWLS hole-in-the-cloud scene, before (blue dots) and after (green dots) the straylight correction with the straylight convolution algorithm. Solid black and blue line respectively denote moving average of the straylight before and after the correction. Data is shown for the UV (top panel), UVIS (middle panel), and NIR (bottom panel) spectrometer. In each panel, the two vertical lines denote the limits of the spectral performance range. The horizontal red line denotes the EWLS spatial straylight requirement at L0, and the horizontal cyan line denotes EWLS spatial straylight requirement at L1b.

The UV spectrometer is compliant with both the L0 and L1b EWLS spatial straylight requirement, the UVIS spectrometer is partially compliant with both EWLS spatial straylight requirements, and the NIR spectrometer is not compliant with either of the EWLS spatial straylight requirements. In order to achieve full compliance of the UVIS and NIR spectrometer with the EWLS L1b spatial straylight requirement, it is necessary to correct for the out-of-spectral-range straylight. This correction has been determined for the NIR spectrometer as described in the next section.



## 6.3 Out-of-spectral-band straylight

During the calibration effort unexpected signal was encountered in the NIR spectrometer, which was tentatively attributed to out-of-spectral-band straylight. At that time no means were available for a full calibration and the topic was postponed. At a later point in the program, after integration of the instrument on the satellite, the opportunity arose to perform additional

measurements. A dedicated setup was designed that could characterize the straylight at ambient temperature and pressure using the instrument already integrated on the platform.

Due to the nature of the out-of-band (OOB) straylight, no direct correction is possible in-flight. The out-of-band signal is not measured in-flight and therefore a correction must be based on an assumed out-of-band spectrum. Depending on the illumination, via the Sun port, the Earth port or from internal sources, the spectra can differ. To weigh the out-of-band spec-

trum according to the instrument's response the source-wavelengths and -directions of the out-of-band straylight need to be characterized.

The out-of-band straylight campaign for the NIR detector at Airbus UK in Stevenage in December 2016 and January 2017 consisted of relative measurements performed in ambient. The NIR detector is stabilized to 297 K. The instrument response is measured first for in-band wavelengths and then for out-of-band wavelengths. As a source two narrow linewidth, tunable

continuous wave (CW) lasers are used. The two lasers cover different wavelength ranges: 710–945 nm for the "red" laser and 592–683 nm for the "blue" laser.

The power of the laser is measured with a photodiode for both in- and out-of-band wavelengths. When spectral dependencies of the source and setup are included, the in-band signal strength and swath position can be related to an expected out-of-band signal. To avoid long integration times, the out-of-band wavelengths is measured with a higher laser input power than the

in-band wavelengths. The necessary attenuation is achieved with neutral density (ND) filters.

The setup to measure the NIR out-of-band straylight in ambient consists of the so-called Earth port adapter (EPA) which is attached to the Earth port of TROPOMI, an optical assembly with neutral density (ND) filters and light sources. The Earth port adapter (EPA) is connected directly to the Earth port. Attached to the EPA are 11 fibre collimators. They are aligned to illuminate different swath angles (rows) of the NIR detector. The output of the light sources is coupled via a small integrating

sphere and optical fibres to the collimators.

During the campaign, the short wavelength side of the NIR detector was referred to as the "blue" side, and the long wavelength side as the "red" side. For both red and blue sides, the raw data consists of 11 sets of NIR-detector images, one set for each separate fibre. Each fibre is spatially connected to a part of the detector: fibre number 6 illuminates the nadir rows. The first fibre illuminates the upper rows of the detector, while fibre number 11 illuminates the rows closest to the bottom.

A sequence of $N$ laser measurements $S_{\mathrm{oob}}[f,k], \quad k = 1, \cdots N$ is obtained, where for each $k$ the laser has been tuned to an out-of-band wavelength of $\lambda_k$. Each measurement is a NIR-detector image. For a range of $k$, it will show an elliptical "blob" attached on the right-hand detector boundary, around the rows associated with the fibre, and stretching towards the lower wavelengths. It is possible that the blob is present in the entire detector. For some $k$ just outside the in-band wavelengths, sharp ghosts may be visible, possibly together with a uniform (flat) signal ("film") of low magnitude. Finally, for $k$ with wavelengths





moving farther away from the detector, the OOB straylight will disappear. The ND filter is chosen such that enough signal is present in the detector. The attenuated laser signal is still too high to be used for in-band wavelength; pixel saturation would immediately occur.

A small sequence of $M$ in-band laser measurements $S_{\mathrm{ib}}[f, m], \quad m = 1, \cdots M$ are also obtained. The ND filter is chosen such that the in-band wavelengths will not cause pixel saturation. This gives a signal peak of a few pixels in spectral direction (amounting to ca. $0.2\,\mathrm{nm}$) and ca. 15 pixels in spatial direction. Further, some near-field straylight (ghosts) are present. The exact shape is not important; what matters is the total signal in the direct region (DR). This is needed to normalize the OOB laser measurements. Note that the laser strength depends on the wavelength, this is measured separately. The differences in the DR signal for the different internal wavelengths will be consistent with this wavelength dependency (after correction of the signal for absolute and relative radiance).

During the the laser measurements the out-of-band power is monitored using a photodiode and current meter. The current is re-arranged per intended laser wavelength by matching the timestamps attached to the laser and the current meter. The current, after proper offset correction and calibration, is directly connected to the laser output *including* ND filter transmission. In other words, the evolution of the amount of photons (caused by laser drift, ND filter drift and wavelength dependency of both) reaching the instrument is completely correlated with the photodiode current. Here we assume that the current read-out has a negligible error and that both fibres and collimators show no relevant spectral dependence.

Subtraction of the generic dark current image should have removed all artefacts and gradients. However, some residual gradients tend to remain. Therefore the very last image of each laser measurement series is used as a final background correction, the assumption being, of course, that this last image does not contain any straylight. For the red side, this means that the $840\,\mathrm{nm}$ image serves as background. For the blue side, the $615\,\mathrm{nm}$ image is used. This approach is repeated for each fibre, such that the maximum time difference between the background image and the relevant images is less than three hours.

The typical behaviour of the spatial straylight distribution is shown in Fig. 15 for fibre 4. The straylight "blob" is most prominent at the associated detector row (i.e. row 668) of this fibre, and is somewhat skewed towards the nadir row. In general, as could be expected from the total straylight magnitude, the blob grows when the laser illuminates the wavelengths from $808\,\mathrm{nm}$ to $818\,\mathrm{nm}$, and then decreases in size and strength. From $830\,\mathrm{nm}$ onward, no more out-of-band straylight is visible.

Another general feature is the appearance of a ghost when the laser is at a wavelength just outside the detector range. This ghost is spatially and spectrally small, and moves considerably as a function of the selected laser wavelength. Modelling such behaviour is a difficult task. Because the ghost is spatially located at the same rows as the associated in-band signal, it will probably not form a significant feature in the total detector image. Furthermore, the total energy of the ghost is low compared to the blob visible at wavelengths further from the detector range. For these three reasons, it has been decided to ignore such ghosts. Efforts will be concentrated on modelling the, spatially and spectrally, much more prominent blob.

Smoothing is done in two steps. Firstly, an image is downscaled to a resolution of $100 \times 100$ pixels. Each pixel signal is the local average of a subset of pixel signal where the extent of the subset is based on the local signal-to-noise ratio. That means that a region with rather low signals gives a local average from a lot of pixels, while a local peak will be averaged using only its immediate neighbours. This approach ensures that local maxima do not become smaller while the overall smoothness is





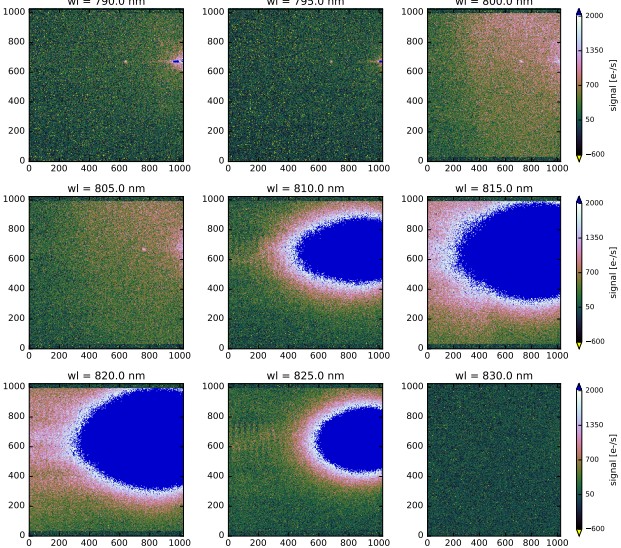

**Figure 15.** The straylight image of the NIR detector for a range of laser wavelengths (wl) on the red side for fibre 4 before normalization. Detector columns are in the horizontal direction, detector rows in the vertical direction. The colour scale is the same for all images.

ensured. In general, the resulting downscaled images tend to be monotonous, without wiggles caused by noise. Very small ghosts almost disappear in the background, as is intended. The second step is that the downscaled image is reverted back to full resolution using bilinear interpolation. In this method, the desired signal-to-noise ratio (SNR) can be varied. The trade-off is between pertaining noisy features that become a CKD error and over-smoothing, thus removing genuine features and lowering

peak values.

After smoothing and normalization, the absolute straylight fractions from both sides can be compared. In Fig. 16 it can be seen that the absolute straylight fraction on the blue side is smaller than on the red side. The assumption is that the measured in-band signals at 666 nm (blue side) and at 780 nm (red side) are equal and that the extrapolation toward the out-of-band wavelengths is constant. Under this assumption of an entirely flat spectral density, the integrated straylight signal from the

red side is about six times larger than the integrated signal from the blue side. More precise: the ratio of the *areas* defined by the red and blue lines is six. Note that the normalized fraction is difficult to interpret: it is the signal sum over all *pixels* at a given out-of-band laser wavelength relative to the associated total signal in the direct in-band region (and, in fact, in the entire detector) at a chosen in-band wavelength.

Validation measurements were performed at the end of the campaign, with one or more fibres connected to a 12" spectralon

integrating sphere with $3 \times 5$ W QTH lamps. This is the same stimulus as was used for the NIR spectrometer during the regular calibration campaign for the PRNU and RELRAD, see Sec. 4.7. The raw CKDs are the same everywhere, but the processed CKDs may differ for several cases, due to the choice of out-of-band spectral extrapolation. A direct extrapolation from the signal measured by TROPOMI (corrected for absolute radiance) is used, instead of information from the integrating sphere





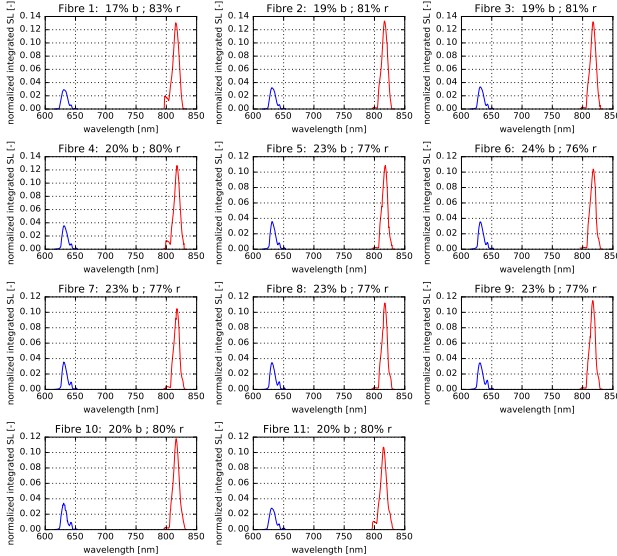

**Figure 16.** Overview of the combined contributions to straylight from both the red, long wavelength side (red line) and the blue, short wavelength side (blue line) sides. A constant (flat) spectrum is assumed. The different fibres illuminate different rows of the NIR detector.

datasheets. The out-of-band straylight correction is done in two steps: first for straylight from the red side, then for straylight from the blue side (Fig. 17). The latter correction is smaller in magnitude.

The overall straylight reduction is shown in the top panel in Fig. 18. Note that no in-band straylight correction has been applied here. Only in a later step, remaining straylight is partially removed by the regular in-band straylight L01b processor

algorithm. In the second panel of the same figure, the expected out-of-band spectral behaviour is shown. In this case, it is a simple linear extrapolation from the in-band signal, since the signal measured by TROPOMI itself was considered more reliable than external sources. In the bottom panel, the evolution of low signals in the image is shown: in general, after correction a Gaussian curve with a mean of zero or slightly positive values is to be expected. The reason is that, after all straylight removal, a pixel signal is either very high (in the rows directly illuminated by the fibre) or zero (all other rows that are not directly

illuminated).

### 6.3.1 Activities for the E1 phase

All potential methods to validate the out-of-band straylight correction in the in-flight phase have to be considered. For the irradiance, the in-flight solar measurements will be used to construct an optimal processed out-of-band correction CKD. For the radiance, construction of the processed CKD is more complex because input from L2 algorithms is needed to analyse the

residuals, it may be necessary to analyse and correct for separate scene characteristics, water vapour dependence might be included by adjusting the non-linear scaling configuration parameter, and the impact of undercorrection has to be analysed.



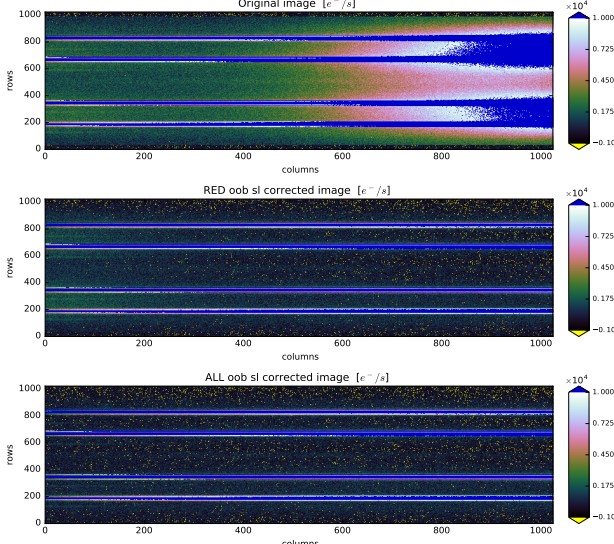

**Figure 17.** Validation measurement with fibres 2, 4, 8 and 10 illuminated by the integrating sphere. Top figure: NIR detector image before correction. Middle figure: image after correction for straylight from the red side only. Bottom figure: image after out-of-band straylight correction for both sides.

## 6.4 Relative radiance

The absolute radiance responsivity (ABSRAD) and the relative radiance responsivity (RELRAD) CKDs together provide the absolute radiance responsivity calibration for all viewing angles of the instrument Earth port. The RELRAD analysis consists of two separate instances of the same algorithm, that separates essentially three different constituents of the signal: a spectral

contribution, a spatially slowly varying part and a spatially fast varying part. First, the algorithm processes the measurements to produce the fast varying part, the PRNU, as described in Section 4.7. Then the straylight correction algorithm is applied. When the signal has been corrected for PRNU and straylight it is subjected to the same algorithm as used for the PRNU, but now to only produce the slowly varying part, the relative radiometric response.

    In order to perform the analysis, a complete detector image is needed to separate the high and low frequency residual

patterns from the signal. The measurement that is used for the derivation of the CKD uses a integrating sphere stimulus that only illuminates $\approx 33$ rows for the UVN and $\approx 8$ rows for the SWIR at a time. The stimulus is described in Section 4.7. This illumination pattern is scanned over the detector by varying the cradle rotation angle with $1°$ at a time (this corresponds to $\approx 8$ rows in UVN, 2 rows for SWIR). Between each two measurements a reference measurement at cradle rotation angle $\phi = 0°$ is performed.

The total scanning sequence over all swath angles takes a considerable amount of time, during which the output light source may vary. This wavelength-dependent lamp drift is calculated using the reference measurements and each measurement is corrected for this drift. Otherwise the drift would be falsely injected into the relative radiometric calibration key data. To arrive




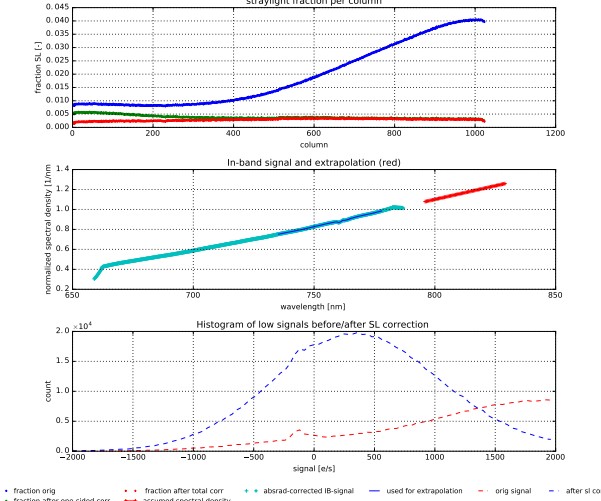

**Figure 18.** Top figure: the total straylight fraction per NIR detector column, before (blue line) and after out-of-band straylight correction for the red side (green line) and both sides (red line). Middle figure: the spectral extrapolation as based on the in-band signal (shown in turquoise). The in-band signal is corrected for absolute radiance and normalized to a characteristic in-band wavelength. The thin blue line denotes the region (roughly between 730 and 780 nm) on which the extrapolation is based. The extrapolation is linear in this case and shown as an orange line. Bottom figure: histogram of the low signals below $2000\,\mathrm{e^- s^{-1}}$, before (blue dashes) and after (red dashes) out-of-band straylight correction.

at a full detector image the partially overlapping measurements need to be "stitched" together. The subsequent steps that are taken in the algorithm are described here in more detail.

### 6.4.1 Correction of the stimulus drift

The reference measurements at cradle rotation angle $\phi = 0°$ must all have the same PRNU and absolute radiometric response, and therefore, any change in the observed signal must stem from the stimulus itself. By comparing all reference measurements over time to the initial, first measurement, the drift of the stimulus can be obtained. The measurements must be corrected for this wavelength dependent drift which otherwise would end up falsely in the relative radiometric response. The stimulus drift can be seen in Fig. 19. Where applicable, e.g. for the SWIR detector, a background image will be subtracted from the illuminated images. For the UV measurements with the integrating sphere, no background subtraction was performed, because it gave inconsistent results.

### 6.4.2 Stitching

Due to the limited field-of-view of the instrument in the thermal vacuum chamber (TVC), only a limited number of detector rows is illuminated at a time. These rows must be stitched together to obtain a full detector image required to derive the





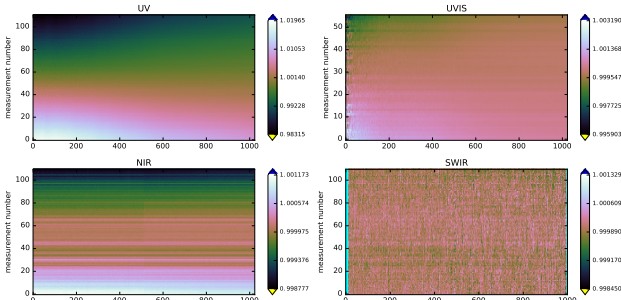

**Figure 19.** The lamp drift as a function of the column (wavelength, horizontal axis) and measurement number (vertical axis) for the UV, UVIS, NIR and SWIR detector. The intensity is normalized with respect to the output of the first measurement. The plots indicate a downward trend in the UV and UVIS output that is stronger for the shorter wavelengths. The NIR channel also shows a downward trend, but one that is nearly independent on the wavelength. The SWIR channel does not show any evidence for drift in the lamp output. The variability of the source output is most pronounced in the NIR channel. The systematic drift is largest for the UVIS channel.

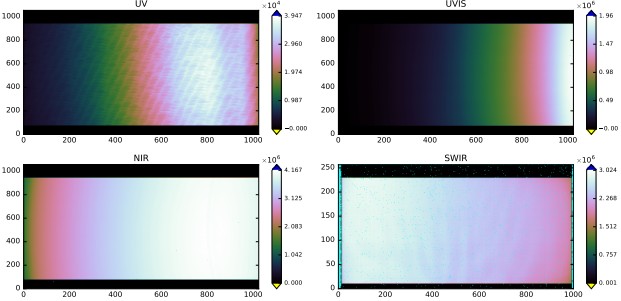

**Figure 20.** The individual swaths have been stitched together by considering the maximum over all measurements. In this way an illuminated detector image is constituted for all detectors (in $e^- s^{-1}$). On the horizontal axis are the detector columns, on the vertical axis the detector rows. The black areas on the bottom and top of the images correspond to detector areas that are not illuminated.

PRNU and relative radiometric response. However, these two contributions actually hamper the stitching process, which is therefore done in two steps. An initial simple step is done by taking, for every pixel, the maximum signal that is observed for all available overlapping measurements. This is for now a good 2-dimensional representation of the signal, see e.g. Fig. 20. In a later step (see Section 6.4.7) a more sophisticated correction for the potential artefacts introduced by this simple approach will be performed.

### 6.4.3 Determine the stimulus spectrum

The output spectrum of the light source is – per definition – independent of the instrument viewing angle or cradle rotation. Any variation of the observed signal in the azimuth (row) direction must thus be attributed to either the optical properties of





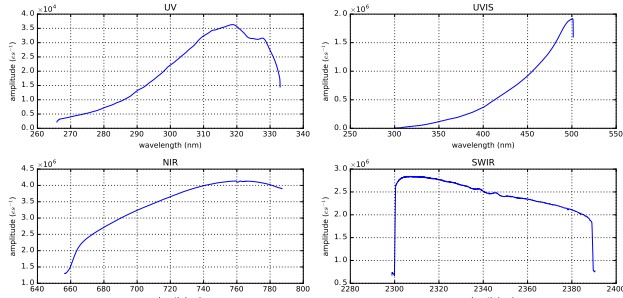

**Figure 21.** The stimulus flux spectrum in observed $e^- s^{-1}$ as a function of the wavelength (horizontal axis) for the UV, UVIS, NIR and SWIR detector. For 310 to 330 nm the UV detector was processed to filter the incoming light, decreasing the relative amplitude with respect to lower wavelengths. For the UV the signal up to approximately 310 nm is not reliable, because the signal is low, and may be straylight. In the NIR spectrum the small oxygen A-band absorption feature at 760 nm can be seen.

the instrument (relative radiometric response) or the detector (PRNU). Therefore, if we divide each observed spectrum in each row by the stimulus spectrum, the PRNU and radiometric response remain. The stimulus spectrum, or the amplitude of the signal per wavelength for all detector pixels, can be obtained using the data from all rows.

First a wavelength is assigned to each detector pixel, and the further steps are done in the wavelength domain. The 1-
5 dimensional spectrum is now calculated as the average over all rows. For the SWIR detector the range of columns covered by one wavelength is quite small, and the signal is quite noisy even though there seems to be no spectral features. For this reason the SWIR spectrum is smoothed using a Chebyshev fit to prevent short scale variations in spectral (column-wise) direction from entering the spectral part of the signal. With this approach these variations are included in the PRNU. This spectrum is normalized to the spectrum at the nadir row.

For the measurements with the integrating sphere, the spectra are shown in Fig. 21. The corrected spectrum is now extended over all rows to yield a 2-dimensional spectrum. This latter spectrum is transformed back from the wavelength domain to the detector domain and referred to as the "stimulus spectrum". If we now divide the original signal by this stimulus spectrum, spectral signatures from the source will be largely removed. Note that due to the spectral smile not all rows see the same wavelength range.

**6.4.4   Calculate (1D) slit irregularity**

As described in Section 6.1, the slit irregularity is disregarded and set to unity. All high-frequency variation after removal of the spectral variation is attributed to the PRNU.





### 6.4.5 Calculate 2D relative radiometric response

The slow variations in the image after removal of the stimulus spectrum are ascribed to the relative radiometric response. They are separated from the high frequency variations by means of performing a 2-dimensional Chebyshev fit to the signal from the illuminated area of the detector. The desired relative radiometric response is now given by the combination of the 1-dimensional

Chebyshev low-pass result and this 2-dimensional filtered data. This combination is then projected upon a wavelength grid and is subsequently column-wise normalized with respect to the value at the reference row for the corresponding wavelength, to ensure that it reflects the *relative* radiometric response. This normalized quantity is projected back to the detector grid, and then returned as the final CKD.

### 6.4.6 Residual spectrum

The column-wise normalization is a consequence from the implementation in the L01b processor that requires that the relative radiometric response is defined to be unity at the reference row. However, if we would now divide the original signal by this relative radiometric response, a column dependent normalization would be falsely introduced. This false normalization can be assigned into a so-called "residual spectrum" $\Lambda_{\mathrm{res}}$,

$$\Lambda_{\mathrm{eff}} = \Lambda \times \Lambda_{\mathrm{res}}, \tag{1}$$

with the stimulus spectrum $\Lambda$ as derived before. For the PRNU derivation the remainder, i.e. the signal after removal of the spectrum and the relative radiometric response, is needed to be centred around unity for all columns. This residual spectrum is also smoothed using a Chebyshev fit to prevent high frequency features to be assigned to stimulus properties.

Once the signal has been divided by the effective spectrum $\Lambda_{\mathrm{eff}}$ and the relative radiometric response, what remains is the PRNU. The results are discussed in Section 4.7.

### 6.4.7 Removal of initial stitch artefacts

The integrating sphere beam profile is not perfectly homogeneous in the sense that the illuminated rows display a small intensity variation that is on average higher on one side of the swath than on the other. This poses a problem during the stitching process and leads to subtle horizontal lines in all aforementioned intermediate results. Because they are subtle, they can best be removed at this stage using a improved stitching algorithm, according to the following scheme: For each column,

we loop along the measurements, or "images" $i$, divided by the spectrum. A mask is generated to include only the well-lit rows of these images, with the medians of their indices as the midpoints $x_{mp,i}$. The overlap between two adjacent measurements $i$ and $i+1$ is $x \in x_o = x_i \cup x_{i+1}$. The error is calculated as:

$$\mathrm{error}_i(a) =$$

$$S_i[x](1 - a(x - x_{mp,i})) - S_{i+1}[x](1 - a(x - x_{mp,i+1})). \tag{2}$$



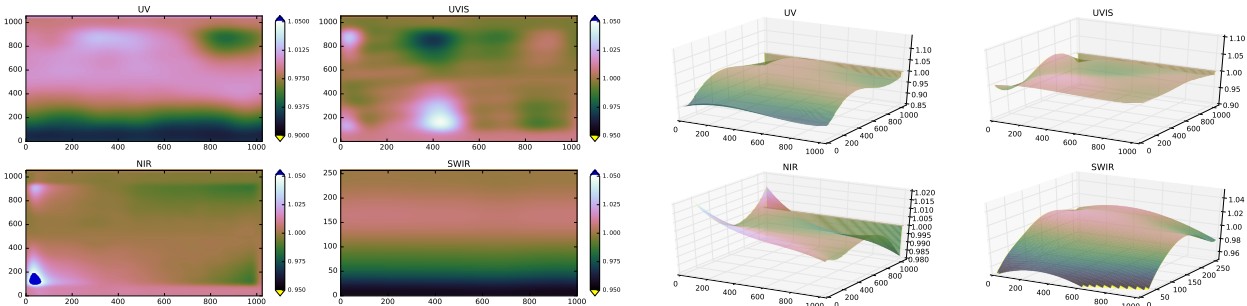

**Figure 22.** The relative radiance response as a function of the detector pixels (horizontal axis the detector column, vertical axis the detector row), for each detector. At the reference row the response is set to unity. The NIR channel does not display large features. For UVIS an asymmetric feature can be seen around column 400. The UV and SWIR channel show a similar decrease of intensity for larger azimuth angles.

In this way the error is the difference of the signal of measurement $i$ and $i+1$ in the overlap area, but both multiplied by the linear function $1 - a(x - x_{mp})$ which is a rotation with respect to their own midpoint, but both by the same amount $a$. This error is minimized for every image ($i$) and column. This gives a value of $a$ for every image and column. Per column, a fit is made through the values of $a$ to reduce fitting artefacts caused by incorrect masking of the signal or large excursions such

as in the UV channel. It is a reasonable assumption to only allow slow variations in the "shape" of the illumination by the integrating sphere. With this fitted value of $a$, or $\langle a \rangle$, the set of images are corrected. Then the images are stitched again using the maximum per row and column.

Subsequently all procedures after and including the derivation of the stimulus spectrum (the determination of spectrum, separation of RELRAD and PRNU) are reiterated to yield the final calibration data for the relative radiance as shown in

Figure 22. The relative radiance in the NIR channel does not display large features. For UVIS there is an asymmetric feature visible around column 400. Both the UV and SWIR channel show a similar decrease of intensity for larger azimuth angles. The relative uncertainty on the relative radiance in Table 9 has been estimated as the standard deviation $\sigma$ of the residuals after double processing.

### 6.5  Absolute radiance

The absolute radiance responsivity calibration (ABSRAD) is the determination of the relation between the electron generation rate in the detector and the absolute spectral radiance at the Earth port. It is derived from measurements of the instrument responsivity using a calibrated FEL lamp that illuminates a calibrated diffuser. Light reflected off the diffuser travels through ambient air and passes the window in the thermal vacuum chamber before it enters the Earth port in vacuum.

The ABSRAD CKD is determined for one reference across-track line-of-sight (swath angle) only. For other swath angles

the RELRAD CKD (see Section 6.4) provides the relation between the electron generation rate for any swath angle and that





for the reference angle. The calibration depends on wavelength and is only valid for this reference swath angle corresponding
to a reference detector row.

Together RELRAD and ABSRAD provide the absolute radiance responsivity calibration for the instrument Earth port.
ABSRAD measurements were performed for various combinations of two calibrated lamps (FEL1 and FEL2), two calibrated

diffusers (DIF1 and DIF2), three lamp-diffuser distances, and three cradle rotation angles. For each measurement a corresponding background measurement was included in the session. The background measurement was performed by shuttering off the direct illumination of the diffuser. The used lamps were NIST calibrated 1000 W QTH FEL lamps with a colour temperature of 3200 K. The diffusers were $300 \times 300 \times$ mm large plates made of sintered PTFE. The diffusers were calibrated both before and after the calibration campaign for the entire TROPOMI wavelength range. The lamp emits light in all directions, baffling

is placed at a distance from the lamp to prevent set-up straylight reaching the instrument, while avoiding overheating of the lamps. Lamp irradiance calibration data, diffuser calibration data of the bidirectional reflectance distribution function (BRDF), and TVC window transmission calibration data, were supplied by industry.

For each experimental condition, a series of repeated images (typically 20–40) is available. These images are averaged to obtain a mean image and a pixel-wise standard deviation is calculated and divided by the square root of the number of

repetitions to obtain the standard deviation of the mean signal. Mean images and standard deviations of the mean are calculated for all illuminated and background measurements, which are then processed in the L01b data processor up to and including the RELRAD correction. Background images are subtracted from the corresponding illuminated images, and results are corrected for the spectral smile.

In the smile-corrected images, the row with line-of-sight (LOS) closest to the optical axis is selected for further processing.

This row is referred to as the "optical axis row". It is assumed that the nominal diffuser angles of incidence and reflection applying for the "true" optical axis, also apply for the optical axis row. Obviously, the maximum possible misalignment between the optical axis row and the true optical axis is half a row. This corresponds to an estimated difference of at most 50 % of the local signal gradient per row between the measured signal and the signal that would have been obtained if the alignment would have been perfect. This difference is estimated from measured spatial signal profiles, and included as a contribution to the

overall absolute ABSRAD CKD error. Rows other than the optical axis row are only used to obtain signal profiles and are not included in the ABSRAD CKD assessment. These rows are illuminated under non-nominal angles of incidence or reflection. Both the BRDF and lamp calibration data are potentially inaccurate for non-nominal angles.

Measured signals are assumed to represent radiance along each pixel's LOS, thus ignoring possible effects of finite pixel footprint on the diffuser. The relative error associated with this assumption is estimated to be 0.02 % for UVN and 0.1 % for

SWIR. The relative error is included in the overall absolute ABSRAD CKD error.

Measurements obtained for nominal lamp-diffuser distances $d_1$, $d_2$ or $d_3$ are tested for compatibility with the $1/r^2$ law. For each nominal distance, the corresponding *physical* (i.e., real) lamp–diffuser distance is calculated taking into account corrections for the lamp filament position offset and average diffuser penetration depth. Compatibility with the $1/r^2$ law is tested by multiplying each signal by the square of the corresponding physical distance, also taking into account a further

geometrical correction for vertical diffuser off-centre distance. This vertical distance has an effect on the lamp–diffuser distance




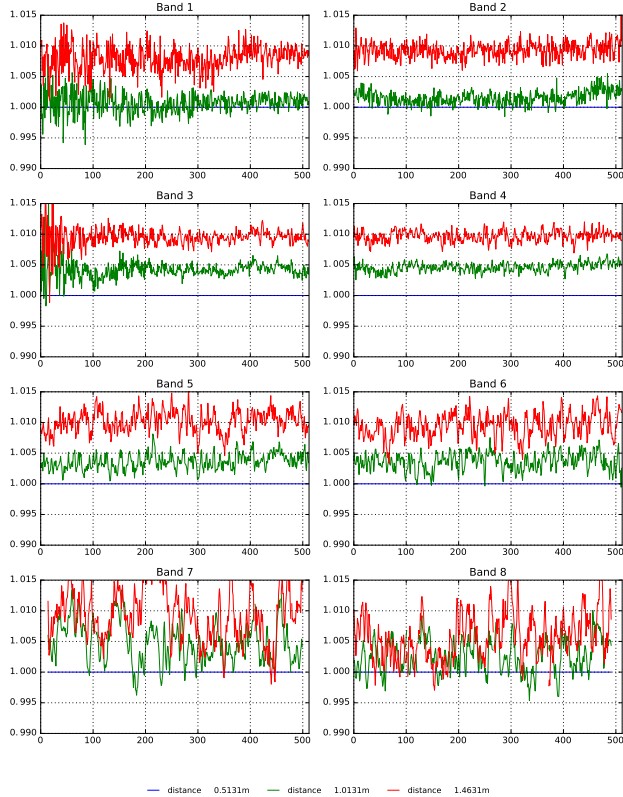

**Figure 23.** Results obtained in bands 1–8 for different lamp–diffuser distances $d_1$, $d_2$ or $d_3$, for lamp FEL1, diffuser DIF1, and cradle rotation angle $0°$. Signals were multiplied by the square of the distance and then divided by the result obtained for distance $d_1$; thus, roughly unity results are expected for distances $d_2$ and $d_3$. Results obtained for distance $d_2$ (green lines) are generally 0.1-0.5 % higher than expected, whereas for distance $d_3$ (red lines) discrepancies up to 1 % are obtained in all detectors.

as well as on the angle of incidence. Distance effects related to inhomogeneous irradiance of the field-of-view of detector pixels in the optical axis row are less then 0.02 %, and are therefore ignored. Results are then expected to be independent of lamp–diffuser distance: taking the ratio of each result and that obtained for distance $d_1$, should equal unity. As shown in Fig. 23 results obtained for distance $d_2$ are generally 0.1-0.5 % higher than expected, whereas for distance $d_3$ discrepancies up to 1 %

5   are obtained in all detectors.

Averaged results obtained for each of the four lamp–diffuser combinations are corrected for spectral features: spectral anomalies were observed in detectors 1–3, which were identified as oxygen A-band absorption (at $\approx 760\,\mathrm{nm}$), or various emission or absorption lines in the lamp output spectra. The spectral range of each anomaly is determined by inspection, and the corresponding area is invalidated and interpolated by a $3^{\mathrm{rd}}$ order Chebyshev fit of the data in two regions flanking the invalidated

10   area. Maximum regions are selected that still yielded a good fit result. If no satisfactory fit can be obtained for any reasonably-



sized fit region, no interpolation is performed, leaving the anomalous area invalidated. Some features comprise closely spaced double spectral lines; these are interpolated using a joint $4^{\text{th}}$ order Chebyshev fit that included the region between the two lines.

After invalidation or interpolation of spectral anomalies, results for each of the four lamp–diffuser combinations are spectrally smoothed in each band separately by low-pass Fourier-filtering. This is not accomplished by traditional Fourier transforms, but by fitting a series of sines to the signal that also included half-wave sines as well as linear and constant terms. The linear term is introduced to account for overall signal gradient (note that this is not possible using mere Fourier transforms, which implicitly assume a periodic signal). The number of terms minus 2 thus equals the maximum number of half-sines used to describe the signal. A minimum number of terms is included that appeared necessary to obtain a good fit: fit residues were inspected for lack-of-fit, which then would call for a larger number of fit terms. If only high-frequency residues remain, the fit result is taken as the filtered signal. Results of the smoothing are shown in Figure 24.

For each of the four combinations of lamps FEL1 or FEL2 and diffusers DIF1 or DIF2, the absolute radiance at the Earth port is calculated from the (interpolated) calibration data of the lamps, diffusers, and TVC window transmission and by applying the appropriate geometrical correction factor. The ABSRAD calibration results for the optical axis row are obtained by dividing absolute radiance at the Earth port by the smoothed results for the corresponding lamp–diffuser combination. Results are then compared pair-wise for each combination obtained with identical lamp or diffuser, and an averaging procedure is applied. For the final ABSRAD CKD the ABSRAD result obtained for the optical axis row is duplicated to all remaining detector rows and the spectral smile is re-applied.

Figure 26 shows the estimated ABSRAD pixel-to-pixel uncertainty, the corresponding variance is the square of this quantity. Wave-like patterns are observed in all detectors. They appear to be correlated with the number of terms included in the spectral smoothing procedure. Therefore, the patterns are probably an artefact of the smoothing procedure and the results should be used only as a general indication of order of magnitude. The relative uncertainty is in the order of $10^{-4}$–$10^{-5}$. This is (much) lower than the measurement noise obtained in the ABSRAD measurements. Indeed, noise is effectively suppressed by the spectral smoothing procedure.

Various effects have been identified that contribute to the absolute ABSRAD uncertainty, these are summarized in Table 9. As the effects causing the uncertainty are considered mostly statistically independent, the individual contributions are added quadratically to obtain the overall absolute ABSRAD uncertainty. The ABSRAD CKD is calculated as an average of results obtained for different lamps, diffusers, and cradle rotation angles. The CKD has been derived from measurement data obtained for *nominal* diffuser angles of incidence or reflection only (i.e., pertaining to the diffuser centre). For angles of incidence or reflection other than the nominal angles, the calibration of diffusers and lamps is potentially inadequate.

The CKD is determined with a small ($< 0.01\,\%$) pixel-to-pixel uncertainty, but a much larger *absolute* pixel uncertainty is obtained. The latter is 1–2 % for UV, 1 % for UVIS and NIR, and 1.5 % for SWIR. These values are within the requirements on Level 1b.

The uncertainty mainly reflects discrepancies between the results obtained for the various combinations of lamps and diffusers. These discrepancies can in part be attributed to lamp or diffuser calibration uncertainty, the remaining differences





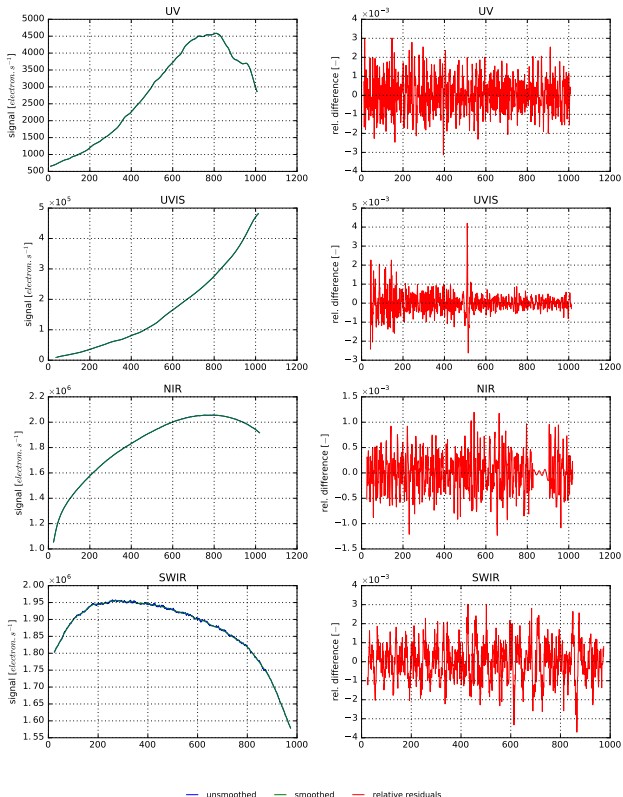

unsmoothed    smoothed    relative residuals

**Figure 24.** Typical results of spectral smoothing. **Left panels**: for each detector, signals before (blue lines) and after smoothing (green lines); except for SWIR, differences between both signals are hardly visible on these scales. **Right panels**: corresponding (relative) residuals as a ratio to the original signal, showing mainly high-frequent differences between the unfiltered and filtered signals. Somewhat lower-frequent features may be observed in de residuals for SWIR (e.g. between columns 850–900), but there is no correlation between these and similar features obtained for diffuser DIF2 (not shown), indicating that the effects are diffuser-borne. The "gap" in the NIR-residuals between columns 800–900 corresponds to the interpolated oxygen A-band.

(0–1.5 %) are unexplained. The contributions to the uncertainty are listed in Table 9. In all detectors, results obtained for different cradle rotation angles, and results obtained for different lamp-diffuser distances, all agree to within 1 %.

## 6.6 Absolute irradiance

The absolute irradiance CKD is the ratio of the spectral irradiance at the internal diffuser (QVD) to the response that is measured. To calculate the irradiance at the diffuser, the irradiance of a FEL lamp, which has been calibrated by NIST at 50 cm, must be converted to the irradiance at the distance between the Sun diffuser and the FEL lamp. The same lamps are used as for the absolute radiance described in Section 6.5. The FEL is considered to be a point source in this setup so the inverse-square law can be used for this conversion. However, the distance between the FEL lamp and the diffuser must be





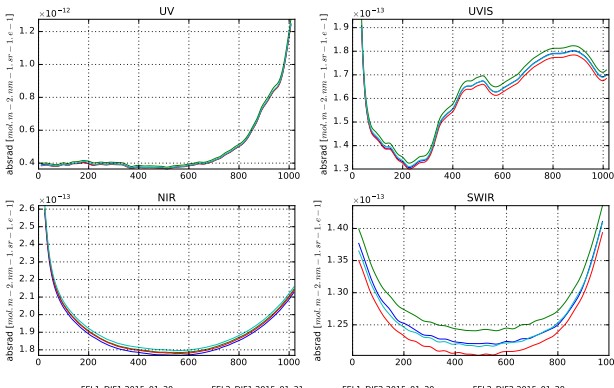

**Figure 25.** ABSRAD calibration for the optical axis row for all spectrometers. Shown is data for lamps FEL1 and FEL2 for both diffusers DIF1 and DIF2. For UVIS, sharp changes in gradient can be observed, e.g. at column 512. They might be related to quantum efficiency changing abruptly with wavelength.

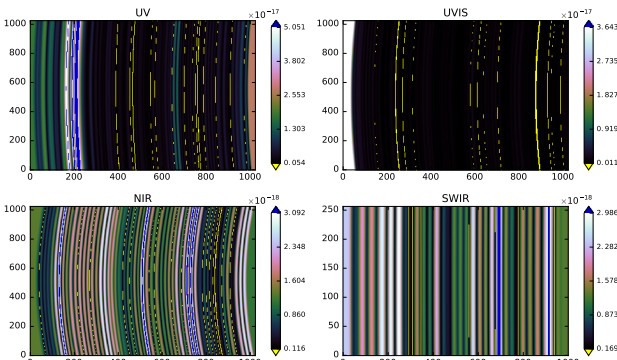

**Figure 26.** Pseudo-colour image of the ABSRAD CKD error, defined as the uncertainty in CKD pixel-to-pixel variation. The colour scale unit is $[\mathrm{mol\,m^{-2}nm^{-1}sr^{-1}}(e^{-})^{-1}]$. The corresponding variance, $\sigma^2_{f\mathrm{absrad}}[\mathbf{x}_{\mathrm{det}}]$ is the square of this quantity. The wave-like patterns observed in all detectors are an artefact of the spectral smoothing procedure.

determined accurately since an error in the distance will be propagated squared into the irradiance. For instance, at $1.5\,\mathrm{m}$, an error of $1\,\mathrm{cm}$ in the distance $(0.67\,\%)$ will result in an error of $1.33\,\%$ in the irradiance. As it turns out, it is not possible to physically measure the distance with the desired accuracy: the coil of the FEL lamp extends a few millimetre and thus makes it impossible to designate a single distance from where the light originates. More important, because the internal diffusers are volume diffusers, there is not just a single surface within the diffuser where the light reflects. Therefore, another approach is necessary to determine the distance: a distance offset is calculated from the measurements and added to the measured distance. Such an alternative approach is unavoidable when using a volume diffuser.



The absolute irradiance measurements are repeated with the FEL lamp placed at three positions: approximately at 1.5, 2.0 and 2.5 m distance from the Sun diffuser. They are labelled $d_1$, $d_2$ and $d_3$ respectively. The FEL lamp is moved to these positions on a rail, which means that the relative distances between those three positions are known with a high accuracy. This gives us, for each detector pixel, three equations with two unknowns: irradiance and lamp position. In principle this could be

solved with a (non-linear) fit. However, instead of calculating millions of fits through three data points (one fit per detector pixel), we can calculate the distance offset analytically for two FEL positions. The three FEL positions can be paired in three combinations ($d_1$ and $d_2$, $d_1$ and $d_3$, $d_2$ and $d_3$) and the distance offsets calculated from these pairs can be compared to each other. If they yield similar values their average can be used as the final distance offset. Note that the above is only valid as long as there is no straylight in the setup that invalidates the point source law.

The QVDs are tilted with respect to the optical axis. During the absolute irradiance measurements the angle between this diffuser normal and the optical axis was about $25°$. Given that the QVD has a width of approximately 5 cm, the end corresponding to a positive swath angle (west) is located about 2 cm further from the FEL lamp than the opposite end (east). The FEL lamp can be viewed as a point source, so due to the inverse squared law the west end will receive a lower irradiance than the east end. If the diffuser is perfect, i.e. light falling on a spot on the diffuser is uniformly spread out over all detector rows,

the derived distance offset will be independent of detector row. For an imperfect diffuser, which spreads the light over a limited number of rows, a row-dependency with a maximum of 2 cm will be present in the distance offset. The data show a variation over the rows of 1 to 2 cm. In orbit no variation of irradiance is expected since the Sun light illuminates the QVD with nearly parallel beams. Therefore this effect is corrected for by using a row-dependent distance offset. The distance offset is calculated for each detector pixel separately and subsequently averaged per row. Then a line is fitted, which yields a linear function that

maps each row to a distance offset. The three linear functions, calculated from the three distance pairs, are averaged to give one distance offset function per detector.

The absolute irradiance CKD is calculated with the FEL lamp Sun port measurements only. In principle one could use the Sun simulator measurements to calculate the instrument bi-directional scattering function (BSDF) and use this to convert the ABSRAD CKD to absolute irradiance values. The benefit of this would be a better signal to noise ratio, since both the Sun sim-

ulator and the FEL over the Earth port give a higher response than the FEL over the Sun port. However, the Sun simulator output was highly unstable and could not be used to calculate the BSDF.

Measurements were executed for both internal diffusers (QVD1 and QVD2) at three distances ($d_1$, $d_2$ and $d_3$). All measurements were repeated with a second FEL lamp. Next to that, a validation measurement was done with FEL-1 at distance $d_3$ with the same instruments settings as the corresponding regular measurement. The validation measurement is not used in the CKD

analysis, only for validation. All measurements where done at cradle rotation $0°$ and tilt -1.28°, the same cradle position that was used as the reference during the relative irradiance measurements.

The CKD is the mean of the absolute irradiance values calculated from FEL1 and FEL2 measurements at the shortest distance $d_1$. The absolute irradiance CKD is plotted for the optical axis rows in Fig. 27.

The absolute irradiance for QVD2 is about 5 % to 10 % higher than the QVD1 absolute irradiance, which means that the

response via QVD2 is 5 % to 10 % lower given the same amount of light. The absolute irradiance increases at the edges of the





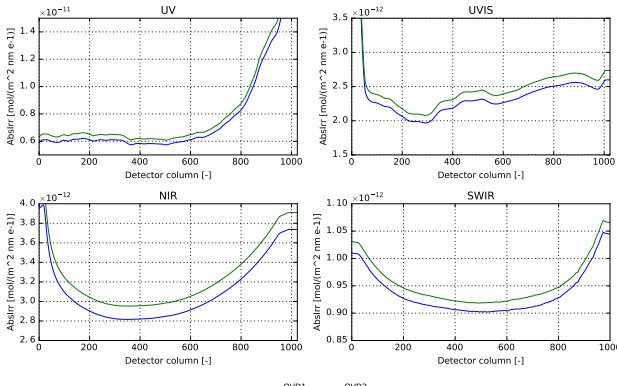

**Figure 27.** Absolute irradiance CKDs at the optical axis row (QVD1 in blue, QVD2 in green) for all spectrometers. For each QVD the CKD is the mean of the absolute irradiance calculated from FEL1 and FEL2 at distance $d_1$.

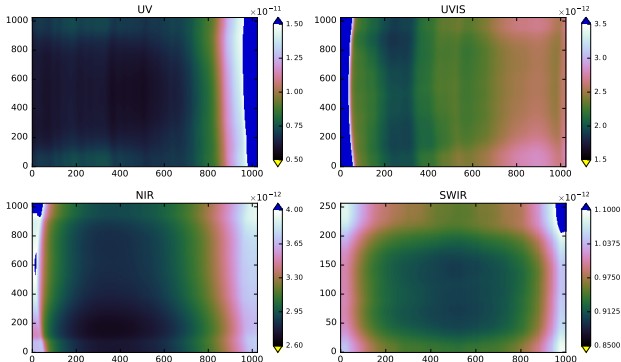

**Figure 28.** Image plots of absolute irradiance CKD of QVD1 for all spectrometers. The dark blue pixels are where the ABSIRR is off the scale, this corresponds to a responsivity close to zero. The values at the edge of the detectors are calculated with nearest-neighbour extrapolation.

detectors where the responsivity of the spectrometers decrease. For detector pixels that do not have an associated wavelength it is not possible to calculate an absolute irradiance. For those pixels the values have been calculated by extrapolation of the nearest valid pixel.

Figures 28 and 29 give an impression of the row dependence of the absolute irradiance CKD. The former shows the absolute
5   irradiance of QVD1 as image plots, the latter shows cross section for a few selected columns. Only the graphs of QVD1 are reported as QVD2 gives similar results. The absolute irradiance varies slowly over the rows, as expected. In the UVN detector the absolute irradiance stays at a similar level for all illuminated detector rows and it increases sharply at rows that are not illuminated. This in contrast with the SWIR detector where the absolute irradiance of the illuminated rows increases slowly for rows closer to the edge.





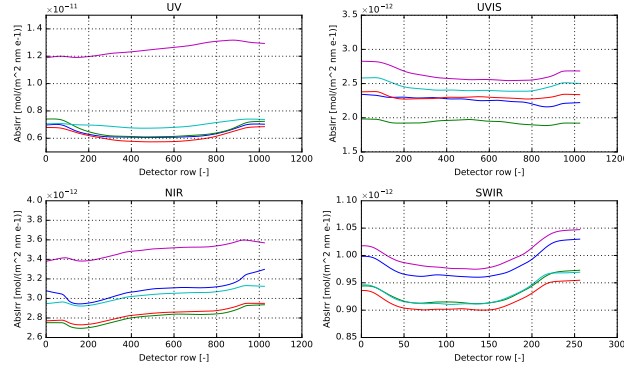

**Figure 29.** The absolute irradiance CKD for QVD1 is shown for each detector in columns 100 (blue line), 300 (green), 500 (red), 700 (turquoise) and 900 (mangenta) for each detector.

The roughness of the diffuser surfaces causes interference patterns (speckle) that are dependent on the wavelength, viewing angle, and location on the diffuser. They are hard to characterize because of the many factors that influence them, so they are smoothed out. Under flight conditions the SWIR diffuser features will be reduced by averaging over the elevation direction with 150 measurements, thus effectively scrambling these patterns. Optionally, if the solar spectrum is assumed to be stable

over a certain time period, the features can be reduced even further by averaging the solar irradiance of orbits that are measured at different azimuth angles. The diffuser features in the SWIR follow a normal distribution with a standard deviation of $0.4\,\%$ for both QVDs. With the UVN detectors no diffuser features could be observed.

To test the repeatability of the analysis, the FEL1 measurements at the longest distance ($d_3$) where performed twice, using the same instrument settings. Even though the baffling of the validation measurement was placed incorrectly, it differs less

than $0.2\,\%$ from the regular measurement. As a validation, the absolute irradiance calculated from the measurements at the three distances are compared. After applying the distance offset, the differences are within $0.5\,\%$. As a further validation, the absolute irradiance calculated with FEL1 and FEL2 are compared: they show differences up to $1.5\,\%$. The CKD is the mean of the absolute irradiance values calculated from FEL1 and FEL2 measurements at the shortest distance $d_1$. The uncertainty for ABSIRR is between $0.8\,\%$ for NIR and $1.2\,\%$ for UV and UVIS. These values are within the requirements on Level 1b. The

different contributions to the uncertainty are shown in Table 9.

### 6.7 Instrument BSDF

The instrument bi-directional scattering function (BSDF) is not directly used in the L01b data processor. It can be expressed as the ratio of the irradiance responsivity to radiance responsivity, which *are* quantities used by the L01b data processor to calculate the solar irradiance and Earth-shine radiance respectively. Level 2 algorithms such as DOAS indirectly take the ratio

of these two quantities when calculating the reflectivity, which makes the instrument BSDF an important parameter in many Level 2 algorithms.





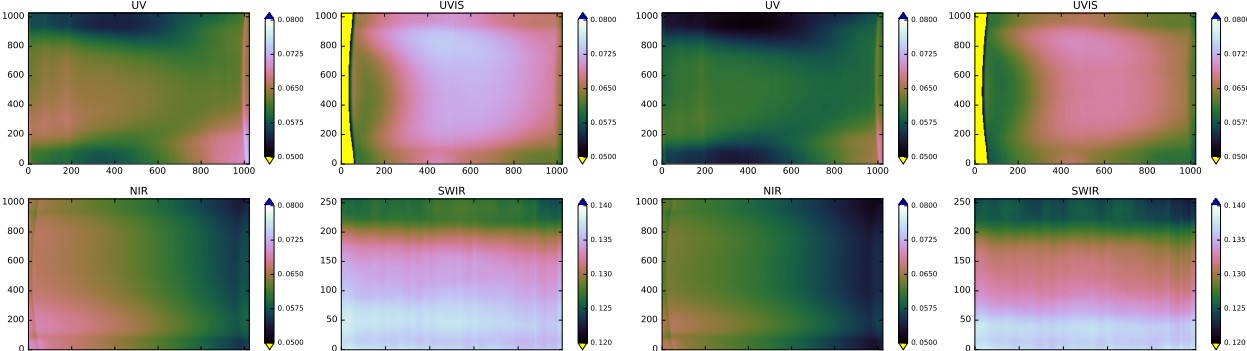

**Figure 30.** The BSDF calculated from the ABSRAD and ASBIRR CKD, which in turn are calculated from the FEL lamp measurements. The top four images show the QVD1 BSDF per spectrometer, the bottom four show the BSDF for QVD2. The yellow and blue pixels have a BSDF that is off the scale, caused by the fact that these pixels had little illumination in one or more of the FEL measurements. Note that the extreme detector rows have been calculated by nearest neighbour extrapolation in the absolute irradiance, and therefore do not represent the actual BSDF.

The instrument BSDF was initially planned to be determined from measurements using the Sun simulator (see Section 6.1). However, this stimulus was not sufficiently stable for this purpose, resulting in uncertainties from 5 % up to 30 %. As a backup, measurements with an integrating sphere as described in Section 4.7 were performed. These measurements resulted in a too low signal-to-noise ratio for bands 1 and 3. Therefore the instrument BSDF has been derived from measurements for the calibration

of absolute radiance and irradiance with the absolute calibrated FEL lamps and external calibrated diffusers (see Section 6.5).

Figure 30 shows an image plot of the FEL-BSDF, which is the BSDF calculated from the ABSRAD and ABSIRR key data. The top four plots show the BSDF of QVD1, the bottom four show the BSDF of QVD2. The BSDF slowly varies as function of wavelength and viewing angle, which is as expected. It lies between 0.55 and 0.75 for UVN and between 1.23 and 1.35 for the SWIR detector.

Figure 31 contains line plots for a few selected rows. These rows correspond to the cradle rotation angles: -50°, -30°, 0°, 30° and 50°. Cradle rotation angles -50°, and +50°, are at the edge of the instrument field-of-view. The rows at -30°, 0°, and -30°, are representative for regular viewing angles, the optical axis row is illuminated at rotation angle 0°. Figure 32 shows cross sections of the BSDF for the detector columns 100, 300, 500, 700 and 900. In both the ABSRAD and ABSIRR analysis, the values at the extreme detector rows and columns have been calculated by extrapolation.

Note that the BSDF of QVD1 is about 5 % to 10 % higher than the QVD2 BSDF, which is due to the fact that the absolute irradiance is the same amount lower for QVD1 than for QVD2. This means that the instrument response via QVD2 will be 5 % to 10 % lower given the same amount of light.

The relative uncertainty is found to be 1.1–1.8 % for the UV, 1.2–1.3 % for the UVIS, 1.1 % for the NIR, and 1.5–1.8 % for the SWIR. Apart from band 8, these values are within the requirements on Level 1b. The different contributions to the

uncertainty are listed in Table 9.

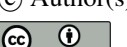


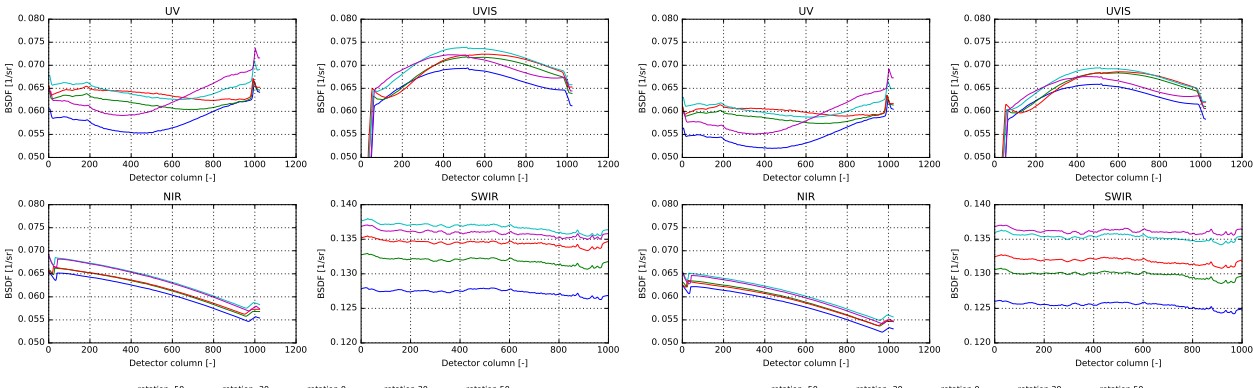

**Figure 31.** The FEL-BSDF plotted for the detector rows corresponding to cradle rotation angles -50°, -30°, 0°, 30°, and 50°. The FEL-BSDF is calculated from the ABSRAD and ASBIRR CKD, which in turn are calculated from the FEL lamp measurements. The top four images show the BSDF for QVD1 per spectrometer, the bottom four show the BSDF for QVD2. Note that the extreme detector columns have been calculated by nearest neighbour extrapolation in the absolute irradiance, and therefore do not represent the actual BSDF.

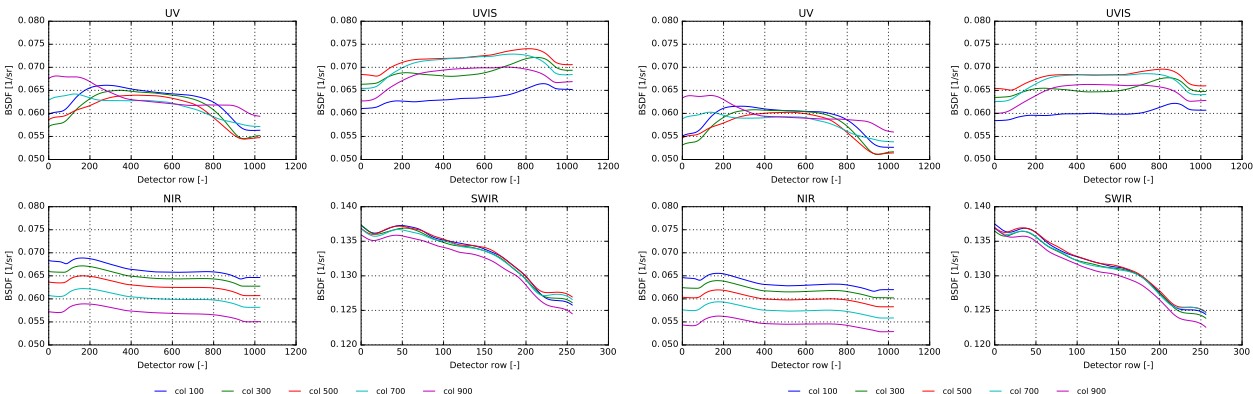

**Figure 32.** The FEL-BSDF plotted for the detector columns 100, 300, 500, 700 and 900. The FEL BSDF is calculated from the ABSRAD and ASBIRR calibration key data, which in turn are calculated from the FEL lamp measurements. The top four images show the BSDF for QVD1 per spectrometer, the bottom four show the BSDF for QVD2. Note that the extreme detector rows have been calculated by nearest neighbour extrapolation in the absolute irradiance, and therefore do not represent the actual BSDF.

## 6.8 Relative irradiance

The relative irradiance factor is the correction of the signal for the angle under which the Sun port is illuminated. A range of measurements with the Sun simulator was performed for different azimuth (seasonal variation, $\varphi$) and elevation ($\epsilon$) angles. The Sun simulator is a collimated and homogenized 1000 W Xe lamp illuminating the instrument via a calibrated diffuser. The same diffuser was used for the absolute radiance calibration (see Sec. 6.5). These measurements are normalized with





**Table 9.** Summary of the relative standard uncertainty components per detector for ABSRAD, RELRAD, ABSIRR and instrument BSDF The quantities marked with $^*$ are not part of the BSDF uncertainty because they divide out. Note that setup straylight cannot be reasonably estimated and might be included in the components "Unexplained lamp/measurement discrepancy". The total uncertainty was obtained by quadratically adding each individual contribution.

| CKD | Uncertainty source | UV | UVIS | NIR | SWIR |
|---|---|---|---|---|---|
| ABSRAD | Measurement noise [%] | <0.01 | <0.01 | <0.01 | <0.01 |
| ABSRAD | Finite pixel foot print [%] | 0.02 | 0.02 | 0.02 | 0.1 |
| ABSRAD | Optical axis row mis-alignment [%] | 0.05 | 0.1 | 0.1 | 0.7 |
| ABSRAD | Cradle rotation discrepancy [%] | 0.5 | 0.5 | 0.5 | 0.2 |
| ABSRAD | Lamp-diffuser distance [%] | 0.06 | 0.06 | 0.06 | 0.06 |
| ABSRAD, ABSIRR$^*$ | Lamp calibration uncertainty [%] | 0.8 | 0.4–0.8 | 0.3 | 0.3–0.5 |
| ABSRAD | Diffuser calibration uncertainty [%] | 0.3 | 0.3 | 0.2 | 1.0 |
| ABSRAD, ABSIRR$^*$ | TVC window calibration [%] | 0.1 | 0.06 | 0.06 | 0.1 |
| ABSRAD | Unexplained measurement discrepancy [%] | 0.0 - 1.5 | 0.0 - 0.5 | 0.6 | 0.0 - 0.9 |
| RELRAD | $\sigma$ after double processing [%] | 0.1 | 0.07 | 0.04 | 0.1 |
| ABSIRR | Measurement noise (after smoothing)[%] | 0.03 | 0.01 | 0.001 | 0.001 |
| ABSIRR | Spectral features (after smoothing)[%] | 0.15 | 0.0 | 0.0 | 0.15 |
| ABSIRR | Alignment error[%] | 0.016 | 0.016 | 0.016 | 0.016 |
| ABSIRR | Distance offset [%] | 0.5 | 0.5 | 0.5 | 0.5 |
| ABSIRR | Unexplained FEL lamp discrepancy [%] | 0.7 | 0.9 | 0.6 | 0.7 |
| | **Total uncertainty ABSRAD** | **1–2 %** | **1 %** | **1 %** | **1.5 %** |
| | **Total combined uncertainty ABSIRR** | **1.2 %** | **1.2 %** | **0.8 %** | **1.0 %** |
| | **Total combined uncertainty FEL-BSDF** | **1.1–1.8 %** | **1.2–1.3 %** | **1.1 %** | **1.5–1.8 %** |

measurements taken at the *reference* angle $\varphi_0, \epsilon_0$, an angle close to the nominal Sun angle. The reference measurements were taken at regular intervals to also monitor possible drifts in the output of the stimulus.

The CKD is a correction factor for all detector pixels and for each azimuth–elevation angle pair, with which irradiance measurements are to be multiplied in order to remove the angular dependence of the Sun port. The CKD is defined as coefficients

5 of a polynomial for pixels on a reduced grid, so that the L01b processor interpolates between the pixels to cover the whole detector, and expands the polynomial to cover all possible azimuth and elevation angles within the valid range.

The signal for each $\varphi, \epsilon$ is averaged over the valid measurements and the corresponding background (dark) image is subtracted. The relative irradiance is the ratio of the signal image with respect to the signal image at the nominal angle.

The reference measurements are pixel-wise interpolated over time to provide for each measurement a reference close in time

10 with which normalization can be performed. In this way, both the relative irradiance is calculated and the stimulus drift is taken care of. However, analysis showed that for the later measurement sessions the output of the reference measurements does not





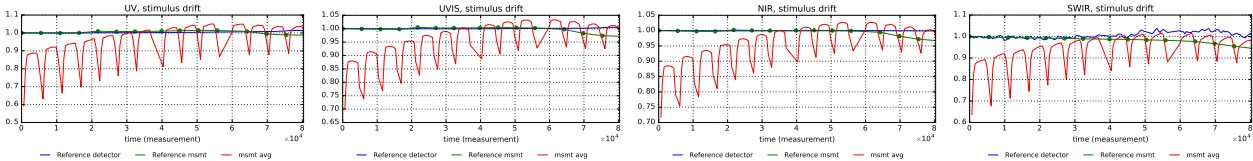

**Figure 33.** The drift of the stimulus with respect to the first available measurement during the QVD2 calibration as measured by the reference detectors for the stimulus (blue) and the TROPOMI reference measurements (green), for the different spectrometer wavelength ranges. The measurement average (red) displays a relatively large variability because of the vignetting caused by the partial blocking of the stimulus for large elevation angles.

seem to correspond to the output of the normal measurements. This is attributed to short term source instabilities. Therefore, this regularization method is not used in the derivation of the CKDs, even though some measurements would benefit from some kind of normalization. The output of reference detectors monitoring the source output can – in principle – be used to estimate the stimulus drift during the measurement, using estimates for the offset of the reference detectors. However, the trends in
the reference detectors were found to differ substantially from the trends in the reference measurements performed with the instrument, so the reference detectors were considered unreliable and discarded.

For each detector pixel the irradiance as a function of discrete values of the azimuth and elevation angle can be retrieved. However, this yields a large dataset with high uncertainty due to noise in the measurements. Therefore, a coarse detector grid $r_c, c_c$ is used for the computation. For these selected detector pixels the measurements are averaged over an area of typically
10 pixels to reduce variability. Then a low-order ($4 \times 4$-dimensional) Chebyshev fit is made by extrapolation of the fit. This fit is valid between azimuth angle $-12° < \varphi < 12°$ and elevation angle $-6° < \epsilon < 6°$. The parameters of this fit are the key data used in the L01b processor. The measurements that suffer from vignetting are excluded in the construction of the polynomial fit by inspection.

### 6.8.1   Stimulus drift

Because the instrument was scanned along the azimuth and elevation angle, a drift in stimulus output could be responsible for a pattern in the relative irradiance, such as a line with constant azimuth angle $\varphi$, because for each azimuth angle the instrument was scanned across the elevation angles. It can be derived from either the reference detector signals, or from the reference measurements that were taken at the reference cradle position (tilt = -1,278°, rotation = 0°) after each azimuth scan. The measurements per cradle angle are short enough to take one reference detector value per measurement to calculate the
relative stimulus output. In Fig. 33 the output of the reference detectors and the mean of the reference measurements is shown.

First of all, it appears that the reference detector signals show a different behaviour than the measurements taken with the instrument at the reference angle. The temporal resolution of the reference detectors is substantially higher than the instrument data: a measurement at the reference cradle position was performed each time that the cradle azimuth angle changed (only once in about every 70 min). More importantly, for all TROPOMI detectors the signal of the measurements at the reference



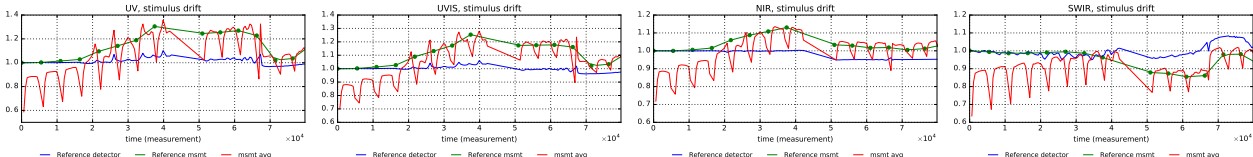

**Figure 34.** As Fig. 33, but now for QVD1. The stimulus drift is hard to quantify because the offset in the reference detector (blue) seems to change in the last measurement session, where the measurements at the reference angle (green) diverge from the normal measurements (red). In this case the relative irradiance would improve when the measurements at the reference angle were used to correct the data.

angle decreases for the later part of the measurement session (1 % for UV to 3.5 % for NIR, and more than 5 % for SWIR). This behaviour is as of yet not explained. When a correction factor based upon the reference detectors is used, the validation of the CKD via double processing gives unsatisfactory results. Therefore, at the moment no stimulus correction is used in the algorithm. An important effect is that in this way the relative irradiance is not identical to unity for the measurements at the

nominal angle, with deviations up to 3 %. This can easily be retrieved using proper normalization, but it is questionable if this would lead to a better CKD. Eventually the CKD for QVD1 was discarded due to the source instabilities. The values of QVD2 were copied into the CKD for QVD1.

In Fig. 35 the NIR detector image taken for different azimuth and elevation angles via QVD2, is divided by the corresponding reference image. This results in the relative irradiance. The references are interpolations of different reference measurements

to the time of measurement of the corresponding azimuth and elevation angle. The normalization with the reference images makes the xenon line features disappear. This indicates that the measurements with different azimuth and elevation angles share the same spectral features with the same respective ratios. Furthermore, the detector is evenly illuminated, with slight horizontal features that switch sign when decreasing the azimuth angle.

The relative irradiance as a function of azimuth and elevation angle $(\varphi, \epsilon)$ is given in Fig. 36 for a number of pixels. The

signal shows a smooth dependence on the azimuth and elevation angle per pixel, and that the maximal signal is attained for azimuth/elevation angle $\varphi, \epsilon \approx (-8°, 0°)$. The relative irradiance displays no large (qualitative) variability as a function of the detector pixel.

In Fig. 37 the relative irradiance as a function of azimuth and elevation angle is given for different detector pixels. It is a reasonably smooth function for the azimuth angles tested (-9.7°–10.7°) and for elevation angles between -3.4° and 3.4°.

For larger elevation angles the images are visibly vignetted because the Sun port baffle is blocking the stimulus. This is a characteristic of the instrument, so the same range of elevation angles will be covered in-flight.

The relative irradiance is a 4-dimensional dataset: for each detector pixel the correction factor with respect to any azimuth and elevation angle is required. To reduce the chance of fitting noise, reducing the amount of computations and data in the process, the relative irradiance is considered on a reduced grid: the coarse grid $r_c, c_c$. For each pixel on the coarse grid the data

is fitted in the azimuth-elevation angle plane. This is because the number of measured angles is relatively small (14 and 10, respectively) and the dependence of the relative irradiance on the azimuth and elevation angle seems smooth. The measurement





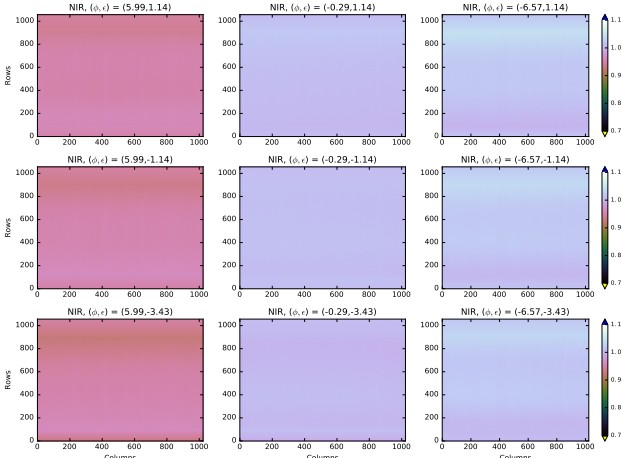

**Figure 35.** The relative irradiance via QVD2 for the NIR detector as function of detector coordinates (row, column) for different azimuth (left, middle and right figure column) and elevation angles (top, middle and bottom figure row), divided by the reference image. When the signal is normalized with respect to a reference image the xenon line features disappear, which indicates that the measurements with different azimuth and elevation angles share the same spectral features with the same respective ratios. Furthermore, the detector is evenly illuminated, with slight horizontal features that switch sign when decreasing the azimuth angle (here from left to right).

noise in the images is not in any way fitted, or expected to decrease by applying a relative irradiance correction factor. The fit was made to both the values corresponding to the single pixels on the coarse grid and to an average of the measurements over a number of pixels around the pixels on the coarse grid.

For the pixels on the coarse grid a 2-dimensional $4^{\text{th}}$ order Chebyshev fit is performed in the azimuth-elevation direction. Both a $4^{\text{th}}$ and a $5^{\text{th}}$ order Chebyshev polynomial in both directions (including crossterms) have been fitted to the data. Because of the vignetting at the largest elevation angles, it is not attempted to fit the decrease in the signal as a functional dependence of the elevation angle. Therefore, the fit is performed using data between elevation angle $-3.4° < \epsilon < 3.4°$, the azimuth and elevation angles for which no sign of vignetting was visible. The fit was subsequently extrapolated to give a relative irradiance for on the domain $-12° < \varphi < 12°$ and $-6° < \epsilon < 6°$. We know that the instrument will be blocking light coming from the extreme azimuth and elevation angles, but in this way the fit is optimal for the smaller angles. Since an extrapolation is done, the fourth order polynomial is preferred for the CKD because the asymptotic behaviour is symmetric for large arguments, just as the relative irradiance is expected to behave.

The calibration of the relative irradiance yields key data that largely removes the dependency of the Sun port measurements on the azimuth and elevation angle under which they were performed.

The order of the polynomial with which the dependency on the azimuth and elevation angle is fitted, does not seem to influence the success of the calibration much. The residuals after using $4^{\text{th}}$ order Chebyshev polynomials show a slight visible pattern, but care must be taken not to overfit the signal. Double processing yields good results, effectively removing relative irradiance from the measurements using QVD2. The standard deviation of the signal decreases, with a remaining deviation





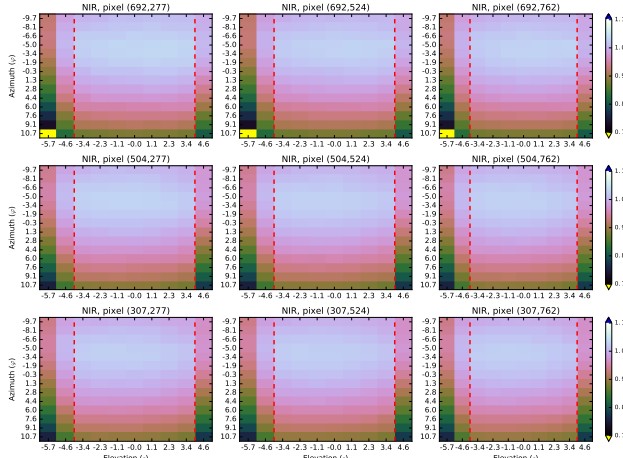

**Figure 36.** The relative irradiance response of the instrument as a function of azimuth angle $\varphi$ and elevation $\epsilon$ at a number of pixels (figures correspond to different pixel locations on the detector) for the NIR detector. The red dashed lines indicate the angles between which the data is considered not to be affected by vignetting due to the baffling. Outside of those lines ($|\epsilon| > 3.4°$) the value of the relative irradiance decreases precipitously. In the azimuth angle direction a constraint seems not to be necessary, see also Fig. 37. The signal shows a smooth dependence on the azimuth and elevation angle per pixel, and that the maximal signal is attained for azimuth/elevation angle $\varphi, \epsilon \approx (-8°, 0°)$. Furthermore, all figures are alike, indicating that the relative irradiance displays no large (qualitative) variability as a function of the detector pixel.

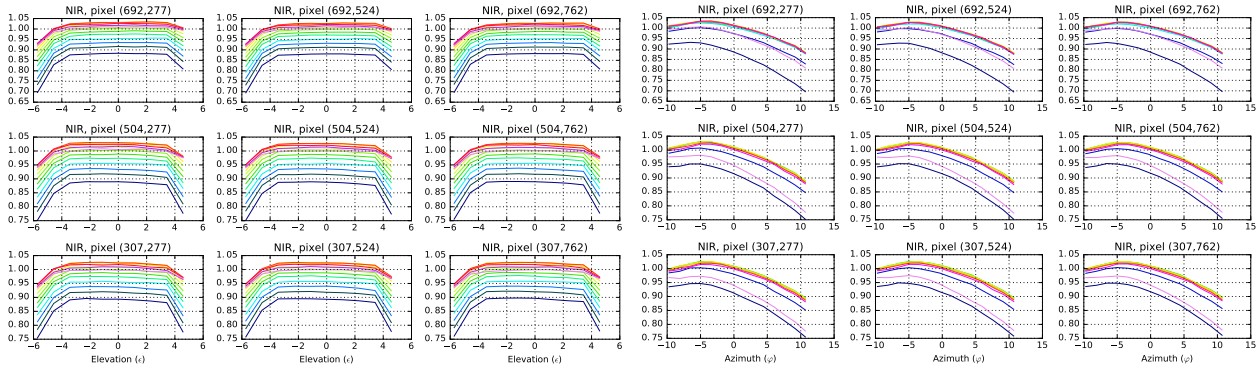

**Figure 37.** The measurements as in Fig. 36 for varying elevation (top half, lines are for different azimuth angles) and azimuth (bottom half, lines are for different elevation angles). The variation of the irradiance as function of the elevation angle shows that for the angles $|\epsilon| > 3.4°$ the measurements have lower signal because the instrument is partially blocked by the entrance baffle. The relative irradiance as a function of the azimuth angle is smooth and well-behaved.

from unity within 0.002 for the NIR and SWIR channel, 0.004 for UVIS and 0.008 for the UV detector for pixels averaged over a small detector area.





The measurements using the nominal diffuser (QVD1) suffered from stimulus stability issues, and were not capable of yielding a reliable CKD. Applying the relative irradiance correction factor from the measurement using QVD2 to this experiment shows improvement of the data, but this is hard to quantify. Both the derivation of a CKD for QVD1 and the validation using the CKD from QVD2 failed.

For these reasons, the calibration of both diffusers will be attempted to be recovered with additional in-flight measurements during the E1 commissioning period after launch.

# 7 Geometric calibration

## 7.1 Pixel response function

The pixel response function (PRF) defines for each pixel the amount of light received from all directions. As the direction
of the light can be defined by two angles, azimuth and elevation, this is a 2-dimensional function. The barycentre of this distribution, i.e. its centre of mass, defines the centrepoint direction from where the detector pixel receives light, and is called the line-of-sight, or LOS.

To be able to determine the PRF per pixel, a star stimulus, which produces a collimated white light beam, is pointed at the Earth port of the instrument. The star stimulus is a $1000\,\mathrm{W}$ Xe source. It is homogenized and collimated to a field of $0.011°$.
The beam is aligned co-linear with an alignment cube on the collimating mirror. The relative orientation of this alignment cube to to the instrument alignment cube is determined with a theodolite.

The instrument is mounted on a cradle which can rotate and tilt. These cradle angles are converted to LOS angles in the instrument reference frame (IRF). For each detector a number of cradle angle settings are measured, covering at least all angle combinations from where any light is received at the detector.

The IRF azimuth and elevation angle of the barycentre of the PRF distribution is not known a priori for each detector pixel. One can however make an initial guess for the 2D azimuth–elevation window around this barycentre which will hold relevant PRF information.

All star stimulus Earth port measurements are processed and annotated with the cradle to IRF alignment correction. Then for all measurements and for each pixel it is checked whether the azimuth and elevation angles of this measurement are within the
azimuth–elevation window determined for that pixel. If it is within the window, the signal value for that pixel is stored with its azimuth and elevation angle. This way, a PRF data product results with dimensions: row $\times$ column $\times$ measurement $\times$ variable $= r \times c \times n_{\mathrm{msmt}} \times v_{s,\phi,\epsilon} = 1025 \times 1024 \times n_{\mathrm{msmt}} \times 3$, the latter being the three variables signal, azimuth and elevation angle.

When this process was completed for the first time, contour and surface plots were made for a number of pixels. The elevation-azimuth window was now checked and adjusted to tightly fit around the useful data, but still including a large enough
area for the entire PRF to fit within the window. In this way, the size of the data product was minimized.

Double measurements are removed such that each azimuth–elevation combination within the measurement grid with a step size of $0.05°$ is present once. Intermediate nan-values are removed, and the volume of each PRF is normalized to 1.





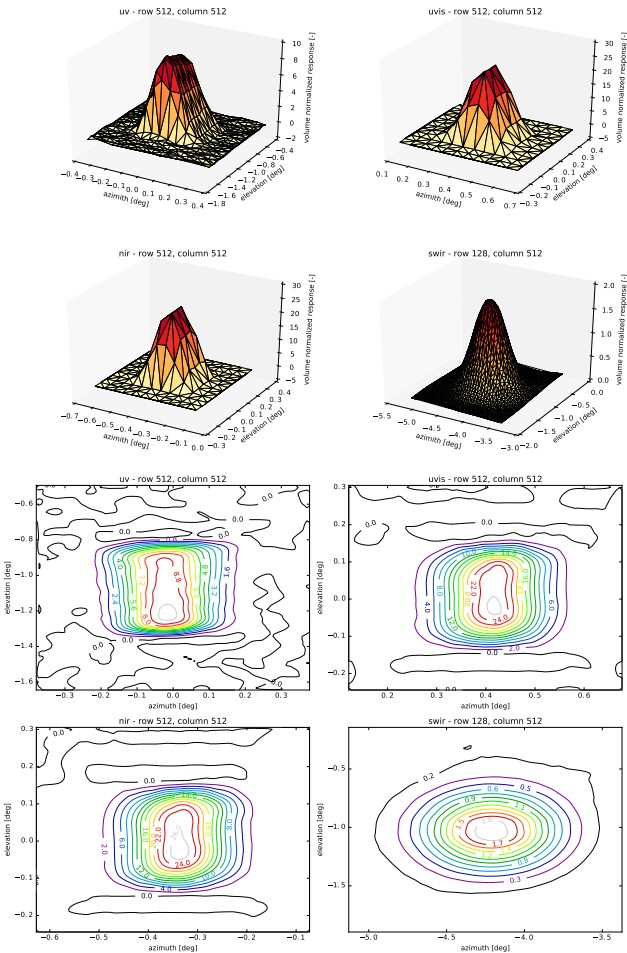

**Figure 38.** PRF surface plot and contour plot for all detectors for a pixel in the middle of the detector. The signal is colour-coded in a tri-surface plot and in a contour plot with a number of iso-lines versus the IRF azimuth–elevation angle window which holds relevant information.

For one pixel in the middle of each detector and one in a corner, a contour plot with 15 iso-lines and a tri-surface plot is shown in Figures 38–39. The contour plot is made from a cubic interpolation of the data to a fine grid. The tri-surface plot draws a node at each data point, and then connects to the closest datapoints in 3D space to form triangular tiles that form a surface.

## 5 7.2 Line-of-sight annotation

A measurement of a single TROPOMI ground pixel consists of multiple spectral pixels, and the ground pixel location associated with it depends thus on the direction of observed light by all detector pixels involved. The CKD needed to determine this ground





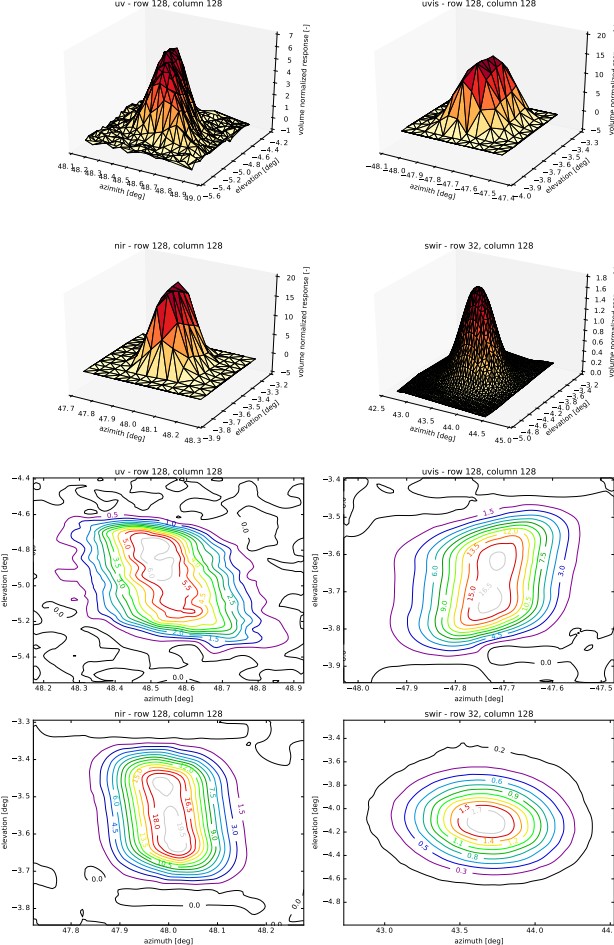

**Figure 39.** PRF surface plot and contour plot for all detectors for a pixel in the corner of the detector. The signal is colour-coded in a tri-surface plot and in a contour plot with a number of iso-lines versus the IRF azimuth–elevation angle window which holds relevant information.

pixel location are the so-called line-of-sight (LOS) angles azimuth and elevation. The geometrical CKD, for each pixel on each detector, contains an elevation and an azimuth angle defined in the instrument reference frame (IRF).

The geolocation algorithm in the L01b data processor determines, for any measurement time instance, from which area on the Earth's surface light was received at a certain TROPOMI detector pixel, using the position and orientation of the spacecraft, and the geometrical CKD. For a more rigorous explanation of the geolocation L01b algorithm the reader is referred to Section 26 and 27 of the L01b ATBD KNMI (2017).

Because the L01b processor only processes one ground pixel per detector row, the average of the LOS angles per detector row is taken. Then, to smooth out measurement uncertainties, a polynomial fit is made of each LOS angle with respect to the detector rows. Part of the detector rows do not receive light during this measurement, while LOS angles of these rows might





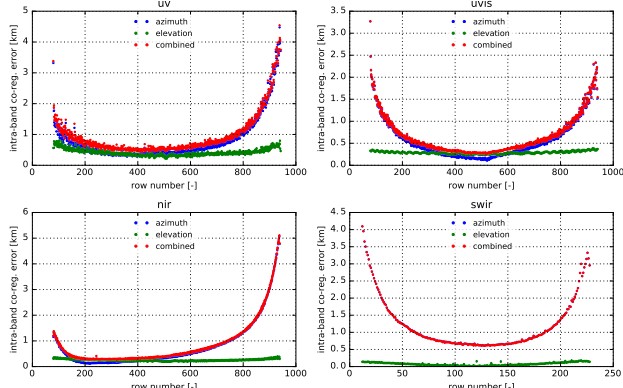

**Figure 40.** Intra-band co-registration error as distance on the surface of the Earth [km] versus row number, for all detectors. For each row, the maximum difference between the elevation (green), azimuth (blue), and the combination of elevation and azimuth (red) is shown for any pixel pair in that row.

be required for L01b processing. Therefore, the polynomial expressions are used to extrapolate the LOS angles over all rows of the detector.

To determine these LOS angles from the calibration measurements a weighted mean method is used. For a certain cradle angle combination, the signal received from the detector will be highest at certain rows and or pixels. This signal for each pixel and each measurement is multiplied with one of the cradle angles corresponding to that measurement and subsequently summed for all measurements. Furthermore, the signal for each pixel is also summed over all measurements separately. The former image is then divided by the latter image, resulting in a weighted mean of the cradle angle for each pixel.

### 7.2.1 Intra-band spatial co-registration

The LOS angles for different detector pixels vary slightly within a single row. A measured spectrum is defined as the data sampled by the detector pixels within a single (binned) row. Because these individual detector pixels all look at slightly different scenes, the wavelengths within a measurement will also stem from slightly different scenes. A chromatic error is thus introduced when averaging these individual detector pixel LOS angles over the row. This effect is referred to as intra-band spatial co-registration.

To quantify the intra-band co-registration error, the minimum and maximum elevation and azimuth of each row is determined. The difference between these extremes can be converted to a distance at the location of the correlating ground pixel. This distance is plotted versus row number in Fig. 40, for both the LOS angles separately and their combination.





### 7.2.2 Inter-band spatial co-registration

For the inter-band co-registration, the ground pixel barycentre locations of different detectors are compared. As the SWIR detector has less rows, the azimuth and elevations values for the 1024 relevant detector rows of the UVN detectors are averaged per 4 rows. This results in 256 rows, exactly the number of significant SWIR rows.

The detectors are aligned with respect to the defined nadir rows, and the difference of all detector pairs in azimuth and elevation angle can be determined, and converted to a distance on the ground at the location of the correlating ground pixel.

This distance is shown for both azimuth and elevation angle and the combination of both in Fig. 41 for all detector pair combinations. The distances increase at the edge of swath due to the combined effect of the LOS not being orthogonal to the Earth and the curvature of the Earth. Between UVIS and NIR the differences are very small. Both the UV and the SWIR spectrometer have an internal slit, this can explain the significant inter-band co-registration error in elevation (along-track) direction between the two.

## 8 Spectral calibration

The spectral calibration consists of the determination of the instrument spectral response function (ISRF) and the wavelength calibration. For the SWIR spectrometer the results from the spectral calibration are reported in van Hees et al. (2017).

For the UVN spectrometers for both CKDs, measurements were performed with spectral line sources (SLS) and a slit function stimulus (SFS).

A SLS produces a limited number of spectral features with a well known wavelength (accuracy $\ll 0.01\,\mathrm{nm}$) and low spectral bandwidth ($\ll 0.01\,\mathrm{nm}$). Measurements using an SLS yield very accurate knowledge on pixel wavelengths near the wavelengths of the spectral lines used. However the characteristic wavelength for detector pixels with wavelengths in between available spectral lines must be estimated by interpolation, which may lead to interpolation errors. For detector pixels beyond the first or last available spectral line within the detector spectral range, the situation is worse, as for these pixels extrapolation is required. In addition, the estimation of the centre of mass of the response for a spectral line may be distorted in case the SLS produces additional spectral lines with a wavelength near the spectral line wavelength. Hence the selection of suitable spectral lines must be performed with care. Two types of spectral line sources were used: A platinum-chromium-neon-argon hollow cathode ($\mathrm{PtCrNeAr}$) type lamp and a mercury-cadmium ($\mathrm{HgCd}$) type lamp.

A slit function stimulus (SFS) consists of an Echelle grating monochromator, producing several spectral lines simultaneously at typically several $\mathrm{nm}$ distance.The absolute wavelength of the lines is only known with limited accuracy, but the relative spectral distances between the lines are well defined. By rotating the Echelle grating, the wavelengths of the produced lines can be shifted in steps of a few hundredth (0.01) of a $\mathrm{nm}$. The measurements from the SFS can be used to yield a wavelength calibration for spectral ranges that are not sufficiently covered by the spectral line sources.

For the ISRF, a pulsed laser produces spectral lines at different wavelengths. The output of the laser can be tuned in the range 210–2600 $\mathrm{nm}$ with stepsizes between 0.05 $\mathrm{nm}$ for the short wavelengths and 1 $\mathrm{nm}$ for the long wavelengths. The bandwidth of



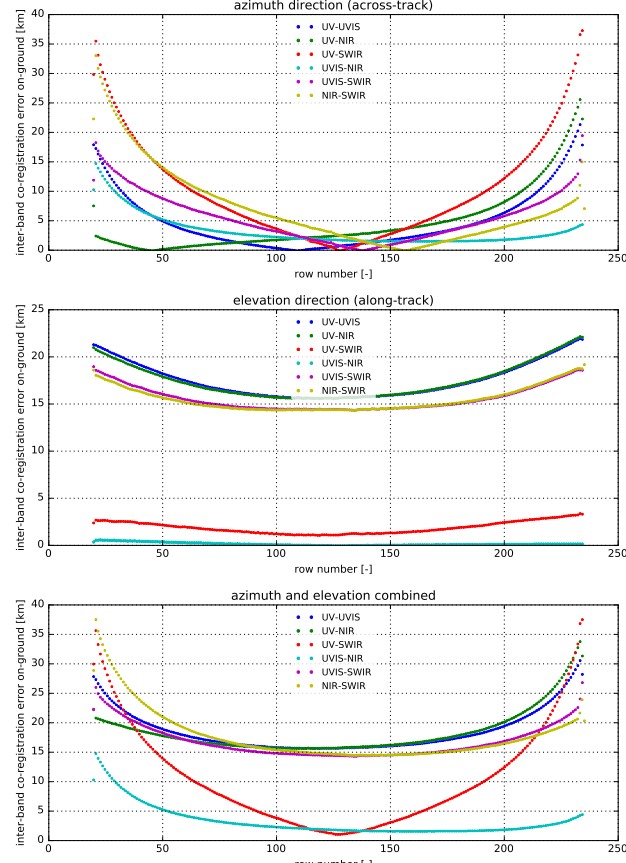

**Figure 41.** Inter-band co-registration error as distance on the surface of the Earth [km] versus row number, for all detector combinations. For UVN, the average of four adjacent rows is taken for comparison to one SWIR row. The large differences between UVIS/NIR and UV/SWIR in elevation direction are as expected from the slit design.

the laser increases towards longer wavelengths. For the NIR the bandwidth is too wide to determine the ISRF, there the SFS is used.

Measurements were performed for both Earth Port and Sun Port (internal diffuser QVD1 only). The stimuli were designed such that for the Earth port a homogeneous field was observed by TROPOMI for at least a number of detector rows per measurement (typically 5 to 7). For the Sun port the field was collimated to mimic the field from the Sun as observed by TROPOMI.

## 8.1 Instrument spectral response function

The instrument spectral response function (ISRF, also known as slit function) characterizes each pixel's spectral response for different wavelengths.





The ISRF is modelled by a function $R_b[\mathbf{x}_{\mathrm{img}}](d\lambda)$, where $\mathbf{x}_{\mathrm{img}}$ represents a detector pixel and $d\lambda$ represents the difference between the observed source wavelength and the pixel characteristic wavelength in nm. The latter wavelength has been defined as the source wavelength for which the centre of mass of the instrument response is at the specified pixel. The shape of the ISRF function is determined by the instrument optics, dispersive elements, and entrance slits, and thus differs per spectrometer.

To obtain the ISRF for the UVN spectrometers measurements were executed with the tunable laser and the SFS. Measurements with the spectral line sources are used to validate the ISRF. Measurements were performed for both Earth Port and Sun Port. By default, the scenes observed by TROPOMI for the Earth Port ISRF measurements were homogeneous within the TROPOMI field-of-view in the flight direction. Some additional measurements were performed to study the influence of inhomogeneous scenes in the flight direction on the ISRF.

The ISRF parametrization step aims at obtaining the ISRF shape for each delta-wavelength. Hence it should accurately interpolate the ISRF shape to the delta-wavelength-grid, and in addition remove any influence of noise in the data points on the obtained ISRF.

The objective for the fit function design is finding a function that inherently satisfies a-priori constraints such as edge values of 0, and provides enough "flexibility" to describe the actual ISRF with sufficient accuracy, i.e. below 1 %. Finding such a

function is not trivial, in particular in case the ISRF is asymmetric. The function eventually used for the UV and NIR detectors is described by the difference between two advanced sigmoid functions that are shifted in wavelength. The resulting function resembles a convolved Gaussian with additional flexibility in the tails. The function is described by the following equation for UV (the $a_2$ parameter indicates the wavelength shift between both sigmoids):

$R_{\mathrm{fit,advanced\_sigmoid}}[d\lambda_s] =$
$$\frac{a_0(1+a_7 d\lambda_s)}{1+e^{-a_3(d\lambda_s-a_2/2)-a_5(d\lambda_s-a_2/2)^3}}$$
$$-\frac{a_0(1+a_7 d\lambda_s)}{1+e^{-a_4(d\lambda_s+a_2/2)-a_6(d\lambda_s+a_2/2)^3}} + a_8 \quad (3)$$

with $d\lambda_s = d\lambda - a_1$ in [nm], the fitted amplitude $a_0$ [nm$^{-1}$], the double sigmoid centre location $a_1$ [nm] and width $a_2$ [nm]. First order ($a_3,a_4$ [nm$^{-1}$]) and third order [$a_5,a_6$ [nm$^{-3}$]) scaling factors for both sigmoids, the skewness of the double

sigmoid function [nm$^{-1}$] and the offset $a_8$ [nm$^{-1}$]. All parameters $a_i$ are fitted. For UVIS a convolved triangle function was used for the v0.1.0 CKD, consisting of a base triangle function that is integrated over a certain window around $d\lambda$. The width of the window could vary with $d\lambda$. Further analysis revealed that a 6$^{\mathrm{th}}$ order generalized exponential function showed slightly lower lack of fit, hence this function was used in the v1.0.0 analysis. It is described by:

$R_{\mathrm{fit,generalized\_exponential}}[d\lambda_s] =$
$$a_0 e^{a_2 d\lambda_s^2 + a_3 d\lambda_s^3 + a_4 d\lambda_s^4 + a_5 d\lambda_s^5 + a_6 d\lambda_s^6} + a_7 \quad (4)$$

with $d\lambda_s = d\lambda - a_1$ in [nm], the fitted amplitude $a_0$ [nm$^{-1}$], the centre location of the ISRF $a_1$ [nm], generalized exponential order parameters of order 2–6 $a_2$–$a_6$ in units [nm$^{-2}$–nm$^{-6}$] and a the fitted offset $a_7$ [nm$^{-1}$].




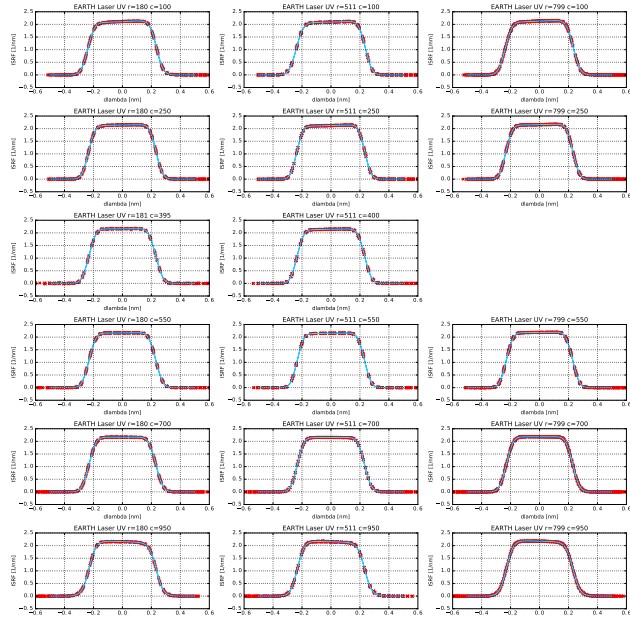

**Figure 42.** Extracted data and parametrized ISRFs for the UV detector, derived from laser measurements.

For NIR, the same fit function as for UV was used (Equation 3), but the skewness parameter $a_7$ was not used in the fit (set to 0). In addition, the parametrization of the ISRFs for NIR from SFS measurements involved an additional convolution step with a modelled SFS line shape: the spectral line shape of spectral lines produced by the SFS is expected to significantly affect the retrieved ISRF shape, which can be checked by comparing differences between ISRFs obtained from the SFS measurements, and ISRFs obtained from SLS measurements.

The ISRF calibration analysis has been performed for the two main stimuli: the slit function stimulus and the tunable laser, and validated using two additional spectral line sources, $PtCrNeAr$ and $HgCd$. The analysis included both Sun port and Earth port measurements. The use of two distinct stimuli has enabled cross-validation and provided redundancy, which has proven to be essential during the calibration campaign, as laser reliability and SNR issues for the SFS reduced the number of usable measurements. The following results were found:

For UV the ISRF shapes via the Earth port are flat-topped, block-like shaped, with a FWHM between 0.45 and 0.5 nm, see Fig. 42. Due to a higher SNR and better detector coverage, the laser measurements have been selected as baseline for the calibration key data for UV, Earth port.

For UVIS the ISRF shapes via the Earth port seem to be a mixture of triangular and Gaussian, with a slight asymmetry, and a FWHM between 0.45 and 0.65 nm, see Fig. 43. Towards higher UVIS wavelengths the FWHM of the laser measurements seems to increase with respect to the SFS measurements, which may indicate a broadening of the measured response bandwidth by the bandwidth of the produced laser line. Due to this potential broadening, the SFS measurements have been selected as baseline for the calibration key data for UVIS Earth port.



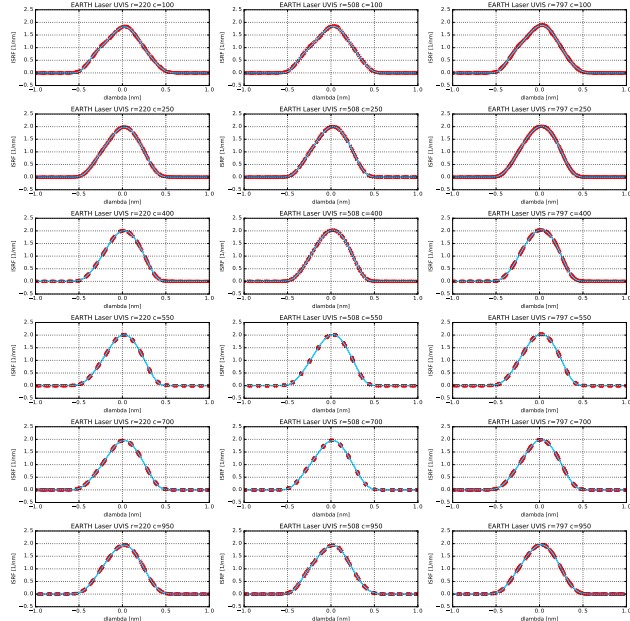

**Figure 43.** Extracted data and parametrized ISRFs for the UVIS detector, derived from laser measurements.

For NIR the ISRF shapes via the Earth port resemble a Gaussian with a rounded top, with a FWHM of 0.34-0.35 nm for the SFS measurements. The laser-derived ISRFs suffer from the expected broadening of lines produced by the stimulus, which increases FWHM to 0.355-0.365 nm, see Fig. 44. Due to the aforementioned broadening, the SFS measurements have been selected as a baseline for the calibration key data for NIR, Earth port.

The accuracy of the parametrized ISRF measured via the Earth port is of the order of 1 % or less for a major part (but not all) of the delta wavelength grid points and detector pixels. ISRF shapes obtained from the Sun port resemble shapes obtained for the Earth port, and average differences in Sun port vs. Earth port FWHM are within 0.6 %.

By changing the cradle tilt angle inhomogeneous illumination fields were created in the along-track direction (across-spectrometer-slit) observed by TROPOMI. The results showed large changes (>10 %) in ISRF shape and centre of mass that

resemble a "cut-off" of part of the response by a missing part of the field. As a consequence the use of the ISRF CKD on scenes with a large scene inhomogeneity as observed by individual pixels may lead to large errors in analysis results. Possible mitigations are marking results from such inhomogeneous scenes as invalid and/or additional modelling of the impact of the scene homogeneity on the ISRF.

## 8.2 Wavelength calibration

Wavelength calibration consists of establishing by means of measurement a characteristic wavelength for each detector pixel, yielding a wavelength map $\lambda_b[\mathbf{x}_{\mathrm{det}}]$ (where $\mathbf{x}_{\mathrm{det}}$ denotes a detector pixel). The term "characteristic wavelength" for a pixel has been defined as the wavelength for which the centre of mass of the instrument response is at the centre of the specified pixel.



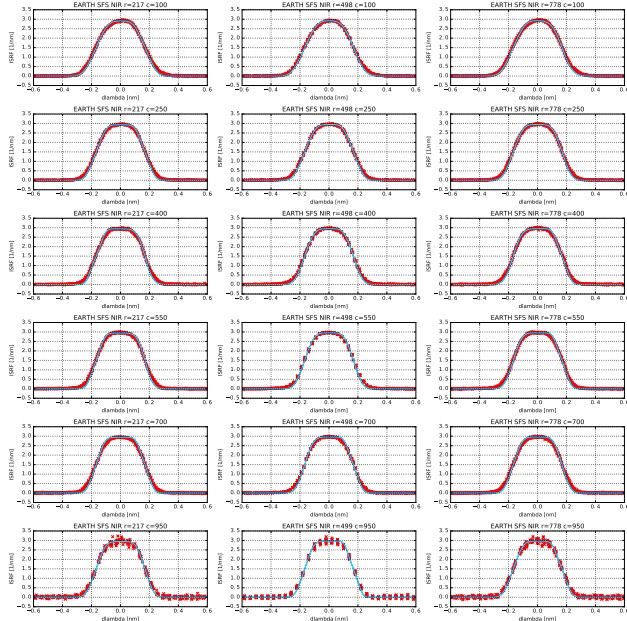

**Figure 44.** Extracted data and parametrized ISRFs for the NIR detector, derived from SFS measurements. The ISRF has been corrected for the line shape in NIR. The dark blue line indicates the parametrized ISRF whereas the light blue line indicates the convolution of ISRF and SFS line shape that was fitted to the data.

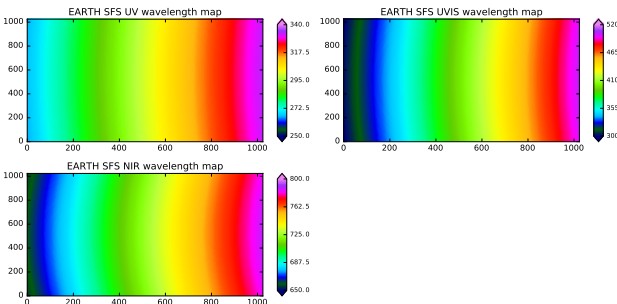

**Figure 45.** Earth port wavelength maps for the UVN detectors, including interpolated/extrapolated areas.

The wavelength calibration depends on external conditions, such as instrument temperature, scene uniformity and instrument stability. The temperature dependency of the spectral calibration has not been studied during the on-ground calibration campaign, primarily due to timing constraints and the presence of thermal control loops on several components within TROPOMI that should reduce the influence of temperature. Some investigative measurements on scene uniformity have been performed, but the spectral calibration is only valid for homogeneous scenes as observed by a detector pixel. The instrument stability has been studied by comparing results for the various measurement series, that were up to 2 months apart.





The spectral calibration measurements performed during the TROPOMI calibration campaign have enabled a wavelength calibration that is accurate over a large area of the detector, thanks to the possibility of cross-calibration and cross-validation of slit function stimulus (SFS) measurements that provide a large spectral coverage, and measurements using two spectral line sources that provide accurate absolute knowledge on wavelengths. The following results were found:

The full (including non-performance) spectral range derived from SFS measurements is 266 to 333 nm for the UV detector, 300 to 502 nm for the UVIS detector, and 656 to 787 nm for the NIR detector. Dispersions are 0.065 nm/pixel for UV, 0.195 nm/pixel for UVIS, and 0.125 nm/pixel for NIR, values that are constant within 5 % over the detector.

From both unintentional issues in field alignment and intentional misalignment to simulate field inhomogeneity in Earth port measurements, it was found that the wavelength calibration is significantly affected by field inhomogeneity such that shifts of
up to 0.1 nm can occur. This dependency is of importance for inhomogeneous scenes that will be observed by the TROPOMI Earth port in-orbit, for example due to the presence of clouds. Measurements with the TROPOMI Earth port observing inhomogeneous fields suggest that the impact of field inhomogeneity on spectral calibration and ISRF can be quantified, provided that the field inhomogeneity is known with sufficient accuracy.

The Sun port SFS measurements have a poor SNR in UV, making the wavelength calibration for UV infeasible. The miti-
gation for this issue is to obtain the Sun port wavelength calibration for UV by adapting (scaling) the Earth port wavelength calibration for UV to the results of the Sun port HgCd SLS measurements in UV.

A comparison of Earth port and Sun port measurements shows agreement of both calibrations to within 0.025 nm for >80 % of the illuminated detector area. Depending on whether the systematic nature of the observed differences is confirmed during the E1 commissioning phase, a separate set of Sun port wavelength calibration key data might be justified. The Earth port SFS
measurements yield a wavelength calibration with fit residuals that are 5, 6, and 4 pm for respectively the UV, UVIS, and NIR detector.

Comparing the Earth port wavelength calibration to the Earth port SLS measurements yields differences typically within ±10 pm, with the exception of the HgCd for the NIR detector that are between -20 and -40 pm. A possible causes for this discrepancy is the occurrence of additional "tails" on either side of the spectral lines produced by the HgCd light source in NIR
that are not found for the PtCrNeAr SLS.

Disregarding the HgCd measurements for NIR, the overall uncertainty of the wavelength calibration is estimated to be of the order of 0.009 nm (9 pm) standard deviation for the Earth port and 0.016 nm (16 pm) for the Sun port.

## 9   Conclusions

All calibration key data for the TROPOMI L01b processor has been derived consistently with the use of the processor itself. The
relative irradiance response calibration and the validation of the geolocation still needs to be performed in-flight. In addition, the out-of-spectral-band straylight correction for the NIR detector has to be validated using in-flight measurements. The impact on L2 retrievals needs to be assessed with in-flight radiance measurements. The results reported here have been accepted by the project to be used for the flight configuration of TROPOMI, launched on October $13^{th}$ 2017.





*Author contributions.* A. Ludewig is optical expert and acted as on-site lead for the KNMI calibration team in Liège, she also planned all the calibration activities. E. Loots as mathematical consultant was responsible for all algorithm definitions and analysed and reported on most electronic calibrations. L. Babić performed the modelling and analysis of the UVN in-band straylight correction. The analysis of the absolute radiometric radiance response was performed by R. Bartstra, while the absolute radiometric response of the irradiance and BSDF

was done by P. Kenter. The relative radiometric response of the radiance and irradiance was done by E. van der Plas, which also included the detectors PRNU. R. Braak was responsible for the development of the tooling required for algorithm implementation in the L01b processor. W. Dierssen and was in charge of the calibration data processing chain and developed the calibration framework together with P. Kenter. R. Landzaat took over the function of W. Dierssen in a later stage. J. Smeets was responsible for the wavelength calibration analysis and the derivation of the UVN ISRF. P.J. Dewitte and D. Schepers supported the on-site calibration effort and were responsible for the inspection

of all measurement data. J. Leloux developed all geometric calibration analysis software and is responsible for the geolocation annotation in the L01b data processor. P. Meijering is responsible for all database engineering required for the calibration processing. D. Schiavini developed part of the correction algorithms in the L01b data processor. N. Rozemeijer acted as system architect and acting lead the L01b data processing development team together with F. Vonk. G. Vacanti derived the calibration accuracy requirements from the higher level system requirements. Q. Kleipool acted as instrument scientist and is project lead of the L01b data processing and calibration development.

P. Veefkind is acting Principal Investigator for the TROPOMI payload on-board the Sentinel-5 Precursor satellite.

    The authors declare that they have no conflict of interest.

*Acknowledgements.* The authors wish to thank the ADSNL calibration team in Liège for the excellent collaboration: Dirk Slootweg, Alexander van Heukelum, Jelle Beetstra, Cees de Haan, Piet Vriend, Barend Ording, Matthijs van der Kooij, Daniel ten Bloemendal, Jan Doornink, Jos Dingjan, Robert Voors, Johan de Vries, Tineke Bakker-van der Veen and Bart Remmerswaal. The authors acknowledge the on-site sup-

port by NSO and ESA in Liège by Harry Förster, Rob Hamann, Sten Ekholm, Charlotte Pachot and Berit Ahlers. We also acknowledge the support of the CSL team by Nathalie Ninane, Marie-Laure Hellin, Fabian Languy, Pascal Blain, Sylvie Liebecq, Christophe Grodent, Pierre Jamotton, Isabelle Domken and Clément Merlin. The work presented in this paper was funded by NSO and ESA.



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
