# Peer review of "Pre-launch calibration results of the TROPOMI payload on-board the Sentinel-5 Precursor satellite"

_Atmospheric Measurement Techniques, 2018_

## Referee Comment (RC1) · Anonymous Referee #1 · 12 Mar 2018

*Pre-launch calibration results of the TROPOMI payload on-board the Sentinel 5 Precursor satellite, Quintus Kleipool et al., MS No.: amt-2018-25,*

**General comments**

Initial paragraph or section evaluating the overall quality of the discussion paper.

The paper is well written and of good quality, with a considerable number of new interesting topics and techniques, and shall certainly be published.
However, I am of the opinion that the quality of the paper can be much improved to be more useful with a comparatively small additional effort, in line with the comments and suggestions provided below. After these comments and suggestions have been adequately addressed, the paper shall certainly be published.

1. On page 1, lines 11-14, a new and promising methodology is introduced in the abstract to quantify residual uncertainties/errors at L1b after 0-1b correction. This point also comes back to some extent in the conclusion section 9. This methodology is to me one of the new important and interesting aspects described in this paper. This methodology can be applied to individual correction factors, as also mentioned in the paper. However, the methodology is not always used consistently throughout the paper, and results of applying this methodology for individual parameters and corrections are not always clearly shown. I feel that the quality of the paper can be improved by improving these aspects and perhaps showing/discussing more results of applying this methodology.
2. The paper discusses the TROPOMI calibration. However, I am of the opinion that the paper would benefit from (briefly) describing a number of critical performance parameters such as signal-to-noise ratio as function of wavelength (for low albedo scenes), spectral/spatial features (from diffusers, coatings, polarisation scrambler, etc.) and polarisation behaviour, even when these parameters are not direct calibration parameters used directly in 0-1b data processing.
3. The title of the paper suggests that the full TROPOMI calibration is described. However, for many parameters the paper focuses on the UV-VIS-NIR spectral range, not on the SWIR wavelength range (there are some exceptions). I propose that the title of the paper is changed to refer to UV-VIS-NIR (preferred), or that a clear reference is given to the remaining parts for the SWIR calibration parameters. See also the examples provided below.
4. Some more comparisons with respect to realistic earth atmosphere low-albedo scenes and signals within absorption peaks shall be presented and included for quantifying stray light at L0 and L1b.
5. The radiometric error budgets presented in table 9 seem somewhat unbalanced / unjustified and in some cases too optimistic. The error budgets in table shall be justified or modified in line with the comments provided below.
6. The intraband and interband coregistration errors don't seem to make sense in view of the spatial sampling distances. This shall be explained in more detail.

**Specific comments**

Section addressing individual scientific questions/issues.

1.
Page 2, line 4:
This is not correct, see also http://www.copernicus.eu/main/overview
I propose to replace this by a quote on that website:
"The Programme is coordinated and managed by the European Commission. It is implemented in partnership with the Member States, the European Space Agency (ESA), the European Organisation for the Exploitation of Meteorological Satellites (EUMETSAT), the European Centre for Medium-Range Weather Forecasts (ECMWF), EU Agencies and Mercator Océan."

2.
Page 2, line 14:
The Sentinel-4 FM1 launch is now planned for 2022. Please correct.

3.
Page 4 line 18 / page 5 line 1:

Please quantify more accurately: "The difference in flight time between the two positions is about 2 seconds".

4.
Page 16, lines 24+25:
Is a non-linearity knowledge of 0.6% compliant with the requirements at L1b? It seems to be rather large. Why is that? Please show some more results from the residuals between measured and fitted curves to quantify the 0.6% (additions to figure 7), also to stress the importance of the new methodology introduced in the abstract (page 1 lines 11-14).

5.
Page 17: Pixel full well capacity.
I guess detector pixel full well capacity in the detector pixels is reached before ADC saturation? Please mention this explicitly. Is this true for all wavelength ranges? Why are the SWIR results not included? If possible, include also SWIR in this section / table.

6.
Section 4.6, detector pixel quality calibration:
Why is SWIR not included? If possible, include also SWIR in this section / table.

7.
Page 20, lines 11+12: Same question as earlier for non-linearity, now for PRNU.
Is a PRNU knowledge of 0.6% compliant with the requirements at L1b? It seems to be rather large. Why is that? "Several validation tests" are mentioned, but no results shown. Please show some more results from the residuals to quantify the 0.6% (additions to figure 8), also to stress the importance of the new methodology introduced in the abstract (page 1 lines 11-14).
Please explain in the text if the PRNU is a purely detector pixel linked effect, or a wavelength linked effect, and why.

8.
Page 22, line 7:
Please quantify the temporal drifts in offset, and the residual errors in L1b for not correcting this effect.

9.
Figure 11:
Please explain what the source is for the blue curves, and why the blue curves seem to have more noise than the red curves for all wavelengths.

10.
Section 6.2, in-band stray light calibration.
Usually signal-to-noise requirements are formulated for low-intensity scenes, i.e. for low albedo scenes in absorption lines. It is fine to report the stray light fractions in the way this is now done in the paper, but these stray light fractions at L0 and L1b shall also be reported with respect to these minimal signals for low albedo and inside the spectral absorption lines, in order to appreciate (quantify) the relative errors in the signals used for fitting L2 data products. Please report stray light fractions at L0 and L1b also (in addition to what is reported now in the paper) with respect to the signals for low albedo, also at wavelengths in the atmospheric absorption lines. Describe clearly (and distinguish between) the various different signal levels used for quantifying stray light fractions at L0 and L1b.
It is acknowledged that the above request is fulfilled to some extent by the hole-in-cloud assessments on pages 28+29, but for these assessments it is not clear what the cloud and hole-in-cloud radiances are and if the radiances in the absorption lines are also accounted for. For example, in the NIR channel significantly higher stray light fractions at L0 and L1b were expected in the O2 absorption bands, but this does not seem to be the case (on the contrary, the stray light fraction at 765 nm is lower). Please explain and quantify and assess what the impact of a hole in the cloud scenario would be on L0 and L1b stray light with a real earth absorption spectrum (low albedo).
In addition, page 29, line 1: Please explain what the spectral / spatial stray light requirements are at L0 and L1b and how they compare with scenes of low albedo and wavelength-dependent signals, also including signals within atmospheric absorption lines.

11.

Section 6.2, in-band stray light calibration.
Please include an overview with quantitative assessments for:
- in-field and in-spectral-band  (correctable) stray light at L0 and L1b.
- in-field and out-of-spectral band (correctable) stray light at L0 and L1b.
- out-of-field (uncorrectable) stray light at L0.

12.
Section 6.2, in-band stray light calibration, table 8, page 28 line 15.
The results in table 8 are applicable for what appears to be a TBD EWLS spectrum. It would be interesting to know what the corresponding numbers would be for a real low-albedo earth spectrum, what stray light correction factors would be obtained.
This would also quantify statements as "a very strong out-of-spectral range straylight contribution" and "This contribution is expected to be smaller in-flight than it is in the on-ground calibration measurements".
Please add some relevant assessments for quantifying L0 and L1b stray light for a real low-albedo earth spectrum.

13.
Section 6.3, out-of-spectral-band straylight.
It would be interesting (essential) to add a number of comparisons between the NIR stray light measurements in TV conditions and ambient conditions: signal-to-noise, dynamic range between measured stray light signal-to-noise and source illumination, stray light as measured between the two.

14.
Section 6.3, out-of-spectral-band straylight, also figure 16.
Please add a plot of the relative stray light (percentage as function of signal at the source wavelength) as function of wavelength in the range 600-1100 nm. It seems virtually all out-of-band stray light in NIR is originating from 620-650 nm and 807-828 nm. Please explain briefly what is causing this, if possible.
Quantify the stray light at L0 and L1b for a hole in the clouds scenario for a low albedo scene from a real earth spectrum, also in earth absorption lines in the NIR wavelength range, for the stray light as shown in figures 15 and 16 (referring to the importance of the new methodology introduced in the abstract (page 1 lines 11-14)).
Quantify the error at L1b in stray light correction accuracy in the NIR wavelength range due to errors in radiance knowledge (since this is out of band) between 620-650 nm and 807-828 nm.

15.
Page 41, figure 23.
The noise shown in these plots is about 1%, suggesting a signal-to-noise ratio of 100. Clarify in the text why this signal-to-noise ratio is so low.

16.
Page 44, figure 25.
Clarify in the text if the gradient observed at e.g. column 512 is also observed in the radiance measurements, which should be the case if it originates from detector quantum efficiency.

17.
Page 44, lines 3+4.
This statement is not agreed / understood, because the distance is referenced with respect to the cross-hair installed in the lamp socket that is used in the same way during calibration at NIST and use during TROPOMI calibration. Please clarify.

18.
Page 45, lines 22-26.
The advantage of the sun simulator would have not been only signal-to-noise, but also a much more flight-representative illumination geometry than a FEL lamp, that emits light to everywhere, because the sun simulator, as the name suggests, would illuminate diffusers more as the sun does. Please clarify.

19.
Page 47, lines 13-15.

The quoted accuracies seem questionable in view of the limitations as described in this paper. It would be interesting (essential to support the statements on accuracy) to show also comparisons between the FEL, integrating sphere and sun simulator measurements for wavelength ranges where this is most useful (also in terms of signal-to-noise). Since for integrating sphere and sun simulator the absolute radiometric scales are not calibrated this exercise would have to include also the BSDF calibration, obviously.

20.
Page 48, lines 3+4, and lines 18-20.
It is written that for bands 1 and 3 the snr (integrating sphere) was too low, but it would still be useful (essential to support statements on accuracy) to show the comparisons for the other bands.
It is not clear how the uncertainties quoted in lines 18-20 are derived / justified. The range in UV is rather large. Clarify how these uncertainties are derived in view of the various FEL, integrating sphere and sun simulator measurements.

21.
Page 49, figures 31+32.
The instrument BSDF should be a property of the differences between earth and sun paths only, i.e. diffusers plus maybe some mirrors. All other contributors drop out in the BSDF. Therefore the BSDF is a smooth function of wavelength. To show this, please plot the FEL-BSDFs in figure 31 as function of wavelength rather than column number, and quantify the differences in the wavelength-band overlap areas.
In addition, compare the FEL BSDF results with those of the integrating sphere for wavelength ranges where this can be done (all bands, except bands 1 and 3?).
These assessments/comparisons should also flow into the uncertainty budgets.

22. Table 9.
There are some questions with respect to table 9.
-   Errors are probably 1-sigma. Please indicate this.
-   Clarify if non-linearity errors (0.6%, page 16) should be included.
-   Clarify if PRNU errors (0.6%, page 20) should be included.
-   Clarify if stray light errors (0.811% UV, 0.527% UVIS, 3.314% NIR, page 28) should be included.
-   The uncertainties quoted for the diffuser calibration are in my view unrealistically low. I would have expected 1-sigma numbers of about 0.5% in UV, 0.4% in UVIS and NIR. Please provide a justification for these low numbers or modify them if necessary.
-   It is not clear why the unexplained measurement discrepancy is given as a rather large range, e.g. 0.0-1.5% in UV, where the high number exceeds by quite a bit the low number given in the total uncertainty ABSRAD and FEL-BSDF. This is not very credible. Please provide a justification for this approach or modify the numbers if necessary (for example by providing a single number of e.g. 1.0% for UV, 0.3% for UVIS and 0.7% for SWIR, similarly to the NIR case).
Furthermore, this table applies to the on-ground calibration (as the paper title suggests, of course), but it is not clear how the numbers given in table 9 would translate into the case for a realistic low-albedo earth spectrum. Please clarify.

23.
Section 6.8, relative irradiance.
The conclusion of this section is that the on-ground calibration measurements were not good enough and that the calibration will have to be (re)done in orbit (page 55, lines 5+6). Is there really an added value for this section? I propose to remove it, or at least shorten it drastically to a few sentences.

24.
Page 58, figure 40.
Figure 40 shows that the coregistration error increases to 4.0 km in UV, 2.0 km for UVIS, 5.0 km in NIR and 3.5 km in SWIR towards the swath edges.
Table 2 gives the across-track and along-track spatial sampling distances for UV, UVIS and NIR of 0.50 degrees (7.2 km) and 0.059 degrees (0.8 km) and 0.16 degrees (2.3 km), respectively, for SWIR.
In view of the numbers given in table 2 the coregistration errors as shown in figure 40 seem to be huge. Please clarify / describe in the text, also highlighting compliance (or not) with the applicable requirements.

25.
Pages 59+60, figure 41.
See also the previous comment.
Interband coregistration errors going in some cases to 10, 20 or 30 km are shown in figure 41. How do these numbers compare with the numbers given in table 2 for across-track and along-track spatial sampling distances and with the applicable requirements (and compliance to those)? Please clarify this in the text.

26.
Section 9, conclusions.
The conclusion section is too short, given the large amount of information presented in this paper. Expand the conclusions with descriptions of what worked well and which accuracies were obtained (or generic) and which problems were encountered and why.
The abstract discussed a new methodology (page 1, lines 11-14), but this concept is not optimally exploited (at least not described) in this paper, not in the conclusions. Consider to expand this.
The statement on "In addition, the out-of-spectral-band straylight correction for the NIR detector has to be validated using in-flight measurements." comes out of the blue, and could have been quantified using the methodology of using the 0-1b processor with real earth atmospheric low-albedo input data. It is not clear how this validation will be done. This sentence is more for section 6.3, where it should be worked out in more detail (see also comment #14), not for the conclusions.

**Technical corrections**

Compact listing of purely technical corrections (typing errors, etc.).

1.
Page 49, figures 31 and 32.
The legends and the figures don't seem to match, because QVD1 seems to be in the left 4 figures, QVD2 in the right 4, unlike the legend states (top vs bottom). Please correct if necessary.

2.
Page 61 shows some equations that are a bit distorted. Please consider correcting this.

---

## Referee Comment (RC2) · Anonymous Referee #2 · 23 Mar 2018

General Comments: This paper is too long. There is a reason that scholarly journals restrict paper lengths to 15 pages, 20 pages at the most. That is because doing so forces the authors to avoid excessive detail and to summarize their findings in a way that helps the reader understand what was performed and what was concluded. The specific details of the TropOMI analysis are of little benefit to readers outside the TropOMI instrument team. No one will attempt to repeat the steps outlined here, so it seems these are included here as a substitute for an internal team report. It is important to describe problems and the general techniques used to address those problems, but by including too much detail the authors fail to provide a useful summary to the readers.

[Figure]

The sections dealing with electronics and with spectral characteristics are well organized and written. The same cannot be said for the sections about radiometric response. These sections would benefit from some hierarchy in the discussions. As it is, the reader is presented with too much detail and not enough overview. What is the calibration philosophy/approach? Why were the measurements performed in the manner they were? Why were the characterized parameters chosen the way they were? These sections could also use more critical evaluation of the results. Do the results make sense? Are the validations sufficient to give us confidence in the error estimates?

General technical: Many of the plots lack axis labels, and some do not even have a description of the axes in the caption. Reference to detector "columns" and "rows" is ubiquitous, and should be replaced more generally with "spectral" or "spatial" dimension.

Specific Technical:

Page 1, Line 20: I don't understand the sentence starting "In case ..." The way this is written implies that there will not be a product problem if random errors are larger than systematic errors. I don't think the authors mean to say this, so I advise a different choice of words. Or simply delete this sentence, because I don't see its relevance in the abstract. The abstract should highlight key points of the paper, and this sentence does not seem to fit that objective.

Page 1, Line 39: I don't understand the term "In-compliance." Do the authors mean non-compliant?

Page 11, Line 3: I don't agree with this description of full-well. Typically, an immediate flattening of the linearity curve indicates register full-well rather than pixel full-well. When the latter occurs it appears as a sharp curve, but over a finite range of integration times. To me, the term "immediate" implies a slope discontinuity in the linearity curve.

[Figure]

Section 6.1: The abbreviation ISRF is not defined until later in the paper.

Section 6.2: This discussion is confusing, and could be clarified by better defining terminology. The authors use the terms in-field, in-band, out-of-field, and far-field but don't clearly explain what stray light falls into each category. This is important because the choice of terms contradicts common definitions used elsewhere. Words like "band" and "range" have subjective interpretations if left undefined. It might be simpler to use the terms spectral and spatial stray light. A schematic or detector image might help to clarify the definitions. From the section title I assume this section pertains to spatial stray light, yet other characterizations are described such as out-of-spectral range.

Where are the detailed descriptions of measurements? This section deserves the same level of detail as Section 6.3 has.

Spatial stray light can be rather difficult to characterize, especially when the instrument is looking out of a chamber through a window. How do you know what portion of the measured SL is contributed by setup and OGSE? Telescope SL is also the simplest of stray light components because it is driven almost entirely by the roughness of the telescope mirrors. Therefore, it is straightforward to model this SL. Have the authors done this as a way to validate their in-band measurements?

The parameters v and w are poorly defined. It sounds like one is spectral and the other spatial, but I cannot tell which is which. This is important for Fig. 14 because the spatial dimension will show the slit image (the telescope stray light) as a stripe illuminating all rows at the source's wavelength. A similar stripe in the spectral dimension can be an indication of a grating defect.

The abbreviation PRF is not defined until later in the paper.

The hole-in-cloud measurement and validation seem to ignore spectral stray light. How is spectral stray light characterized and how is it validated? Past experience with imaging spectrometers has shown that spectral stray light is much more important to science

products than is spatial stray light.

Section 6.3: This type of spectral stray light is more commonly referred to as out-of-range because it is beyond the measurement range of the instrument.

Rather than describing a distinct characteristic of the instrument, as is done with other sections, this one describes a separate measurement campaign. This is confusing, but if the authors feel this needs to be done they should do a better job reconciling this discussion with that of Section 6.2. For example, the authors describe in-band measurements as part of this campaign. Such in-band measurements were also part of the discussion in Section 6.2. Were these the same measurements or different ones. If different, how do they compare? Why was one technique chosen versus the other?

Also, the depth of discussion in this section is in direct contradiction to that of Section 6.2. Section 6.2 has too little description of the measurements and analysis, but Section 6.3 has maybe too much.

Page 18, Line 65: The terms in this equation are not defined.

Figure 16 requires more explanation.

Section 6.4: This section contains multiple subsections, each describing a step in the data reduction. Lacking is a description that ties all these steps together. Why are each of these corrections necessary? Why is it important to separate the radiometric response into low and high frequency components?

The Figure 20 caption is incomplete. What source are we looking at?

Section 6.5: The distinction between ABSRAD and RELRAD is confusing. The authors provide a clear description in Page 24, Lines 4-10. However, Fig. 26 appears to be a combination of ABSRAD and RELRAD, even though the caption talks only of ABSRAD. Furthermore, the BSDF discussion in Section 6.7 is clear about using only ABSRAD, yet Fig. 31 contains row dependence. Does ABSRAD contain RELRAD or not?

[Figure]

Page 27, Lines 3,4: Doesn't this caveat invalidate the distance offset approach the authors are describing? No stray light estimates are provided to prevent the reader from drawing this conclusion.

Section 6.6: This section contains only a brief mention of diffuser feature smoothing. Other than that, there is no discussion of fitting data or separation of high and low frequency components, so the reader must assume this was not undertaken. How is this reconciled with the exhaustive analysis described in Sections 6.4, 6.5 for radiance? Aren't many of the radiance artifacts also present in the irradiance data?

Section 6.7: Given its importance to Level 2 products (as the authors note in lines 39, 40), this should be the primary radiometric description of the paper, yet it appears to be presented only as a validation. Why was so much time and effort placed on the radiance calibration, such as described in Section 6.4 and 6.5, but no effort to ensure that the BSDF calibration is smooth and represents the expected characteristics of the diffusers? The approach taken seems backward, since a smooth, physical BSDF is more important than artifact-free radiances alone. For instance, can the authors explain why the spectral dependence of BSDF has the unusual shapes exhibited in UV and UVIS? And why does it have the structure shown in SWIR? How does the derived BSDF compare to the QVD BRDF?

Page 28, Lines 57, 58: What do these numbers mean and where do they come from? They contradict Figures 30, 31.

Page 38, Lines 71-79 Can the authors speculate why the Earth port and sun port wavelength registration yields significantly different results? This is an unexpected result, is it not?

---

## Author Comment (AC1) · 2 May 2018

**Response to anonymous Referee #1**

| Referee comment | Author's response | Proposed adaptation |
|---|---|---|
| **General comment:** The paper is well written and of good quality, with a considerable number of new interesting topics and techniques, and shall certainly be published. However, I am of the opinion that the quality of the paper can be much improved to be more useful with a comparatively small additional effort, in line with the comments and suggestions provided below. After these comments and suggestions have been adequately addressed, the paper shall certainly be published. | We thank the referee for its thorough review, and hope that our proposed changes will address the comments. | |
| **Comment 1:** On page 1, lines 11-14, a new and promising methodology is introduced in the abstract to quantify residual uncertainties/errors at L1b after 0-1b correction. This point also comes back to some extent in the conclusion section 9. This methodology is to me one of the new important and interesting aspects described in this paper. This methodology can be applied to individual correction factors, as also mentioned in the paper. However, the methodology is not always used consistently throughout the paper, and results of applying this methodology for individual parameters and corrections are not always clearly shown. I feel that the quality of the paper can be improved by improving these aspects and perhaps showing/discussing more results of applying this methodology. | We agree that this new methodology is highly interesting, and we have demonstrated its benefits in a few examples in the paper. We would have liked to show all results, but this would make the paper excessively long. Especially because approximately half of the analysis work on onground calibration went into validation and verification using this method. Thus reporting on these as well would make the paper too long. | We propose to extent the validation and verification analysis on a few extra topics, namely: electronic non-linearity and PRNU. Furthermore we can add a few lines at each section identifying additional validation performed. |
| **Comment 2:** The paper discusses the TROPOMI calibration. However, I am of the opinion that the paper would benefit from (briefly) describing a number of critical performance parameters such as signal-to-noise ratio as function of wavelength (for low albedo scenes), spectral/spatial features (from diffusers, coatings, polarisation scrambler, etc.) and polarisation behaviour, even when these parameters are not direct calibration | agreed | We will update table 3 and the instrument overview with additional parameters. |

| parameters used directly in 0-1b data processing. | | |
|---|---|---|
| **Comment 3:** The title of the paper suggests that the full TROPOMI calibration is described. However, for many parameters the paper focuses on the UV-VIS-NIR spectral range, not on the SWIR wavelength range (there are some exceptions). I propose that the title of the paper is changed to refer to UVVIS-NIR (preferred), or that a clear reference is given to the remaining parts for the SWIR calibration parameters. See also the examples provided below. | This paper covers the calibration of the entire TROPOMI instrument, with the exception of the SWIR detector characterization [Hoogeveen 2013], the SWIR straylight correction [Tol 2017] and SWIR ISRF [van Hees 2017]. All other SWIR calibrations are part of the work presented in this paper (PRNU, RELRAD, ABSRAD, ABSIRR, RELIRR, BSDF, LOS, PRF…). We therefore feel that the title is justified, and propose to leave it as is. | We will update all tables to include the numbers for the SWIR channel as derived in the mentioned references. |
| **Comment 4:** Some more comparisons with respect to realistic earth atmosphere low-albedo scenes and signals within absorption peaks shall be presented and included for quantifying stray light at L0 and L1b. | Unfortunately, we cannot do this with the data available; measuring realistic earth scenes (e.g. zenith sky measurements) was not feasible during onground calibration. Therefore we were forced to restrict the analysis to establishing compliancy with the requirements. These requirements were formulated as the hole-in-the-cloud scene, the closest similarity we can achieve is the scene constructed from EWLS measurements. | We can add some extra detail on why and how the EWLS hole-in-the-cloud validation scene was created and used. |
| **Comment 5:** The radiometric error budgets presented in table 9 seem somewhat unbalanced / unjustified and in some cases too optimistic. The error budgets in table shall be justified or modified in line with the comments provided below. | We can see that this is unclear. The numbers in the table refer to the error in the calibration key data *only*. This error is used in the L01b processor to propagate the total error in the L1b products Radiance and Irradiance. Because the end-user is mostly interested in Reflectance, we have excluded errors (identified with an asterisk) from the CKD as they will cancel out when calculating the Reflectance. | We will adjust the text in the relevant sections to clarify this. It is clear that some extra explanation is needed how the final error in the L1b products is calculated and handled; we will add a paragraph on this. |
| **Comment 6:** The intra-band and inter-band co-registration errors don't seem to make sense in view of the spatial sampling distances. This shall be explained in | Due to the instrument design not all detector pixels observe the same ground scene at the same time. This co-registration mismatch can | We will add some clarification |

| | be large while the spatial sampling distance is small for each individual pixel. | |
|---|---|---|

| Referee comment | Author's response | Proposed adaptation in manuscript |
|---|---|---|
| **Specific comment 1:** Page 2, line 4: This is not correct, see also http://www.copernicus.eu/main/overview I propose to replace this by a quote on that website: "The Programme is coordinated and managed by the European Commission. It is implemented in partnership with the Member States, the European Space Agency (ESA), the European Organisation for the Exploitation of Meteorological Satellites (EUMETSAT), the European Centre for Medium-Range Weather Forecasts (ECMWF), EU Agencies and Mercator Océan." | agreed | We will double check with ESA and change the text. |
| **Specific comment 2:** Page 2, line 14: The Sentinel-4 FM1 launch is now planned for 2022. Please correct. | agreed | We will change the text. |
| **Specific comment 3:** Page 4 line 18 / page 5 line 1: Please quantify more accurately: "The difference in flight time between the two positions is about 2 seconds" | agreed | We will provide the exact time difference at nadir. |
| **Specific comment 4:** Page 16, lines 24+25: Is a non-linearity knowledge of 0.6% compliant with the requirements at L1b? It seems to be rather large. Why is that? Please show some more results from the residuals between measured and fitted curves to quantify the 0.6% (additions to figure 7), also to stress the importance of the new methodology introduced in the abstract (page 1 lines 11-14). | This is indeed an error; the error after validation is a few hundred electrons, far smaller than the 0.6% mentioned. | We will correct the text. |
| **Specific comment 5:** Page 17: Pixel full well capacity. I guess detector pixel full well capacity in the detector pixels is reached before ADC saturation? Please mention this explicitly. Is this true for all wavelength ranges? Why are the SWIR results not included? If possible, include also SWIR in this section / table. | PFW capacity varies per CCD, but is more or less equal for all detector pixel on a CCD. The electronic gain in each band is chosen such that Register Full Well occurs before ADC saturation. The only exception is band 1, in which the fixed gain is so high that PFW can | We will add a comment on the ADC saturation in section 2.7.3. We will also add/quote the results for SWIR. |

| | never be reached, but ADC saturation can. The SWIR PFW was calibrated on unit level by SRON. | |
|---|---|---|
| **Specific comment 6:** Section 4.6, detector pixel quality calibration: Why is SWIR not included? If possible, include also SWIR in this section / table | The SWIR DPQF was calibrated on unit level by SRON. | We will add the SWIR results in table 5. |
| **Specific comment 7:** Page 20, lines 11+12: Same question as earlier for non-linearity, now for PRNU. Is a PRNU knowledge of 0.6% compliant with the requirements at L1b? It seems to be rather large. Why is that? "Several validation tests" are mentioned, but no results shown. Please show some more results from the residuals to quantify the 0.6% (additions to figure 8), also to stress the importance of the new methodology introduced in the abstract (page 1 lines 11-14). Please explain in the text if the PRNU is a purely detector pixel linked effect, or a wavelength linked effect, and why. | This is also an error; the error after validation is a smaller than the 0.6% mentioned. PRNU is a difficult subject to quantify. Fortunately PRNU cancels out in the calculation of the Reflectance. | We will add more validation results and discussion on the accuracy. |
| **Specific comment 8:** Page 22, line 7: Please quantify the temporal drifts in offset, and the residual errors in L1b for not correcting this effect | agreed | We will update the text |
| **Specific comment 9:** Figure 11: Please explain what the source is for the blue curves, and why the blue curves seem to have more noise than the red curves for all wavelengths. | The source of the blue curves is the integrating sphere. These do not have higher noise than the red curves. The cyan curves do; these stem from QTH2 measurements that had severe problems due to the stimulus shape and output. | We will update the text |
| **Specific comment 10:** Section 6.2, in-band stray light calibration. Usually signal-to-noise requirements are formulated for low-intensity scenes, i.e. for low albedo scenes in absorption lines. It is fine to report the stray light fractions in the way this is now done in the paper, but these stray light fractions at L0 and L1b shall also be reported with respect to these minimal signals for low albedo and inside the spectral absorption lines, in order to appreciate (quantify) the relative errors in the signals | See also comment 4. We agree that the straylight correction performance with realistic earth spectra and various albedos is interesting. However, this is out of scope for this paper due to the lack of measured realistic earth scenes, and because all applicable requirements were formulated as a linear fraction at L1b level using the hole-in-cloud scene. This validation scene has no spectral | We will explain in more detail the character of the observed straylight and that spectral features only play a minor role. |

| | | |
|---|---|---|
| used for fitting L2 data products. Please report stray light fractions at L0 and L1b also (in addition to what is reported now in the paper) with respect to the signals for low albedo, also at wavelengths in the atmospheric absorption lines. Describe clearly (and distinguish between) the various different signal levels used for quantifying stray light fractions at L0 and L1b. It is acknowledged that the above request is fulfilled to some extent by the hole-in-cloud assessments on pages 28+29, but for these assessments it is not clear what the cloud and hole-in-cloud radiances are and if the radiances in the absorption lines are also accounted for. For example, in the NIR channel significantly higher stray light fractions at L0 and L1b were expected in the O2 absorption bands, but this does not seem to be the case (on the contrary, the stray light fraction at 765 nm is lower). Please explain and quantify and assess what the impact of a hole in the cloud scenario would be on L0 and L1b stray light with a real earth absorption spectrum (low albedo). in addition, page 29, line 1: Please explain what the spectral / spatial stray light requirements are at L0 and L1b and how they compare with scenes of low albedo and wavelength-dependent signals, also including signals within atmospheric absorption lines. | structure, only spatial. Some L0 performance is presented though. During the inflight commissioning phase the straylight performance will be assessed as suggested, and we plan to report on this in a future paper. | |
| **Specific comment 11:** Section 6.2, in-band stray light calibration. Please include an overview with quantitative assessments for: in-field and in-spectral-band (correctable) stray light at L0 and L1b. in-field and out-of-spectral band (correctable) stray light at L0 and L1b. out-of-field (uncorrectable) stray light at L0. | agreed | We will add a table with these numbers. |
| **Specific comment 12:** Section 6.2, in-band stray light calibration, table 8, page 28 line 15. The results in table 8 are applicable for what appears to be a TBD EWLS spectrum. It would be interesting to know what the corresponding numbers would be for a real low-albedo | Also see comment 4 and 10; this is out of scope for this paper due to the lack of measured realistic earth scenes. During the inflight commissioning phase the straylight performance will be assessed as suggested, | |

| earth spectrum, what stray light correction factors would be obtained. This would also quantify statements as "a very strong out-of-spectral range straylight contribution" and "This contribution is expected to be smaller in-flight than it is in the on-ground calibration measurements". Please add some relevant assessments for quantifying L0 and L1b stray light for a real low-albedo earth spectrum | and we plan to report on this in a future paper. | |
|---|---|---|
| **Specific comment 13:** Section 6.3, out-of-spectral-band straylight. It would be interesting (essential) to add a number of comparisons between the NIR stray light measurements in TV conditions and ambient conditions: signal-to-noise, dynamic range between measured stray light signal-to-noise and source illumination, stray light as measured between the two. | Under TV conditions we only measured with a Xenon lamp with high-pass filter. The source out-of-band spectrum and its power is not known, and therefore only a qualitative assessment is possible. | We can add some extra information regarding dynamic range and noise for the ambient campaign. |
| **Specific comment 14:** Section 6.3, out-of-spectral-band straylight, also figure 16. Please add a plot of the relative stray light (percentage as function of signal at the source wavelength) as function of wavelength in the range 600-1100 nm. It seems virtually all out-of-band stray light in NIR is originating from 620-650 nm and 807-828 nm. Please explain briefly what is causing this, if possible. Quantify the stray light at L0 and L1b for a hole in the clouds scenario for a low albedo scene from a real earth spectrum, also in earth absorption lines in the NIR wavelength range, for the stray light as shown in figures 15 and 16 (referring to the importance of the new methodology introduced in the abstract (page 1 lines 11-14)). Quantify the error at L1b in stray light correction accuracy in the NIR wavelength range due to errors in radiance knowledge (since this is out of band) between 620-650 nm and 807-828 nm | It is correct that all straylight originates from these wavelengths, see figure 16. The instrument prime has not given a conclusive reason where the straylight originates in the optics. During the inflight commissioning phase the straylight performance will be assessed as suggested, and we plan to report on this in a future paper. | We will add explicitly where the source wavelengths are. |
| **Specific comment 15:** Page 41, figure 23. The noise shown in these plots is about 1%, suggesting a signal-to-noise ratio of 100. Clarify in the text why this signal-to- | This is not noise but diffuser features. | We will clarify this in the text. |

| | | |
|---|---|---|
| noise ratio is so low | | |
| **Specific comment 16:** Page 44, figure 25. Clarify in the text if the gradient observed at e.g. column 512 is also observed in the radiance measurements, which should be the case if it originates from detector quantum efficiency. | The observed gradient is the combined result of detector quantum efficiency and optical throughput of the spectrometer. The caption is not explaining this clearly. | We will update the caption. |
| **Specific comment 17:** Page 44, lines 3+4. This statement is not agreed / understood, because the distance is referenced with respect to the crosshair installed in the lamp socket that is used in the same way during calibration at NIST and use during TROPOMI calibration. Please clarify | We agree, we mean that the coil of the FEL lamp extends a few millimeter in the vertical direction. Therefore it is not the ideal point source as we treat it. Therefore the 1/r^2 law will not yield a unique distance for the optical pathlength to and within the internal diffuser. | We will explicitly mention that we cannot locate the exact point inside the volume diffuser due to this problem. |
| **Specific comment 18:** Page 45, lines 22-26. The advantage of the sun simulator would have not been only signal-to-noise, but also a much more flight-representative illumination geometry than a FEL lamp, that emits light to everywhere, because the sun simulator, as the name suggests, would illuminate diffusers more as the sun does. Please clarify. | Agreed. | We will add the field geometry to the sentence. |
| **Specific comment 19:** Page 47, lines 13-15. The quoted accuracies seem questionable in view of the limitations as described in this paper. It would be interesting (essential to support the statements on accuracy) to show also comparisons between the FEL, integrating sphere and sun simulator measurements for wavelength ranges where this is most useful (also in terms of signal-to-noise). Since for integrating sphere and sun simulator the absolute radiometric scales are not calibrated this exercise would have to include also the BSDF calibration, obviously | We do not have a reliable measurement of the instrument BSDF due to instabilities with the Sun Simulator and SNR issues with the integrating sphere. Therefore the BSDF is calculated as the fraction between ABSRAD / ABSIRR. None of these three methods give the same result within the error bars. We are forced to use the FEL measurements, also because they have good SNR. The errors presented are realistic from our point of view, but, these do not include the geometric errors, which we cannot validate due to lack of suitable measurements. We plan to validate this with inflight measurements and report it in a future paper. | We can add a figure with this comparison. |

| | | |
|---|---|---|
| **Specific comment 20:** Page 48, lines 3+4, and lines 18-20. It is written that for bands 1 and 3 the snr (integrating sphere) was too low, but it would still be useful (essential to support statements on accuracy) to show the comparisons for the other bands. It is not clear how the uncertainties quoted in lines 18-20 are derived / justified. The range in UV is rather large. Clarify how these uncertainties are derived in view of the various FEL, integrating sphere and sun simulator measurements | See comment 19. | We can add a figure with this comparison. |
| **Specific comment 21:** Page 49, figures 31+32. The instrument BSDF should be a property of the differences between earth and sun paths only, i.e. diffusers plus maybe some mirrors. All other contributors drop out in the BSDF. Therefore the BSDF is a smooth function of wavelength. To show this, please plot the FEL-BSDFs in figure 31 as function of wavelength rather than column number, and quantify the differences in the wavelength-band overlap areas. In addition, compare the FEL BSDF results with those of the integrating sphere for wavelength ranges where this can be done (all bands, except bands 1 and 3?). These assessments/comparisons should also flow into the uncertainty budgets | The captions and the figures have gotten mixed up. | We will plot the BSDF as a function of wavelength as suggested. |
| **Specific comment 22:** Table 9. There are some questions with respect to table 9. - Errors are probably 1-sigma. Please indicate this. Clarify if non-linearity errors (0.6%, page 16) should be included. Clarify if PRNU errors (0.6%, page 20) should be included. Clarify if stray light errors (0.811% UV, 0.527% UVIS, 3.314% NIR, page 28) should be included. The uncertainties quoted for the diffuser calibration are in my view unrealistically low. I would have expected 1-sigma numbers of about 0.5% in UV, 0.4% in UVIS and NIR. Please provide a justification for these low numbers or modify them if | See general comment 4. We understand that this is unclear. The numbers in the table refer to the error in the calibration key data *only*. This error is used in the L01b processor to propagate the total error in the L1b products Radiance and Irradiance. Because the end-user is mostly interested in Reflectance, we have excluded errors (identified with an asterisk) from the CKD as they will cancel out when calculating the Reflectance. We will double check the reported accuracies | We will adjust the text in the relevant sections to clarify this.

It is clear that some extra explanation is needed how the final error in the L1b products is calculated and handled; we will add a paragraph on this. |

| | for the diffuser calibration. | |
|---|---|---|
| necessary. - It is not clear why the unexplained measurement discrepancy is given as a rather large range, e.g. 0.0-1.5% in UV, where the high number exceeds by quite a bit the low number given in the total uncertainty ABSRAD and FEL-BSDF. This is not very credible. Please provide a justification for this approach or modify the numbers if necessary (for example by providing a single number of e.g. 1.0% for UV, 0.3% for UVIS and 0.7% for SWIR, similarly to the NIR case). Furthermore, this table applies to the on-ground calibration (as the paper title suggests, of course), but it is not clear how the numbers given in table 9 would translate into the case for a realistic low-albedo earth spectrum. Please clarify | The unexplained measurement discrepancy range is the range over the detector; we will clarify this. | |
| **Specific comment 23:** Section 6.8, relative irradiance. The conclusion of this section is that the on-ground calibration measurements were not good enough and that the calibration will have to be (re)done in orbit (page 55, lines 5+6). Is there really an added value for this section? I propose to remove it, or at least shorten it drastically to a few sentences | agreed | We will shorten this substantially. |
| **Specific comment 24:** Page 58, figure 40. Figure 40 shows that the coregistration error increases to 4.0 km in UV, 2.0 km for UVIS, 5.0 km in NIR and 3.5 km in SWIR towards the swath edges. Table 2 gives the across-track and along-track spatial sampling distances for UV, UVIS and NIR of 0.50 degrees (7.2 km) and 0.059 degrees (0.8 km) and 0.16 degrees (2.3 km), respectively, for SWIR. In view of the numbers given in table 2 the coregistration errors as shown in figure 40 seem to be huge. Please clarify / describe in the text, also highlighting compliance (or not) with the applicable requirements | Due to the instrument design not all detector pixels observe the same ground scene at the same time. This co-registration mismatch can be large while the spatial sampling distance is small for each individual pixel. | We will clarify the definitions. |
| **Specific comment 25:** Pages 59+60, figure 41. See also the previous comment. Interband coregistration | See comment 24. | We will clarify the definitions. |

| | | |
|---|---|---|
| errors going in some cases to 10, 20 or 30 km are shown in figure 41. How do these numbers compare with the numbers given in table 2 for across-track and along-track spatial sampling distances and with the applicable requirements (and compliance to those)? Please clarify this in the text | | |
| **Specific comment 26:** Section 9, conclusions. The conclusion section is too short, given the large amount of information presented in this paper. Expand the conclusions with descriptions of what worked well and which accuracies were obtained (or generic) and which problems were encountered and why. The abstract discussed a new methodology (page 1, lines 11-14), but this concept is not optimally exploited (at least not described) in this paper, not in the conclusions. Consider to expand this. The statement on "In addition, the out-of-spectral-band straylight correction for the NIR detector has to be validated using in-flight measurements." comes out of the blue, and could have been quantified using the methodology of using the 0-1b processor with real earth atmospheric low-albedo input data. It is not clear how this validation will be done. This sentence is more for section 6.3, where it should be worked out in more detail (see also comment #14), not for the conclusions | Agreed | We will rewrite the conclusion, and include future work during the commissioning phase. |

| Referee comment | Author's response | Proposed adaptation |
|---|---|---|
| **Technical correction 1:** Page 49, figures 31 and 32. The legends and the figures don't seem to match, because QVD1 seems to be in the left 4 figures, QVD2 in the right 4, unlike the legend states (top vs bottom). Please correct if necessary | This is correct, manuscript versus article style in latex. | We will adapt this. |
| **Technical correction 2:** Page 61 shows some equations that are a bit distorted. Please consider | This is correct, manuscript versus article style in latex. | We will adapt this. |

| correcting this | | |
|---|---|---|

---

## Author Comment (AC2) · 2 May 2018

**Response to anonymous Referee #2**

| Referee comment | Author's response | Proposed adaptation |
|---|---|---|
| **General Comment 1:** This paper is too long. There is a reason that scholarly journals restrict paper lengths to 15 pages, 20 pages at the most. That is because doing so forces the authors to avoid excessive detail and to summarize their findings in a way that helps the reader understand what was performed and what was concluded. The specific details of the TROPOMI analysis are of little benefit to readers outside the TROPOMI instrument team. No one will attempt to repeat the steps outlined here, so it seems these are included here as a substitute for an internal team report. It is important to describe problems and the general techniques used to address those problems, but by including too much detail the authors fail to provide a useful summary to the readers. | We thank the referee for its thorough review and comments. We understand the paper seems unusual long, however, there is a reason for this. The original documentation can never be made publically available due to their proprietary nature. This means that this is the only occasion for the calibration team to report on the results obtained, and how and why some choices were made. We went to great length to reduce the contents of the calibration analysis documentation from 1200 pages to a single paper. We feel that for such an important mission the length of this paper is justified. The level of detail has been reduced so far as possible to remain useful for user of TROPOMI data, and calibration experts of future missions. | We will try to shorten certain sections a bit further, also to make room for some additions requested by the other reviewer. |
| **General Comment 2:** The sections dealing with electronics and with spectral characteristics are well organized and written. The same cannot be said for the sections about radiometric response. These sections would benefit from some hierarchy in the discussions. As it is, the reader is presented with too much detail and not enough overview. What is the calibration philosophy/approach? Why were the measurements performed in the manner they were? Why were the characterized parameters chosen the way they were? These sections could also use more critical evaluation of the results. Do the results make sense? Are the validations sufficient to give us confidence in the error estimates? | We also agree that the radiometric section would benefit from balancing the different topics and parts within a topic. | We will take your questions and suggestions along when restructuring the radiometric part. |

| | | |
|---|---|---|
| **General technical 1:** Many of the plots lack axis labels, and some do not even have a description of the axes in the caption. Reference to detector "columns" and "rows" is ubiquitous, and should be replaced more generally with "spectral" or "spatial" dimension. | agreed | We will recheck all figure and captions. |

| Referee comment | Author's response | Proposed adaptation |
|---|---|---|
| **Technical Comment 1:** Page 1, Line 20: I don't understand the sentence starting "In case : : :" The way this is written implies that there will not be a product problem if random errors are larger than systematic errors. I don't think the authors mean to say this, so I advise a different choice of words. Or simply delete this sentence, because I don't see its relevance in the abstract. The abstract should highlight key points of the paper, and this sentence does not seem to fit that objective | agreed | We will remove the sentence. |
| **Technical Comment 2:** Page 1, Line 39: I don't understand the term "In-compliance." Do the authors mean non-compliant | We assume you mean line 19? | Will change to 'not compliant' |
| **Technical Comment 3:** Page 11, Line 3: I don't agree with this description of full-well. Typically, an immediate flattening of the linearity curve indicates register full-well rather than pixel full-well. When the latter occurs it appears as a sharp curve, but over a finite range of integration times. To me, the term "immediate" implies a slope discontinuity in the linearity curve | We assume you mean page 17, line 1. For TROPOMI, the register full well capacity is about three times larger than the pixel full well capacity as shown in table 6. See also our response to referee #1 Specific comment 5 | We will rephrase the word 'immediate flattening' |
| **Technical Comment 4:** Section 6.1: The abbreviation ISRF is not defined until later in the paper | agreed | Will be added. |
| **Technical Comment 5:** Section 6.2: This discussion is confusing, and could be clarified by better defining | agreed | We will explicitly define the different straylight terms as used |

| | | for TROPOMI. |
|---|---|---|
| terminology. The authors use the terms in-field, in-band, out-of-field, and far-field but don't clearly explain what stray light falls into each category. This is important because the choice of terms contradicts common definitions used elsewhere. Words like "band" and "range" have subjective interpretations if left undefined. It might be simpler to use the terms spectral and spatial stray light. A schematic or detector image might help to clarify the definitions. From the section title I assume this section pertains to spatial stray light, yet other characterizations are described such as out-of-spectral range. | | |
| **Technical Comment 6:** Where are the detailed descriptions of measurements? This section deserves the same level of detail as Section 6.3 has. Spatial stray light can be rather difficult to characterize, especially when the instrument is looking out of a chamber through a window. How do you know what portion of the measured SL is contributed by setup and OGSE? | We have calibrated the out-of-field straylight, and described it on page 26, line 7. | We can add some additional description of the relevant measurements in section 6.2. And also include the commissioning of the setup to address setup straylight. |
| **Technical Comment 7:** Telescope SL is also the simplest of stray light components because it is driven almost entirely by the roughness of the telescope mirrors. Therefore, it is straightforward to model this SL. Have the authors done this as a way to validate their in-band measurements? | Modelling of straylight was partially done for certain components by industrial parties. For the L01b processing this is not sufficient because it requires the total straylight response of the integrated instrument as build and not as designed. | |
| **Technical Comment 8:** The parameters v and w are poorly defined. It sounds like one is spectral and the other spatial, but I cannot tell which is which. This is important for Fig. 14 because the spatial dimension will show the slit image (the telescope stray light) as a stripe illuminating all rows at the source's wavelength. A similar stripe in the spectral dimension can be an indication of a grating defect | agreed | We will add the definition. |
| **Technical Comment 9:** The abbreviation PRF is not defined until later in the paper | agreed | We will include the abbreviation. |

| | | |
|---|---|---|
| **Technical Comment 10:** The hole-in-cloud measurement and validation seem to ignore spectral stray light. How is spectral stray light characterized and how is it validated? Past experience with imaging spectrometers has shown that spectral stray light is much more important to science products than is spatial stray light. | We agree that the hole-in-cloud does not provide information on spectral straylight. The calibration showed that the straylight is dominated by near-field, which has both a spectral and a spatial component. This was measured with a laser source and is the basis of the current straylight correction in the L01b processor. Spectral ghost were sufficient small to not be corrected. | We will add our response to the text. |
| **Technical Comment 11:** Section 6.3: This type of spectral stray light is more commonly referred to as out-of-range because it is beyond the measurement range of the instrument. Rather than describing a distinct characteristic of the instrument, as is done with other sections, this one describes a separate measurement campaign.
This is confusing, but if the authors feel this needs to be done they should do a better job reconciling this discussion with that of Section 6.2. For example, the authors describe in-band measurements as part of this campaign. Such in-band measurements were also part of the discussion in Section 6.2. Were these the same measurements or different ones. If different, how do they compare? Why was one technique chosen versus the other? Also, the depth of discussion in this section is in direct contradiction to that of Section 6.2. Section 6.2 has too little description of the measurements and analysis, but Section 6.3 has maybe too much. | This section indeed describes a different measurement campaign and therefore deserves a different treatment. However, we agree that more effort can be put in the consolidation with section 6.2 | We will adapt both section in line with your comments. |
| **Technical Comment 12:** Page 18, Line 65: The terms in this equation are not defined | Which line number do you mean exactly? | We will recheck all equations nonetheless. |
| **Technical Comment 13:** Figure 16 requires more explanation | agreed | We will expand the caption. |
| **Technical Comment 14:** Section 6.4: This section contains multiple subsections, each describing a step in | agreed | We will restructure and shorten the section. |

| | | |
|---|---|---|
| the data reduction. Lacking is a description that ties all these steps together. Why are each of these corrections necessary? Why is it important to separate the radiometric response into low and high frequency components | | |
| **Technical Comment 15:** The Figure 20 caption is incomplete. What source are we looking at? | Agreed | We will update the caption |
| **Technical Comment 16:** Section 6.5: The distinction between ABSRAD and RELRAD is confusing. The authors provide a clear description in Page 24, Lines 4-10. However, Fig. 26 appears to be a combination of ABSRAD and RELRAD, even though the caption talks only of ABSRAD.
Furthermore, the BSDF discussion in Section 6.7 is clear about using only ABSRAD, yet Fig. 31 contains row dependence. Does ABSRAD contain RELRAD or not | Figure 26 is not a combination; we understand the confusion however, and will explain better in the text.

In the BSDF discussion ABSRAD is normalized with ABSIRR, which has a row dependence. RELRAD does not enter this equation. We will clarify. | We will clarify the text. |
| **Technical Comment 17:** Page 27, Lines 3,4: Doesn't this caveat invalidate the distance offset approach the authors are describing? No stray light estimates are provided to prevent the reader from drawing this conclusion | The line numbers the referee uses do not seem to match the single column manuscript online, but appear to come from a double column version. Is this correct? It makes tracking comments rather difficult. Please clarify which sentence you are commenting on, so we can respond. | |
| **Technical Comment 18:** Section 6.6: This section contains only a brief mention of diffuser feature smoothing. Other than that, there is no discussion of fitting data or separation of high and low frequency components, so the reader must assume this was not undertaken. How is this reconciled with the exhaustive analysis described in Sections 6.4, 6.5 for radiance? Aren't many of the radiance artifacts also present in the irradiance data | The derivation of ABSIRR is indeed less complicated than RELRAD. The latter needs stitching of multiple measurements, and onwards a separation into RELRAD and PRNU without smoothing. For ABSIRR no separation is needed, only a smoothing to remove diffuser features due to speckle. | |
| **Technical Comment 19:** Section 6.7: Given its importance to Level 2 products (as the authors note in lines 39, 40), this should be the primary radiometric | In section 6.7, second paragraph we explain that we would rather have measured the BSDF as a primary calibration parameter, and then to | We will clarify the section and add a figure. |

| | | |
|---|---|---|
| description of the paper, yet it appears to be presented only as a validation. Why was so much time and effort placed on the radiance calibration, such as described in Section 6.4 and 6.5, but no effort to ensure that the BSDF calibration is smooth and represents the expected characteristics of the diffusers? The approach taken seems backward, since a smooth, physical BSDF is more important than artifact-free radiances alone. For instance, can the authors explain why the spectral dependence of BSDF has the unusual shapes exhibited in UV and UVIS? And why does it have the structure shown in SWIR? How does the derived BSDF compare to the QVD BRDF | use in onwards with ABSRAD to yield ABSIRR. The direct BSDF measurement was not possible due to stimulus failure. We were onwards forced to recover by taking the backwards approach of using the FEL lamps. We did check whether the resulting BSDF was artifact-free. On request from reviewer #1 we will already add a figure addressing the smoothness of the BSDF. | |
| **Technical Comment 20:** Page 28, Lines 57, 58: What do these numbers mean and where do they come from? They contradict Figures 30, 31 | We cannot find this due to the line numbering problem. Please clarify so we can respond. | |
| **Technical Comment 21:** Page 38, Lines 71-79 Can the authors speculate why the Earth port and sun port wavelength registration yields significantly different results? This is an unexpected result, is it not | The accuracy of the measurements is about $2/3^{rd}$ of the observed difference between the earth and sun port. Theoretically they should be the same. We cannot determine whether this is significant or not, and will address this after commissioning and report in a future paper. | |

---

## Author Response (AR1)

**Response to anonymous Referee #1**

| Referee comment | Author's response | Rework performed |
|---|---|---|
| **General comment:** The paper is well written and of good quality, with a considerable number of new interesting topics and techniques, and shall certainly be published. However, I am of the opinion that the quality of the paper can be much improved to be more useful with a comparatively small additional effort, in line with the comments and suggestions provided below. After these comments and suggestions have been adequately addressed, the paper shall certainly be published. | We thank the referee for its thorough review, and hope that our proposed changes will address the comments. | In line with the comments from both reviewers we have thoroughly restructured the paper, and shortened where possible. All figures were newly made too.

We thank the referee for this comment, as we think the paper looks better now. |
| **Comment 1:** On page 1, lines 11-14, a new and promising methodology is introduced in the abstract to quantify residual uncertainties/errors at L1b after 0-1b correction. This point also comes back to some extent in the conclusion section 9. This methodology is to me one of the new important and interesting aspects described in this paper. This methodology can be applied to individual correction factors, as also mentioned in the paper. However, the methodology is not always used consistently throughout the paper, and results of applying this methodology for individual parameters and corrections are not always clearly shown. I feel that the quality of the paper can be improved by improving these aspects and perhaps showing/discussing more results of applying this methodology. | We agree that this new methodology is highly interesting, and we have demonstrated its benefits in a few examples in the paper. We would have liked to show all results, but this would make the paper excessively long. Especially because approximately half of the analysis work on onground calibration went into validation and verification using this method. Thus reporting on these as well would make the paper too long. | To demonstrate the new method we have added the closed loop validation figures for two extra topics, namely: electronic non-linearity and PRNU. We have also add a few lines at important sections identifying additional validation performed. |
| **Comment 2:** The paper discusses the TROPOMI calibration. However, I am of the opinion that the paper | agreed | We have combined tables 1, 2 and 3 into two tables, and added |

| | | |
|---|---|---|
| would benefit from (briefly) describing a number of critical performance parameters such as signal-to-noise ratio as function of wavelength (for low albedo scenes), spectral/spatial features (from diffusers, coatings, polarisation scrambler, etc.) and polarisation behaviour, even when these parameters are not direct calibration parameters used directly in 0-1b data processing. | | additional parameters on signal to noise, detector size and polarization sensitivity. |
| **Comment 3:** The title of the paper suggests that the full TROPOMI calibration is described. However, for many parameters the paper focuses on the UV-VIS-NIR spectral range, not on the SWIR wavelength range (there are some exceptions). I propose that the title of the paper is changed to refer to UVVIS-NIR (preferred), or that a clear reference is given to the remaining parts for the SWIR calibration parameters. See also the examples provided below. | This paper covers the calibration of the entire TROPOMI instrument, with the exception of the SWIR detector characterization [Hoogeveen 2013], the SWIR straylight correction [Tol 2017] and SWIR ISRF [van Hees 2017]. All other SWIR calibrations are part of the work presented in this paper (PRNU, RELRAD, ABSRAD, ABSIRR, RELIRR, BSDF, LOS, PRF…). We therefore feel that the title is justified, and propose to leave it as is. | We have updated all tables to include the numbers for the SWIR channel as derived in the mentioned references. |
| **Comment 4:** Some more comparisons with respect to realistic earth atmosphere low-albedo scenes and signals within absorption peaks shall be presented and included for quantifying stray light at L0 and L1b. | Unfortunately, we cannot do this with the data available; measuring realistic earth scenes (e.g. zenith sky measurements) was not feasible during onground calibration. Therefore we were forced to restrict the analysis to establishing compliancy with the requirements. These requirements were formulated as the hole-in-the-cloud scene, the closest similarity we can achieve is the scene constructed from EWLS measurements. | We added some extra detail on why and how the EWLS hole-in-the-cloud validation scene was created and used. We also explained in more detail why realistic Earth scenes are not included/feasible in this paper. |
| **Comment 5:** The radiometric error budgets presented in table 9 seem somewhat unbalanced / unjustified and in some cases too optimistic. The error budgets in table shall be justified or modified in line with the comments provided below. | We can see that this is unclear. The numbers in the table refer to the error in the calibration key data *only*. This error is used in the L01b processor to propagate the total error in the L1b products Radiance and Irradiance. Because the end-user is mostly interested in Reflectance, we have excluded errors | We have adjusted the text in the relevant sections to clarify this. It is clear that some extra explanation was needed how the final error in the L1b products is calculated and handled; we have |

| | (identified with an asterisk) from the CKD as they will cancel out when calculating the Reflectance. | added a paragraph on this. |
|---|---|---|
| **Comment 6:** The intra-band and inter-band co-registration errors don't seem to make sense in view of the spatial sampling distances. This shall be explained in more detail. | Due to the instrument design not all detector pixels observe the same ground scene at the same time. This co-registration mismatch can be large while the spatial sampling distance is small for each individual pixel. | We have added more clarification |

| Referee comment | Author's response | Rework performed |
|---|---|---|
| **Specific comment 1:** Page 2, line 4: This is not correct, see also http://www.copernicus.eu/main/overview I propose to replace this by a quote on that website: "The Programme is coordinated and managed by the European Commission. It is implemented in partnership with the Member States, the European Space Agency (ESA), the European Organisation for the Exploitation of Meteorological Satellites (EUMETSAT), the European Centre for Medium-Range Weather Forecasts (ECMWF), EU Agencies and Mercator Océan." | agreed | We double checked with ESA and change the text. |
| **Specific comment 2:** Page 2, line 14: The Sentinel-4 FM1 launch is now planned for 2022. Please correct. | agreed | We have changed the text. |
| **Specific comment 3:** Page 4 line 18 / page 5 line 1: Please quantify more accurately: "The difference in flight time between the two positions is about 2 seconds" | agreed | We now provide the exact time difference at nadir. |
| **Specific comment 4:** Page 16, lines 24+25: Is a non-linearity knowledge of 0.6% compliant with the requirements at L1b? It seems to be rather large. Why is that? Please show some more results from the residuals between measured and fitted curves to quantify the 0.6% (additions to figure 7), also to stress the importance of the new methodology introduced in the abstract (page 1 lines 11-14). | This is indeed an error; the error after validation is a few hundred electrons, far smaller than the 0.6% mentioned. | We have corrected the text and added a closed-loop validation figure to support this. |
| **Specific comment 5:** Page 17: Pixel full well capacity. I | PFW capacity varies per CCD, but is more or | We will add a comment in section |

| | | |
|---|---|---|
| guess detector pixel full well capacity in the detector pixels is reached before ADC saturation? Please mention this explicitly. Is this true for all wavelength ranges? Why are the SWIR results not included? If possible, include also SWIR in this section / table. | less equal for all detector pixel on a CCD. The Register Full Well capacity (RFW) is sufficiently large to hold 2 to 3 times the PFW during binning. The electronic gain in each band is chosen such that RFW occurs before ADC saturation. The only exception is band 1, in which the fixed gain is so high that PFW can never be reached, but ADC saturation can. The SWIR PFW was calibrated on unit level by SRON. | 2.7.2. We will also add/quote the results for SWIR. |
| **Specific comment 6:** Section 4.6, detector pixel quality calibration: Why is SWIR not included? If possible, include also SWIR in this section / table | The SWIR DPQF was calibrated on unit level by SRON. | We have added the SWIR results in table 4. |
| **Specific comment 7:** Page 20, lines 11+12: Same question as earlier for non-linearity, now for PRNU. Is a PRNU knowledge of 0.6% compliant with the requirements at L1b? It seems to be rather large. Why is that? "Several validation tests" are mentioned, but no results shown. Please show some more results from the residuals to quantify the 0.6% (additions to figure 8), also to stress the importance of the new methodology introduced in the abstract (page 1 lines 11-14). Please explain in the text if the PRNU is a purely detector pixel linked effect, or a wavelength linked effect, and why. | This is indeed an error; the error after validation is a smaller than the 0.6% mentioned. PRNU is a difficult subject to quantify. PRNU cancels however out in the calculation of the Reflectance. | We have add more validation results and a figure showing the accuracy obtained. |
| **Specific comment 8:** Page 22, line 7: Please quantify the temporal drifts in offset, and the residual errors in L1b for not correcting this effect | Residual errors are sufficiently small not to be corrected for in the L01b data processor, and the drift in offset is addressed by a dynamic correction. | We have clarified this section. |
| **Specific comment 9:** Figure 11: Please explain what the source is for the blue curves, and why the blue curves seem to have more noise than the red curves for all wavelengths. | The source of the blue curves is the integrating sphere. These do not have higher noise than the red curves. The cyan curves do; these stem from QTH2 measurements that had severe problems due to the stimulus shape and output. | Because the slit irregularity correction in the L01b is not needed we have removed this section. |
| **Specific comment 10:** Section 6.2, in-band stray light | See also comment 4. We agree that the | We have explain in more detail |

| | | |
|---|---|---|
| calibration. Usually signal-to-noise requirements are formulated for low-intensity scenes, i.e. for low albedo scenes in absorption lines. It is fine to report the stray light fractions in the way this is now done in the paper, but these stray light fractions at L0 and L1b shall also be reported with respect to these minimal signals for low albedo and inside the spectral absorption lines, in order to appreciate (quantify) the relative errors in the signals used for fitting L2 data products. Please report stray light fractions at L0 and L1b also (in addition to what is reported now in the paper) with respect to the signals for low albedo, also at wavelengths in the atmospheric absorption lines. Describe clearly (and distinguish between) the various different signal levels used for quantifying stray light fractions at L0 and L1b. It is acknowledged that the above request is fulfilled to some extent by the hole-in-cloud assessments on pages 28+29, but for these assessments it is not clear what the cloud and hole-in-cloud radiances are and if the radiances in the absorption lines are also accounted for. For example, in the NIR channel significantly higher stray light fractions at L0 and L1b were expected in the O2 absorption bands, but this does not seem to be the case (on the contrary, the stray light fraction at 765 nm is lower). Please explain and quantify and assess what the impact of a hole in the cloud scenario would be on L0 and L1b stray light with a real earth absorption spectrum (low albedo). in addition, page 29, line 1: Please explain what the spectral / spatial stray light requirements are at L0 and L1b and how they compare with scenes of low albedo and wavelength-dependent signals, also including signals within atmospheric absorption lines. | straylight correction performance with realistic earth spectra and various albedos is interesting. However, this is out of scope for this paper due to the lack of measured realistic earth scenes, and because all applicable requirements were formulated as a linear fraction at L1b level using the hole-in-cloud scene. This validation scene has no spectral structure, only spatial. Some L0 performance is presented though. During the inflight commissioning phase the straylight performance will be assessed as suggested, and we plan to report on this in a future paper. | the character of the observed straylight and that spectral features only play a minor role. This section has been restructured altogether to address more referee comments. |
| **Specific comment 11:** Section 6.2, in-band stray light calibration. Please include an overview with quantitative assessments for: in-field and in-spectral-band | agreed | We have added a table with these numbers. |

| | | |
|---|---|---|
| (correctable) stray light at L0 and L1b. in-field and out-of-spectral band (correctable) stray light at L0 and L1b. out-of-field (uncorrectable) stray light at L0. | | |
| **Specific comment 12:** Section 6.2, in-band stray light calibration, table 8, page 28 line 15. The results in table 8 are applicable for what appears to be a TBD EWLS spectrum. It would be interesting to know what the corresponding numbers would be for a real low-albedo earth spectrum, what stray light correction factors would be obtained. This would also quantify statements as "a very strong out-of-spectral range straylight contribution" and "This contribution is expected to be smaller in-flight than it is in the on-ground calibration measurements". Please add some relevant assessments for quantifying L0 and L1b stray light for a real low-albedo earth spectrum | Also see comment 4 and 10; this is out of scope for this paper due to the lack of measured realistic earth scenes. During the inflight commissioning phase the straylight performance will be assessed as suggested, and we plan to report on this in a future paper. | We have clarified this in the text. |
| **Specific comment 13:** Section 6.3, out-of-spectral-band straylight. It would be interesting (essential) to add a number of comparisons between the NIR stray light measurements in TV conditions and ambient conditions: signal-to-noise, dynamic range between measured stray light signal-to-noise and source illumination, stray light as measured between the two. | Under TV conditions we only measured with a Xenon lamp with high-pass filter. The source out-of-band spectrum and its power is not known, and therefore only a qualitative assessment is possible. | We have added some extra information regarding dynamic range and noise for the ambient campaign. We also explain why the delta campaign does not provide information about in-band straylight which therefore cannot be compared to the results from Liege. |
| **Specific comment 14:** Section 6.3, out-of-spectral-band straylight, also figure 16. Please add a plot of the relative stray light (percentage as function of signal at the source wavelength) as function of wavelength in the range 600-1100 nm. It seems virtually all out-of-band stray light in NIR is originating from 620-650 nm and 807-828 nm. Please explain briefly what is causing this, if possible. Quantify the stray light at L0 and L1b for a hole in the clouds scenario for a low albedo scene from a real earth spectrum, also in earth absorption lines in the NIR | It is correct that all straylight originates from these wavelengths, see figure 16. The instrument prime has not given a conclusive reason where the straylight originates in the optics. During the inflight commissioning phase the straylight performance will be assessed as suggested, and we plan to report on this in a future paper. | We added explicitly where the source wavelengths are. |

| | | |
|---|---|---|
| wavelength range, for the stray light as shown in figures 15 and 16 (referring to the importance of the new methodology introduced in the abstract (page 1 lines 11-14)). Quantify the error at L1b in stray light correction accuracy in the NIR wavelength range due to errors in radiance knowledge (since this is out of band) between 620-650 nm and 807-828 nm | | |
| **Specific comment 15:** Page 41, figure 23. The noise shown in these plots is about 1%, suggesting a signal-to-noise ratio of 100. Clarify in the text why this signal-to-noise ratio is so low | This is not noise but diffuser features. | We have clarified this in the caption. |
| **Specific comment 16:** Page 44, figure 25. Clarify in the text if the gradient observed at e.g. column 512 is also observed in the radiance measurements, which should be the case if it originates from detector quantum efficiency. | The observed gradient is the combined result of detector quantum efficiency and optical throughput of the spectrometer. The caption is not explaining this clearly. | We have clarified this in the caption. |
| **Specific comment 17:** Page 44, lines 3+4. This statement is not agreed / understood, because the distance is referenced with respect to the crosshair installed in the lamp socket that is used in the same way during calibration at NIST and use during TROPOMI calibration. Please clarify | We agree, we mean that the coil of the FEL lamp extends a few millimeter in the vertical direction. Therefore it is not the ideal point source as we treat it. Therefore the 1/r^2 law will not yield a unique distance for the optical pathlength to and within the internal diffuser. | We have explicitly mentioned that we cannot locate the exact point inside the volume diffuser due to this problem. |
| **Specific comment 18:** Page 45, lines 22-26. The advantage of the sun simulator would have not been only signal-to-noise, but also a much more flight-representative illumination geometry than a FEL lamp, that emits light to everywhere, because the sun simulator, as the name suggests, would illuminate diffusers more as the sun does. Please clarify. | Agreed. | We have added the field geometry to the sentence. |
| **Specific comment 19:** Page 47, lines 13-15. The quoted accuracies seem questionable in view of the limitations as described in this paper. It would be interesting (essential to support the statements on accuracy) to show also comparisons between the FEL, integrating sphere and sun simulator measurements for | We do not have a reliable measurement of the instrument BSDF due to instabilities with the Sun Simulator and SNR and setup straylight issues with the integrating sphere. Therefore the BSDF is calculated as the fraction between ABSRAD / ABSIRR. None of these three | We have clarified this problem extensively in the text at various locations. |

| | | |
|---|---|---|
| wavelength ranges where this is most useful (also in terms of signal-to-noise). Since for integrating sphere and sun simulator the absolute radiometric scales are not calibrated this exercise would have to include also the BSDF calibration, obviously | methods give the same result within the error bars. We are forced to use the FEL measurements, also because they have good SNR. The errors presented are realistic from our point of view, but, these do not include the geometric errors, which we cannot validate due to lack of suitable measurements. We plan to validate this with inflight measurements and report it in a future paper. | |
| **Specific comment 20:** Page 48, lines 3+4, and lines 18-20. It is written that for bands 1 and 3 the snr (integrating sphere) was too low, but it would still be useful (essential to support statements on accuracy) to show the comparisons for the other bands. It is not clear how the uncertainties quoted in lines 18-20 are derived / justified. The range in UV is rather large. Clarify how these uncertainties are derived in view of the various FEL, integrating sphere and sun simulator measurements | See comment 19. | We have clarified this BSDF problem extensively in the text at various locations. |
| **Specific comment 21:** Page 49, figures 31+32. The instrument BSDF should be a property of the differences between earth and sun paths only, i.e. diffusers plus maybe some mirrors. All other contributors drop out in the BSDF. Therefore the BSDF is a smooth function of wavelength. To show this, please plot the FEL-BSDFs in figure 31 as function of wavelength rather than column number, and quantify the differences in the wavelength-band overlap areas. In addition, compare the FEL BSDF results with those of the integrating sphere for wavelength ranges where this can be done (all bands, except bands 1 and 3?). These assessments/comparisons should also flow into the uncertainty budgets | The captions and the figures have gotten mixed up. | We have clarified this BSDF problem extensively in the text at various locations. We have added a figure to show that the BSDF is indeed a smooth function of wavelength. The additional figure is still in the column domain, which does not matter because the pixel wavelength grid is highly regular. |
| **Specific comment 22:** Table 9. There are some | See general comment 4. We understand that | We have adjust the text in the |

| | | |
|---|---|---|
| questions with respect to table 9. - Errors are probably 1-sigma. Please indicate this. Clarify if non-linearity errors (0.6%, page 16) should be included. Clarify if PRNU errors (0.6%, page 20) should be included. Clarify if stray light errors (0.811% UV, 0.527% UVIS, 3.314% NIR, page 28) should be included. The uncertainties quoted for the diffuser calibration are in my view unrealistically low. I would have expected 1-sigma numbers of about 0.5% in UV, 0.4% in UVIS and NIR. Please provide a justification for these low numbers or modify them if necessary. - It is not clear why the unexplained measurement discrepancy is given as a rather large range, e.g. 0.0-1.5% in UV, where the high number exceeds by quite a bit the low number given in the total uncertainty ABSRAD and FEL-BSDF. This is not very credible. Please provide a justification for this approach or modify the numbers if necessary (for example by providing a single number of e.g. 1.0% for UV, 0.3% for UVIS and 0.7% for SWIR, similarly to the NIR case). Furthermore, this table applies to the on-ground calibration (as the paper title suggests, of course), but it is not clear how the numbers given in table 9 would translate into the case for a realistic low-albedo earth spectrum. Please clarify | this is unclear. The numbers in the table refer to the error in the calibration key data *only*. This error is used in the L01b processor to propagate the total error in the L1b products Radiance and Irradiance. Because the end-user is mostly interested in Reflectance, we have excluded errors (identified with an asterisk) from the CKD as they will cancel out when calculating the Reflectance. We will double check the reported accuracies for the diffuser calibration. The unexplained measurement discrepancy range is the range over the detector; we will change this to a single number. | relevant sections to clarify this. The diffuser calibration accuracies have been double checked, and were indeed optimistic; we have adjusted them in the table. It is clear that some extra explanation was needed how the final error in the L1b products is calculated and handled; we have added a paragraph on this. |
| **Specific comment 23:** Section 6.8, relative irradiance. The conclusion of this section is that the on-ground calibration measurements were not good enough and that the calibration will have to be (re)done in orbit (page 55, lines 5+6). Is there really an added value for this section? I propose to remove it, or at least shorten it drastically to a few sentences | agreed | We have shortened this substantially; the section still has value because the QVD1 calibration was useable for the early inflight commissioning. |
| **Specific comment 24:** Page 58, figure 40. Figure 40 shows that the coregistration error increases to 4.0 km in UV, 2.0 km for UVIS, 5.0 km in NIR and 3.5 km in SWIR towards the swath edges. Table 2 gives the across-track | Due to the instrument design not all detector pixels observe the same ground scene at the same time. This co-registration mismatch can be large while the spatial sampling distance is | We have clarified the definitions and the text. |

| | | |
|---|---|---|
| and along-track spatial sampling distances for UV, UVIS and NIR of 0.50 degrees (7.2 km) and 0.059 degrees (0.8 km) and 0.16 degrees (2.3 km), respectively, for SWIR. In view of the numbers given in table 2 the coregistration errors as shown in figure 40 seem to be huge. Please clarify / describe in the text, also highlighting compliance (or not) with the applicable requirements | small for each individual pixel. | |
| **Specific comment 25:** Pages 59+60, figure 41. See also the previous comment. Interband coregistration errors going in some cases to 10, 20 or 30 km are shown in figure 41. How do these numbers compare with the numbers given in table 2 for across-track and along-track spatial sampling distances and with the applicable requirements (and compliance to those)? Please clarify this in the text | See comment 24. | We have clarified the definitions and the text. |
| **Specific comment 26:** Section 9, conclusions. The conclusion section is too short, given the large amount of information presented in this paper. Expand the conclusions with descriptions of what worked well and which accuracies were obtained (or generic) and which problems were encountered and why. The abstract discussed a new methodology (page 1, lines 11-14), but this concept is not optimally exploited (at least not described) in this paper, not in the conclusions. Consider to expand this. The statement on "In addition, the out-of-spectral-band straylight correction for the NIR detector has to be validated using in-flight measurements." comes out of the blue, and could have been quantified using the methodology of using the 0-1b processor with real earth atmospheric low-albedo input data. It is not clear how this validation will be done. This sentence is more for section 6.3, where it should be worked out in more detail (see also comment #14), not for the conclusions | Agreed | We have completely rewritten the conclusion, and included future work during the commissioning phase. |

| Referee comment | Author's response | Rework performed |
| --- | --- | --- |
| **Technical correction 1:** Page 49, figures 31 and 32. The legends and the figures don't seem to match, because QVD1 seems to be in the left 4 figures, QVD2 in the right 4, unlike the legend states (top vs bottom). Please correct if necessary | This is correct, manuscript versus article style in latex. | We will leave as is in this manuscript version, but check that in the two column paper version the captions match the figures. |
| **Technical correction 2:** Page 61 shows some equations that are a bit distorted. Please consider correcting this | This is correct, manuscript versus article style in latex. | We will leave as is in this manuscript version, but check that in the two column version the formulae match the figures. |

**Response to anonymous Referee #2**

| Referee comment | Author's response | Rework performed |
|---|---|---|
| **General Comment 1:** This paper is too long. There is a reason that scholarly journals restrict paper lengths to 15 pages, 20 pages at the most. That is because doing so forces the authors to avoid excessive detail and to summarize their findings in a way that helps the reader understand what was performed and what was concluded. The specific details of the TROPOMI analysis are of little benefit to readers outside the TROPOMI instrument team. No one will attempt to repeat the steps outlined here, so it seems these are included here as a substitute for an internal team report. It is important to describe problems and the general techniques used to address those problems, but by including too much detail the authors fail to provide a useful summary to the readers. | We thank the referee for its thorough review and comments.

We agree the paper seems unusual long, however, there is a reason for this. The original documentation can never be made publically available due to their proprietary nature. This means that this is the only occasion for the calibration team to report on the results obtained, and how and why some choices were made. We feel that for such an important mission the length of this paper is unavoidable. | We have restructured the paper and removed technical details wherever possible, also to make room for some additions requested by the other referee. All figures were newly made too.

Doing so we have reduced the paper with 10 pages in manuscript style (including the new additions). We thank the referee for this comment, as we think the paper looks better now. |
| **General Comment 2:** The sections dealing with electronics and with spectral characteristics are well organized and written. The same cannot be said for the sections about radiometric response. These sections would benefit from some hierarchy in the discussions. As it is, the reader is presented with too much detail and not enough overview. What is the calibration philosophy/approach? Why were the measurements performed in the manner they were? Why were the characterized parameters chosen the way they were? These sections could also use more critical evaluation of the results. Do the results make sense? Are the validations sufficient to give us confidence in the error estimates? | We also agree that the radiometric section would benefit from balancing the different topics and parts within a topic. | We have completely restructured the section in a more logical order, and supplied an introduction to explain the philosophy and approach chosen. We also made sure that all topics get a balanced/more equal attention. Details were removed where possible and validation results have been added were necessary. |

| | | |
|---|---|---|
| **General technical 1:** Many of the plots lack axis labels, and some do not even have a description of the axes in the caption. Reference to detector "columns" and "rows" is ubiquitous, and should be replaced more generally with "spectral" or "spatial" dimension. | agreed | We have updated all figures and improved the captions where needed reflecting the meaning of the axis. |

| Referee comment | Author's response | Rework performed |
|---|---|---|
| **Technical Comment 1:** Page 1, Line 20: I don't understand the sentence starting "In case : : :" The way this is written implies that there will not be a product problem if random errors are larger than systematic errors. I don't think the authors mean to say this, so I advise a different choice of words. Or simply delete this sentence, because I don't see its relevance in the abstract. The abstract should highlight key points of the paper, and this sentence does not seem to fit that objective | agreed | We have removed the sentence. |
| **Technical Comment 2:** Page 1, Line 39: I don't understand the term "In-compliance." Do the authors mean non-compliant | We assume you mean line 19? | Will changed to 'not compliant' |
| **Technical Comment 3:** Page 11, Line 3: I don't agree with this description of full-well. Typically, an immediate flattening of the linearity curve indicates register full-well rather than pixel full-well. When the latter occurs it appears as a sharp curve, but over a finite range of integration times. To me, the term "immediate" implies a slope discontinuity in the linearity curve | We assume you mean page 17, line 1. For TROPOMI, the register full well capacity is about three times larger than the pixel full well capacity as shown in table 6. See also our response to referee #1 Specific comment 5 | We rephrased as:" The pixel full well is visible as flattening of the graph of pixel charge versus exposure time and indicates pixel saturation" |
| **Technical Comment 4:** Section 6.1: The abbreviation ISRF is not defined until later in the paper | agreed | added. |
| **Technical Comment 5:** Section 6.2: This discussion is confusing, and could be clarified by better defining | agreed | We have explicitly define the different straylight terms as used |

| | | |
|---|---|---|
| terminology. The authors use the terms in-field, in-band, out-of-field, and far-field but don't clearly explain what stray light falls into each category. This is important because the choice of terms contradicts common definitions used elsewhere. Words like "band" and "range" have subjective interpretations if left undefined. It might be simpler to use the terms spectral and spatial stray light. A schematic or detector image might help to clarify the definitions. From the section title I assume this section pertains to spatial stray light, yet other characterizations are described such as out-of-spectral range. | | for TROPOMI. We have also restructured the straylight discussion and added tables and figures. |
| **Technical Comment 6:** Where are the detailed descriptions of measurements? This section deserves the same level of detail as Section 6.3 has. Spatial stray light can be rather difficult to characterize, especially when the instrument is looking out of a chamber through a window. How do you know what portion of the measured SL is contributed by setup and OGSE? | We have calibrated the out-of-field straylight, and described it on page 26, line 7. It is much smaller than the in-band straylight. | And also included some sentences on the commissioning of the setup to address setup straylight to justify why we believe that the setup straylight is sufficiently small. |
| **Technical Comment 7:** Telescope SL is also the simplest of stray light components because it is driven almost entirely by the roughness of the telescope mirrors. Therefore, it is straightforward to model this SL. Have the authors done this as a way to validate their in-band measurements? | Modelling of straylight was partially done for certain components by industrial parties. For the L01b processing this is not sufficient because it requires the total straylight response of the integrated instrument as build and not as designed. | No rework was performed here. |
| **Technical Comment 8:** The parameters v and w are poorly defined. It sounds like one is spectral and the other spatial, but I cannot tell which is which. This is important for Fig. 14 because the spatial dimension will show the slit image (the telescope stray light) as a stripe illuminating all rows at the source's wavelength. A similar stripe in the spectral dimension can be an indication of a grating defect | agreed | We have add the definition and also remade the figure with better axis and caption. |
| **Technical Comment 9:** The abbreviation PRF is not defined until later in the paper | agreed | We have included the abbreviation. |

| | | |
|---|---|---|
| **Technical Comment 10:** The hole-in-cloud measurement and validation seem to ignore spectral stray light. How is spectral stray light characterized and how is it validated? Past experience with imaging spectrometers has shown that spectral stray light is much more important to science products than is spatial stray light. | We agree that the hole-in-cloud does not provide information on spectral straylight. The calibration showed that the straylight is dominated by near-field, which has both a spectral and a spatial component. This was measured with a laser source and is the basis of the current straylight correction in the L01b processor. Spectral ghost were sufficient small to not be corrected. | We have elaborated on this in the text. |
| **Technical Comment 11:** Section 6.3: This type of spectral stray light is more commonly referred to as out-of-range because it is beyond the measurement range of the instrument. Rather than describing a distinct characteristic of the instrument, as is done with other sections, this one describes a separate measurement campaign.
This is confusing, but if the authors feel this needs to be done they should do a better job reconciling this discussion with that of Section 6.2. For example, the authors describe in-band measurements as part of this campaign. Such in-band measurements were also part of the discussion in Section 6.2. Were these the same measurements or different ones. If different, how do they compare? Why was one technique chosen versus the other? Also, the depth of discussion in this section is in direct contradiction to that of Section 6.2. Section 6.2 has too little description of the measurements and analysis, but Section 6.3 has maybe too much. | This section indeed describes a different measurement campaign and therefore deserves a different treatment. We agree that more effort can be put in the consolidation with section 6.2. | We have reworked both section completely, added a table with results and an additional figure. We also removed unnecessary detail in the out-of-range calibration to balance the detail in both sections. We also explain why two different methods were required, and why the in-band measurements from the delta campaign cannot be compared to the main campaign. |
| **Technical Comment 12:** Page 18, Line 65: The terms in this equation are not defined | Which line number do you mean exactly? | We have rechecked all equations. |
| **Technical Comment 13:** Figure 16 requires more explanation | agreed | We have expanded the caption. |
| **Technical Comment 14:** Section 6.4: This section contains multiple subsections, each describing a step in | agreed | We have restructured and shortened the section. |

| | | |
|---|---|---|
| the data reduction. Lacking is a description that ties all these steps together. Why are each of these corrections necessary? Why is it important to separate the radiometric response into low and high frequency components | | |
| **Technical Comment 15:** The Figure 20 caption is incomplete. What source are we looking at? | Agreed | This figure has been deleted in the restructuring. |
| **Technical Comment 16:** Section 6.5: The distinction between ABSRAD and RELRAD is confusing. The authors provide a clear description in Page 24, Lines 4-10. However, Fig. 26 appears to be a combination of ABSRAD and RELRAD, even though the caption talks only of ABSRAD. Furthermore, the BSDF discussion in Section 6.7 is clear about using only ABSRAD, yet Fig. 31 contains row dependence. Does ABSRAD contain RELRAD or not | Figure 26 is not a combination; we understand the confusion however, and will explain better in the text. In the BSDF discussion ABSRAD is normalized with ABSIRR, which has a row dependence. RELRAD does not enter this equation. We will clarify. | We have clarified the text. |
| **Technical Comment 17:** Page 27, Lines 3,4: Doesn't this caveat invalidate the distance offset approach the authors are describing? No stray light estimates are provided to prevent the reader from drawing this conclusion | Page 45, line 9. Setup straylight is indeed a factor that is hard to quantify, and the point source method indicates that problems exist. It does however give an indication of the error in the calibration, that is included in the total error budget. | This effect is included in the error budget. We have updated table 6. |
| **Technical Comment 18:** Section 6.6: This section contains only a brief mention of diffuser feature smoothing. Other than that, there is no discussion of fitting data or separation of high and low frequency components, so the reader must assume this was not undertaken. How is this reconciled with the exhaustive analysis described in Sections 6.4, 6.5 for radiance? Aren't many of the radiance artifacts also present in the irradiance data | The derivation of ABSIRR is indeed less complicated than RELRAD. The latter needs stitching of multiple measurements, and onwards a separation into RELRAD and PRNU without smoothing. For ABSIRR no separation is needed, only a smoothing to remove diffuser features due to speckle. | We have updated the relevant sections to make this more clear. |
| **Technical Comment 19:** Section 6.7: Given its importance to Level 2 products (as the authors note in lines 39, 40), this should be the primary radiometric description of the paper, yet it appears to be presented | In section 6.7, second paragraph we explain that we would rather have measured the BSDF as a primary calibration parameter, and then to use in onwards with ABSRAD to yield ABSIRR. | We have updated the relevant sections to make this more clear, and added a figure showing the smoothness of the BSDF. |

| | | |
|---|---|---|
| only as a validation. Why was so much time and effort placed on the radiance calibration, such as described in Section 6.4 and 6.5, but no effort to ensure that the BSDF calibration is smooth and represents the expected characteristics of the diffusers? The approach taken seems backward, since a smooth, physical BSDF is more important than artifact-free radiances alone. For instance, can the authors explain why the spectral dependence of BSDF has the unusual shapes exhibited in UV and UVIS? And why does it have the structure shown in SWIR? How does the derived BSDF compare to the QVD BRDF | The direct BSDF measurement was not possible due to stimulus failure. We were onwards forced to recover by taking the backwards approach of using the FEL lamps. We did check whether the resulting BSDF was artifact-free. On request from reviewer #1 we will already add a figure addressing the smoothness of the BSDF. | |
| **Technical Comment 20:** Page 28, Lines 57, 58: What do these numbers mean and where do they come from? They contradict Figures 30, 31 | We cannot find this due to the line numbering problem. Please clarify so we can respond. | ? |
| **Technical Comment 21:** Page 38, Lines 71-79 Can the authors speculate why the Earth port and sun port wavelength registration yields significantly different results? This is an unexpected result, is it not | The accuracy of the measurements is about $2/3^{rd}$ of the observed difference between the earth and sun port. Theoretically they should be the same. We cannot determine whether this is significant or not, and will address this after commissioning and report in a future paper. | We have clarified this in the text. |

---

## Referee Report (RR1)

*Pre-launch calibration results of the TROPOMI payload on-board the Sentinel 5 Precursor satellite, Quintus Kleipool et al., MS No.: amt-2018-25,*

**General comments**

Initial paragraph or section evaluating the overall quality of the discussion paper.

The paper is well written and of good quality, with a considerable number of new interesting topics and techniques, and shall certainly be published.
There remain a number of issues that I think would improve the quality of the paper, also in comparison with similar papers of other missions and instruments (for comparison), in line with the comments and suggestions provided below. After these comments and suggestions have been addressed, the paper shall certainly be published. Since this is the second submission I leave it to the scientific editor to decide if and how he/she wants to proceed with the implementation of the comments and suggestions below.

1. Table 6:
I (still) think that the uncertainties presented in table 6 for absolute radiance, absolute irradiance and BSDF are unrealistically low at 1-sigma, given the complications and issues described in the text in this paper:
a. There is a relatively large unexplained measurement discrepancy that dominates ABSRAD and BSDF.
b. There is a relatively large unexplained FEL lamp discrepancy in ABSIRR and BSDF.
c. The external diffuser calibration accuracy is quite low. It is mentioned in the text that the diffuser(s) have been calibrated twice at two different institutes, but no results or comparisons are given. I would suggest to add a small section with these external diffuser BSDF calibration results, since this is an important contributor to the BSDF accuracy.
d. The text refers in some cases to the fact that the preferred sun beam simulator method could not be used for more accurate BSDF calibration accuracy. This seems to suggest that with a well working sun beam simulator the results could have been much more accurate, while the results with the FEL lamps are already now quite accurate. This seems strange.
e. On page 25 the impact of stray light in TROPOMI measurement data is mentioned, which is explained further in sections 6.6 (in-band stray light) and 6.7 (out-of-spectral-range stray light). It is not clear from the text if stray light was (had to be) corrected for the calculation of the radiometric CKD for which the accuracies are given in table 6. It is not clear what additional uncertainty this would add to the accuracies in table 6. Please explain this in more detail in a few sentences and, if necessary, add a line with uncertainty due to all types of stray light. See also table 7 and 8, which suggest that stray light can add up to 1-5% uncertainty before correction and 1-3% after correction, or figure 25, which seems to suggest even higher uncertainties due to stray light in some wavelength areas (e.g. 450-500 nm, and in NIR).
f. The accuracies listed in table 6 suggest that with the use of the CKD presented in this paper in-orbit comparisons of TROPOMI L1b measurement data to sun irradiance, earth radiance and earth reflectance spectra should agree within some 1-2% at 1-sigma. Do the authors think that this will be the case? To me that seems unlikely, given the above uncertainties.

2. Section 9:
The conclusions are still very qualitative. I think it would be useful to add a few quantitative numbers for some of the key parameters / CKD.

**Specific comments**

Section addressing individual scientific questions/issues.

1.
Page 15, line 4:
In figure 3 the smear correction appears after the dark current correction, while here it seems to be the other way around. Please explain, since I understood from the text that the idea is to use the operational 0-1b data processor in the process of deriving CKD.

2.
Page 15, line 17:
For the detector exposure smear correction the reader is referred to ATBDKNMI (2017) (issue 8.0.0). In that ATBD a rather complex matrix inversion method is described. However, I doubt that this smear correction has been implemented / activated in the 0-1b operational data processing software. Please confirm that this is the case, or else describe (in a few sentences) how the smear is corrected with a simplified approach.

3.
Page 25, line 20. Please clarify in a few words what this "optical feature" is (since it seems to be quite important, since it causes stray light).

4.
Page 27, line 15ff:
I have some concern with the CKD smoothing procedures. This is normally avoided, to avoid that important instrument spectral features are removed from CKD and then show up in L2 fit residues. What are the main reasons for performing this rather unusual CKD smoothing? Is it diffuser (speckle) features, or also other effects? Are the spectral diffuser features mainly from the external diffuser plate, or also (partly) from the internal diffusers? Please explain this is a bit more detail in the text.

5.
Page 31, lines 2-4:
This argument about the FEL lamp cross-hair is a bit strange, since the lamp is calibrated by NIST using the same cross-hair target. I therefore don't quite understand how this specific effect can lead to uncertainties in the distance. Please explain.

6.
Page 35, figure 18:
It is essential to also provide a plot with the BSDF CKD plotted as function of wavelength, showing also the band overlap regions, in order to see if uncertainties exist in the band overlap spectral ranges. Please add such a plot (this should be easy to produce from the data that is already there).

**Technical corrections**

Compact listing of purely technical corrections (typing errors, etc.).

1.
Page 18, figure 7:
The horizontal units are electrons. This is understood and agreed.
The vertical units seem to be unitless [-], with values up to 15000. This is not understood. Is this really unitless, then how is this to be interpreted? Or should it be electrons after all? Please clarify.

---

## Author Response (AR2)

**Response to anonymous Referee #1**

| Referee comment | Author's response | Rework performed |
|---|---|---|
| **General comment:** The paper is well written and of good quality, with a considerable number of new interesting topics and techniques, and shall certainly be published. There remain a number of issues that I think would improve the quality of the paper, also in comparison with similar papers of other missions and instruments (for comparison), in line with the comments and suggestions provided below. After these comments and suggestions have been addressed, the paper shall certainly be published. Since this is the second submission I leave it to the scientific editor to decide if and how he/she wants to proceed with the implementation of the comments and suggestions below. | We would like to thank the referee for his/her valuable review. | Added acknowledgement. |
| **Comment 1 a + b:** Table 6: I (still) think that the uncertainties presented in table 6 for absolute radiance, absolute irradiance and BSDF are unrealistically low at 1-sigma, given the complications and issues described in the text in this paper:
 a. There is a relatively large unexplained measurement discrepancy that dominates ABSRAD and BSDF.
 b. There is a relatively large unexplained FEL lamp discrepancy in ABSIRR and BSDF. | The errors given in the table exclude the errors that are caused by the stimulus. We know these errors exist, but cannot quantify them due to the lack of independent measurements.

 It is true that this is a large source of error in the uncertainty budget, which we do not understand, but nonetheless is real because we could quantify it by measurement repetition. | Made unknown contributions explicit. |
| **Comment 1 c:** The external diffuser calibration accuracy is quite low. It is mentioned in the text that the diffuser(s) have been calibrated twice at two different institutes, but no results or comparisons are given. I would suggest to add a small section with these external diffuser BSDF calibration results, since this is an important contributor to the BSDF accuracy. | We assume that with 'low' you mean good? The obtained accuracy is in our opinion the best that can be achieved for these kind of measurements with the current technology. The error reported is the combined result of both calibrations. | Clarified in the text. |
| **Comment 1 d:** The text refers in some cases to the fact that the preferred sun beam simulator method could not be used for more accurate BSDF calibration accuracy. This seems to suggest that with a well working sun beam simulator the results could have been much more accurate, while the results with the FEL lamps are already now quite accurate. This seems strange. | If the sun beam simulator setup had worked as anticipated, the resulting accuracy of the instrument BSDF calibration would have been much better than it is now. This is due to the fact that with a properly working sun beam BSDF calibration setup all common instrument errors should cancel out, and the main | No rework performed. |

| Referee comment | Author's response | Rework performed |
|---|---|---|
| | source of error remaining is the external diffuser calibration error. As mentioned the latter error is in the range of 0.5% to 1.0%. Also the SNR would have been much higher than the FEL improving the UV and UVIS. Using the FEL lamps however, more error sources and noise enter the calibration and the resulting error is significantly higher, as listed in the table. | |
| **Comment 1 e:** On page 25 the impact of stray light in TROPOMI measurement data is mentioned, which is explained further in sections 6.6 (in-band stray light) and 6.7 (out-of-spectral-range stray light). It is not clear from the text if stray light was (had to be) corrected for the calculation of the radiometric CKD for which the accuracies are given in table 6. It is not clear what additional uncertainty this would add to the accuracies in table 6. Please explain this in more detail in a few sentences and, if necessary, add a line with uncertainty due to all types of stray light. See also table 7 and 8, which suggest that stray light can add up to 1-5% uncertainty before correction and 1-3% after correction, or figure 25, which seems to suggest even higher uncertainties due to stray light in some wavelength areas (e.g. 450-500 nm, and in NIR). | We confirm that both the in-band and out-of-band straylight corrections were applied during the calculation of the radiometric CKDs. This is guaranteed by the use of the production L01b data processor in CKD derivation.

The radiometric calibration itself includes the total response of the instrument, and is therefore internally consistent as long as straylight is corrected.

The fact that the straylight correction is not perfect, does not affect the conclusion because it is considered true signal and only has to be consistent with the ISRF and PRF. | No rework performed. |
| **Comment 1 f:** The accuracies listed in table 6 suggest that with the use of the CKD presented in this paper in-orbit comparisons of TROPOMI L1b measurement data to sun irradiance, earth radiance and earth reflectance spectra should agree within some 1-2% at 1-sigma. Do the authors think that this will be the case? To me that seems unlikely, given the above uncertainties. | The reported uncertainties reflect all error sources in the on-ground calibration as far as we could identify. But they exclude the unknown errors due to known sources. We agree that the reported errors are potentially underestimated. We look forward to the in-orbit comparison for further validation. | We have clarified this at various places in the text. |
| | | |
| **Comment 2:** Section 9:
The conclusions are still very qualitative. I think it would be useful to add a few quantitative numbers for some of the key parameters / CKD. | Agreed. | We repeat the important numbers now in the conclusion. |

| Referee comment | Author's response | Rework performed |
|---|---|---|
| **Specific comment 1:** Page 15, line 4:
In figure 3 the smear correction appears after the dark current correction, while here it seems to be the other way around. | The smear correction in the L01b data processor is implemented after the dark-current correction, which is the correct order. Dark-current is however a special | No further rework. |

| | case because in order to derive the dark-current CKD, the measurements have to corrected for smear first as mentioned in the text. This special condition is handled in the calibration framework during the derivation. | |
|---|---|---|
| **Specific comment 2:** Page 15, line 17:
For the detector exposure smear correction the reader is referred to ATBD KNMI (2017) (issue 8.0.0). In that ATBD a rather complex matrix inversion method is described. However, I doubt that this smear correction has been implemented / activated in the 0-1b operational data processing software. Please confirm that this is the case, or else describe (in a few sentences) how the smear is corrected with a simplified approach. | In addition to the rather complex matrix method a simplified algorithm is also described in the ATBD. However it is indeed not explicitly clear that the latter has been selected for implementation in the L01b data processor. | The ATBD is currently being revised, and we will make it more explicit in the next release. No rework for this paper. |
| **Specific comment 3:** Page 25, line 20. Please clarify in a few words what this "optical feature" is (since it seems to be quite important, since it causes stray light). | Agreed, the optical feature is now thought to be due to scattering at the inside of one of the mounts that holds the last lens to form the image on the detector. | Added to the text. |
| **Specific comment 4:** Page 27, line 15ff:
I have some concern with the CKD smoothing procedures. This is normally avoided, to avoid that important instrument spectral features are removed from CKD and then show up in L2 fit residues. What are the main reasons for performing this rather unusual CKD smoothing? Is it diffuser (speckle) features, or also other effects? Are the spectral diffuser features mainly from the external diffuser plate, or also (partly) from the internal diffusers? Please explain this is a bit more detail in the text. | We agree that smoothing of CKD should be avoided. The smoothing here is indeed for the external diffuser features (speckle) that should not enter the instrument calibration. | Clarified in the text. |
| **Specific comment 5:** Page 31, lines 2-4:
This argument about the FEL lamp cross-hair is a bit strange, since the lamp is calibrated by NIST using the same cross-hair target. I therefore don't quite understand how this specific effect can lead to uncertainties in the distance. Please explain. | The point here is that the lamp is not used at the same distance as it was calibrated at NIST. To port the calibration to another distance the $1/r^2$ law is used, but this is only valid for true point-sources. The FEL lamp's double helicoil is not a point source at the distance used in our setup, and thus an error will be introduced. | We clarified the sentence, and removed the confusing remark about the cross-hair. |
| **Specific comment 6:** Page 35, figure 18:
It is essential to also provide a plot with the BSDF CKD plotted as function of wavelength, showing also the band overlap regions, in order to see if uncertainties exist in the band overlap spectral ranges. Please add such a plot (this should be easy to produce from the data that is already there). | Agreed. | Figure added. |

| Referee comment | Author's response | Rework performed |
|---|---|---|
| **Technical correction 1:** Page 18, figure 7:
The horizontal units are electrons. This is understood and agreed. The vertical units seem to be unitless [-], with values up to 15000. This is not understood. Is this really unitless, then how is this to be interpreted? Or should it be electrons after all? Please clarify. | Agreed, the unit is [electrons] | Figure updated. |

**Response to anonymous Referee #2**

| Referee comment | Author's response | Rework performed |
|---|---|---|
| **General Comment:** The reviewed Revision 5 is significantly improved over Revision 3. Sections are better organized and balanced, and more background information is provided to contextualize the presentation. While no further rewrite of the paper is needed, I have several comments that I would like addressed prior to publication. | We would like to thank the referee for his/her valuable review. | Added acknowledgement. |
| **Comment 1:** The paper contains references to rows and columns throughout. While such terminology is appropriate when discussing detector effects, it is less so for issues such as radiometric response. Some figures, especially but not limited to Fig. 18, would be more informative if plotted versus wavelength. | Many calibrations are indeed detector related. For these no unique wavelength can be assigned due to the row dependent spectral smile. Figures that could be updated are 11, 12, 13, 15 and 18 because they only display the optical axis detector row. | We have updated figures 11, 12, 13, 15 and 18. Data are now plotted as a function of wavelength. Fixed a typo in caption of Fig 18. |
| **Comment 2:** The authors are still underplaying the importance of the BSDF calibration. The revised discussion makes clear there were pre-launch calibration problems that prevented a high-quality set of BSDF calibrations. But the instrument still has BSDF calibrations, as presented in Section 6.3. It is noteworthy that this section precedes sections dealing with relative radiance and irradiance, suggesting that these somehow do not affect the BSDF. There are three basic spectral frequency regimes of interest for the BSDF, listed with increasing importance to science products: wavelength-independent errors, broad wavelength dependence, and spectral structure. The authors present a relative BSDF uncertainty for each sensor channel, but do not say over what spectral frequency these values apply. The authors state the BSDFs are spectrally smooth, so the stated uncertainties clearly do not describe residual spectral structure. Fig. 18 suggests (it would benefit from improved quality) the BSDFs are not perfectly smooth, so the authors should provide an uncertainty component for the residual structure. This section or the conclusions would benefit from some critical discussion of the results. By simply looking at Fig. 18 and with some knowledge of aluminum reflectance it is possible to see significant problems in the UV channel. Are the other channels better understood? | We see that in the process of restructuring the section, it may have become unclear that the RELRAD is also part of the instrument BSDF. We have clarified this in the text.

We have improved the quality of figure 18, and agree that the claim to spectral smoothness is not fully justifiable.

The overall curves of ABSRAD and ABSIRR are similar for each spectrometer, but the shape of these curves differ strongly per spectrometer. These are real instrument properties defined by mirror reflectances, dichroic transmission curves, (graded) coatings and detector coatings. When calculating the BSDF these features do not all cancel out. This could be caused by folding mirror properties or residual speckle on the internal or external diffusers, or artifacts of the optical stimulus.

These features are not understood, and no further validation is possible due to the lack of proper | We have updated the figure, clarified the use of RELRAD, and extended the discussion of the BSDF result in line with our response. |

| | | |
|---|---|---|
| | independent on-ground measurements. Therefore, this constitutes an unknown error we cannot quantify and include in the error budget. | |
| **Comment 3:** Section 6.6, entitled In-band straylight, contains a nice description of terminology but never tells us what in-band refers to. Is it all stray light that is not out-of-range? Or all stray light that is not out-of-band? The authors (in Section 6.7) seem to use these two terms interchangeably. In other words does in-band stray light ignore inter-band stray light, e.g. from VIS wavelengths into UV wavelengths? | Agreed, text is unclear. | We have made the list on page 41 line 6 more specific and added an additional bullet for clarification. Also added specifically the term in-band where needed in the text (page 41 line 29). |
| **Comment 4:** Section 6.6 contains the statement, "The SLRF describes the relative straylight response of the system and is derived from on-ground calibration measurements." Which measurements were used and how were they used? | Agreed, this is not clear. The laser measurements were used to derive the SLRF. | We have clarified the text. (page 42, line 8) |
| **Comment 5:** The discussion in Section 6.6 centers on the EWLS measurements, which evidently did not involve spectral cutoff filters. The omission of information about spectral stray light is glaring, regardless of how the requirements were defined. The numbers shown in Table 8 and in Fig. 25 are therefore rather misleading to the average reader. The authors state at the end of the section that laser PSFs form the basis for the processing correction, yet no quantitative assessment is provided. Even if the authors cannot estimate the effectiveness of the correction, they should report the total stray light correction at several wavelengths within each band. | The order in which we have presented the different calibrations may be confusing. We have chosen to discuss the less important characterizations first, and only later in the section we come to the main calibration of near-field straylight which is done with the laser stimulus. The EWLS is used for various characterizations and also to create validation scenes; cut-off filters have also been used as described on page 41 line 26.

At the end of the section we come to the in-band-near-field straylight which fully covers all *spectral* straylight (and also a spatial component). These are corrected simultaneously by a 2D convolution method. This accuracy of this method is assessed quantitatively as presented in table 8. | We have added a few lines to the text to make this more explicit.

Fixed double reference to a single figure on page 44 line 15. |
| **Comment 6:** Table 8 needs more explanation, either in the caption or in the text. What is the definition of stray light percent (what is the value in the denominator)? Same comment applies to Fig. 25. | For table 8 the definition is the total integrated (straylight)signal outside the direct region divided by the integrated (EWLS) signal inside the direct region.

For the figure 25 the definition is similar to the system requirement for a hole-in-the-cloud scene: the straylight at the center of the hole divided by the EWLS signal outside the hole. | Updated caption table 8 and figure 25. |

| Referee comment | Author's response | Rework performed |
|---|---|---|
| **Technical Comment 1:** Figure 17 appears to be mislabeled. Different lines should be columns instead of rows. | Agreed, caption is correct, but labels in figure are wrong. | Figure updated. |
| **Technical Comment 2:** Section 6.6, paragraph 3. Tol et al. (2018) should be followed by a period or semi-colon | Agreed. | Added semicolon. |
| **Technical Comment 3:** Figure 23 caption. The fifth sentence is poorly formed. | Agreed. | Caption updated. |